# Atomic-scale insights into surface reconstruction and transformation in Co-Cr spinel oxides during the oxygen evolution reaction

Biao He[1], Pouya Hosseini[1,2], Tatiana Priamushko[3], Oliver Trost[4], Eko Budiyanto[5], Christoph Bondue[4], Jonas Schulwitz[6], Aleksander Kostka[7], Harun Tüysüz[5,8], Martin Muhler[6], Serhiy Cherevko[3], Kristina Tschulik[2,4] & Tong Li[1] ✉

Optimizing the activity and longevity of oxygen evolution reaction (OER) electrocatalysts requires an atomic-scale understanding of multiple reconstruction and transformation processes occurring in the surface and sub-surface regions of the electrocatalyst. Herein, a multimodal method combining X-ray absorption fine structure and photoemission spectroscopy, in situ Raman spectroscopy, transmission electron microscopy and atom probe tomography with electrochemical measurements is employed to unveil how the changes in oxidation states, atomic coordination, structure and composition on ~20 nm $CoCr_2O_4$ and $Co_2CrO_4$ spinel nanoparticle surfaces affect OER activity and stability in alkaline media. $CoCr_2O_4$ undergoes an activation process and subsequently retains high OER activity for extended durations. The activation of $CoCr_2O_4$ is induced by a steady and substantial Cr dissolution that facilitates bulk incorporation and intercalation of hydroxide ions, coupled with the highly reversible $(Co_{Td}^{II},Cr)(OH)_2 \leftrightarrow (Co_{Oct}^{III},Cr)OOH$ transformation, which enhances OER activity and stability. In comparison, a ~2 nm thick amorphous self-limiting Cr-based (oxy)hydroxide forms on $Co_2CrO_4$ upon cycling, contributing to OER activity. As OER proceeds, such Cr-based (oxy)hydroxide layers on $Co_2CrO_4$ are depleted from the surfaces, leading to deteriorating activity. Overall, this study demonstrates that continuous Cr dissolution triggers an intercalation-assisted $(Co_{Td}^{II},Cr)(OH)_2 \leftrightarrow (Co_{Oct}^{III},Cr)OOH$ transformation that can promote the OER activity and stability of Co-based spinels.

With ever-increasing energy demands and environmental concerns, access to affordable and renewable energy is essential for sustaining global prosperity and economic growth. Electrocatalytic water splitting, when combined with renewable electricity generation technologies such as fuel cells, is expected to emerge as a low-emission method of sustainable energy conversion and storage[1]. However, improving the efficiency of water electrolysers remains challenging, mainly due to limitations in the cost and performance of electrocatalysts at the anode, where the oxygen evolution reaction (OER) occurs[2]. Although benchmark noble metal-

based oxides ($IrO_2$ and $RuO_2$) exhibit an acceptable combination of activity and stability towards OER in acidic media[3–5], their high cost and scarcity limit their potential for large-scale industrial application. Ni-based electrocatalysts show promising OER stability in alkaline media but the activity is limited[6]. To meet global energy demands, it is thus essential to reduce the cost of electrolysers and develop more affordable, sustainable and efficient OER electrocatalysts.

Co spinel-type oxides have emerged as attractive electrolyser candidates due to their low cost, promising OER performance and robust durability[7–10] over a broad pH range[11]. Their OER activity can be improved by mixing with a secondary 3 d transition metal cation, such as Fe[12–14], Mn[15–21], Cr[22–24], Ni[25,26], etc. However, the role of the secondary 3 d transition metal cation (M) and its synergistic effects with Co sites on the OER activity and stability of Co-M spinels remains elusive. In particular, the effects of Cr on the OER performance of the Co spinels are far less well investigated and understood, compared to that of e.g., Fe[12–14] and Mn[15–21]. Metal-oxygen covalency or $e_g$ electron occupancy theories, developed to predict the OER activity of perovskites, can be employed to explain the effects of 3 d transition metal cations on the OER activity of Co-M spinels[27–30]. Previous work[24] has suggested that adding Cr to $Co_3O_4$ can increase Co-O valency, which promotes electron transfer between Co cations and intermediate adsorbates, thereby enhancing OER activity compared to as-synthesized $Co_3O_4$. Such single descriptors may predict the trend of electrocatalyst performance in the pristine state. However, these descriptors often neglect the effects of dynamic elementary processes such as oxidation, transformation, reconstruction, and dissolution on the changes of the electrocatalytically active surfaces, particularly for OER, during which the surface region undergoes substantial and dynamic changes. For instance, a recent study[23] reported that Cr dissolves in $CoCr_2O_4$, exposing more Co sites and creating oxygen vacancies or defects that promote the formation of Co oxyhydroxides. Despite insightful early studies[5,23,31–33], there is a significant lack of mechanistic understanding of how various 3 d transition metal cations, especially Cr, in Co spinels affect surface reconstruction, transformation, and other elementary processes during OER. Improved understanding is key to constructing molecular dynamic models or developing new descriptors for predicting OER activity and stability. Furthermore, Co-based spinels are particularly complex electrocatalysts. Adding different amounts of Cr alters both the Co oxidation state and the atomic configuration (i.e., tetrahedral or octahedral sites), which in turn affects the thermodynamics and/or kinetics of OER elementary processes. Therefore, this study aims to (i) elucidate the roles of Cr and Co in the surface reconstruction, transformation and other elementary processes of Co-Cr spinels with varying Co/Cr ratios (i.e., $CoCr_2O_4$ and $Co_2CrO_4$); and (ii) determine how these processes affect the activity and stability of Co-Cr spinels towards OER.

To achieve these aims, it is crucial to understand how the surface state of electrocatalysts evolves during the concurrent elementary processes that occur at different reaction stages. This requires a comprehensive correlation of surface oxidation state, structure, composition, elemental distribution, dissolution in electrolytes, and electrochemical behaviors. It is nearly impossible to obtain complete information regarding the surface state of the electrocatalyst using only a single technique. Indeed, even operando and in situ spectroscopy techniques[32,34–36] only provide a few aspects of the required surface details, such as surface oxidation state from X-ray photoelectron spectroscopy (XPS) or chemical species from Infrared spectroscopy. Additionally, electrocatalytic properties vary across a single electrocatalyst particle surface[37], since local surface defects serve as active sites or promote the formation of active species. Acquiring atomic-scale structural and compositional details on the uppermost atomic layers of electrocatalysts remains challenging for most techniques. Therefore, it is essential to establish multimodal methods that

link the oxidation state, structure, morphology and compositional details of electrocatalyst surfaces at the atomic scale with activity and stability during electrocatalytic reactions. Atom probe tomography (APT) has demonstrated its potential in revealing the 3D distribution of individual atoms with a sub-nanometre spatial resolution on electrocatalytically active surfaces during OER[38–42]. Herein, we employ a multimodal method by combining X-ray absorption fine structure spectroscopy (XANES), XPS, in situ Raman spectroscopy, transmission electron microscopy (TEM), APT with electrochemical measurements, and inductively coupled plasma mass spectrometry (ICP-MS) to correlate surface state changes with OER activity and stability. XPS and XANES provide information on the oxidation state of uppermost surfaces (~5 nm) and cation coordination in the bulk volume (~20 μm in depth), respectively. In situ Raman spectroscopy reveals the formation of intermediate species on the bulk electrocatalysts during the electrolytic reaction. TEM and APT provide nanoscale and atomic-scale information on structure, composition, and elemental distribution. Our study reveals that Cr dissolves substantially across almost the entire $CoCr_2O_4$ nanoparticle at the onset of OER, generating cation and oxygen vacancies. These vacancies promote pronounced intercalation and incorporation of hydroxide ions, facilitating a highly reversible $(Co_{Td}^{II}, Cr)(OH)_2 \leftrightarrow (Cr_{Oct}^{III}, Cr)OOH$ transformation, which contributes to the high activity and stability of activated $CoCr_2O_4$. In comparison, a 1–2 nm amorphous Cr-based hydroxide layer forms on $Co_2CrO_4$ nanoparticle surfaces. As OER proceeds, the Cr (oxy)hydroxide layer is depleted, driven by steady Cr dissolution and high solubility, leading to considerable activity deterioration. Overall, this study provides mechanistic insights into how dynamic surface reconstruction and transformation affect the activity and stability of mixed Co-Cr spinel oxides towards OER.

## Results and discussion

### Preliminary structural characterization and electrochemical measurements

$CoCr_2O_4$ and $Co_2CrO_4$ nanoparticles were prepared using a one-step alkali-driven coprecipitation approach, followed by calcination. Both $CoCr_2O_4$ and $Co_2CrO_4$ exhibit a cubic spinel structure ($Fd\bar{3}m$), as revealed by the powder X-ray diffraction (XRD) data shown in Fig. S1a, b. The lattice constant of $CoCr_2O_4$ obtained through Rietveld XRD refinement analysis is $8.309 \pm 0.002$ Å, which is slightly higher than that of $Co_2CrO_4$ ($8.232 \pm 0.002$ Å). This difference is attributed to the fact that $CoCr_2O_4$ contains more Cr(III), which has a larger ionic radius (0.615 Å) compared to Co(III), with a radius of 0.545 Å[43]. Additionally, the sizes of the $CoCr_2O_4$ and $Co_2CrO_4$ nanoparticles are $17.3 \pm 4.6$ nm and $18.5 \pm 5.5$ nm, respectively (Fig. S1c, d). The TEM/energy-dispersive X-ray spectroscopy (EDX) images shown in Fig. S1e, f reveal that Co and Cr are uniformly distributed across the nanoparticles with Co/Cr ratios of $0.6 \pm 0.1$ and $2.3 \pm 0.1$ for $CoCr_2O_4$ and $Co_2CrO_4$ nanoparticle samples, respectively (Table S1).

To compare the electrocatalytic activity of $CoCr_2O_4$ and $Co_2CrO_4$, we performed linear sweep voltammetry (LSV) and cyclic voltammetry (CV) measurements using a rotating disk electrode (RDE) in $O_2$-saturated 1.0 M purified KOH[44] (see details in the Experimental Section). A 90% Ohmic drop (iR) correction was applied to compensate for the potential loss. Fig. 1a, d presents the LSV and CV curves of $CoCr_2O_4$ and $Co_2CrO_4$ after the 1st, 10th, 100th, 500th and 1000th CV cycles from 1.0 to 1.65 V vs. RHE, with current densities normalized to the geometric surface area of the glassy carbon electrode (0.196 $cm^2$). To ensure experimental reproducibility, each electrochemical measurement was conducted at least three times under the same conditions (the deviation from three LSV measurements was plotted in Fig. S2). $CoCr_2O_4$ exhibited slightly better OER activity than $Co_2CrO_4$, as the overpotential of $CoCr_2O_4$ after the first cycle is ~370 mV at 10 mA $cm^{-2}$, lower than that of $Co_2CrO_4$ ($395 \pm 2$ mV) (Fig. 1a, b, black curves). The LSV curves normalized to electrochemical active surface area (ECSA)

 2

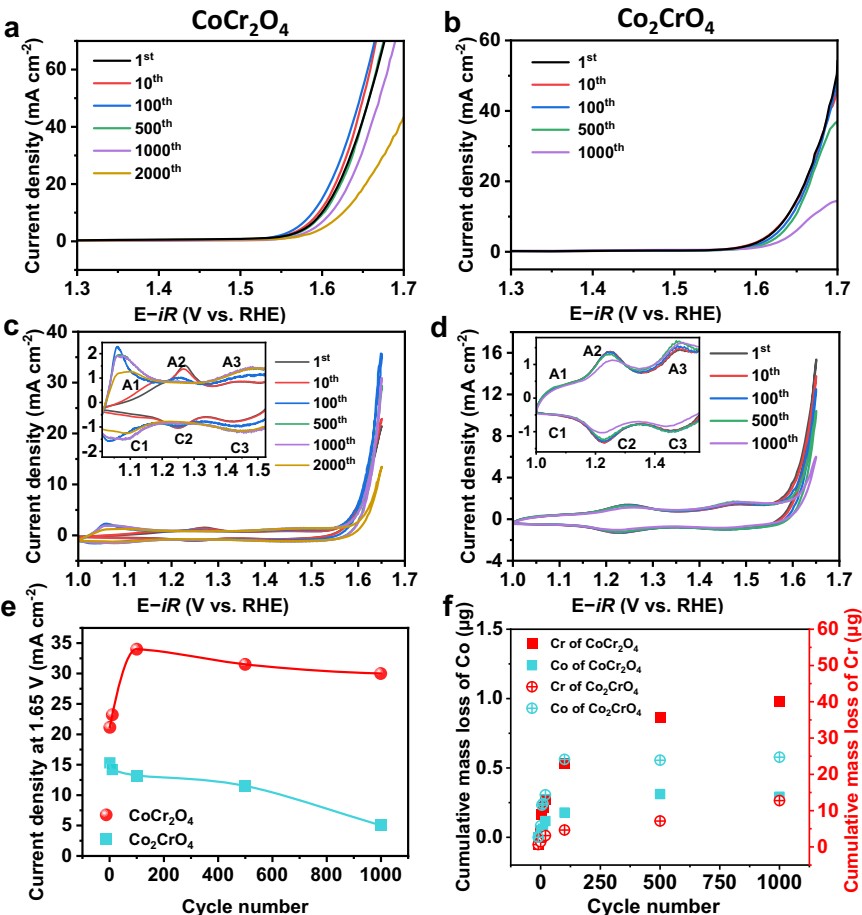

**Fig. 1 | Electrochemical measurements of OER performances for Co-Cr spinel nanoparticles. a**, **b** show the linear sweep voltammetry (LSV) curves recorded at a scan rate of 10 mV s$^{-1}$ of CoCr$_2$O$_4$ nanoparticle after 1, 10, 100, 500, 1000, and 2000 cycles of cyclic voltammetry (CV) measurements and Co$_2$CrO$_4$ nanoparticle after 1, 10, 100, 500, and 1000 cycles of CV measurements, respectively; CV profiles recorded at a scan rate of 50 mV s$^{-1}$ of **c** CoCr$_2$O$_4$ nanoparticle in 1, 10, 100, 500, 1000, and 2000 cycles and **d** Co$_2$CrO$_4$ nanoparticle in pristine, 10, 100, 500 and 1000 cycles, the corresponding current density changes for both samples as a function of CV during 1000 cycles at 1.65 V vs. RHE are given in (**e**); **f** cumulative mass loss comparison of metal dissolution for Co and Cr of CoCr$_2$O$_4$ and Co$_2$CrO$_4$

from the inductively coupled plasma mass spectrometry (ICP-MS) data. ICP-MS data were measured with selected different CV cycles, i.e. after immersing the electrode into the electrolyte (denoted as 0 cycle), after 5, 10, 20, 100, 500, and 1000 CV cycles without refilling the electrolyte. The compensation resistances are 10.5 ± 0.1 Ω for CoCr$_2$O$_4$ and 27.9 ± 0.2 Ω for Co$_2$CrO$_4$. All measurements are conducted in 1 M KOH with a pH value of 14.00 ± 0.01, and on 0.196 cm$^2$ glassy carbon electrode with a mass loading of ~0.05 mg at room temperature. The rotation speed is 1600 rpm and the compensation resistances (90%) are done automatically by the potentiostat.

of pristine CoCr$_2$O$_4$ and Co$_2$CrO$_4$ also indicate a similar trend (Fig. S3a and Fig. S5a). Interestingly, CoCr$_2$O$_4$ is activated during the first 100 CV cycles, as evidenced by a continuous decrease in its overpotential to 343 ± 2 mV (at 10 mA cm$^{-2}$) after 100 cycles (Fig. 1a, blue curve). Additionally, the overpotential of CoCr$_2$O$_4$ increases by only ~10 mV after 1000 cycles and by ~27 mV after 2000 CV cycles compared to the first cycle (Fig. 1a, purple and orange curves), indicating that the OER activity of CoCr$_2$O$_4$ decays rather slowly. In comparison, no activation was observed for Co$_2$CrO$_4$ (Fig. 1b). Despite this, the activity of Co$_2$CrO$_4$ is maintained during the first 500 cycles, as its overpotential at 10 mA cm$^{-2}$ increases by only ~8 mV after 500 cycles (Fig. 1b, green curve). However, after 1000 cycles, the overpotential of Co$_2$CrO$_4$ increases to 445 ± 2 mV(compared to 395 mV after the 1$^{st}$ cycle; Fig. 1b, purple curve), suggesting a pronounced activity deterioration. The pronounced activation of CoCr$_2$O$_4$ in the first 100 cycles was also demonstrated by a rapid increase in OER current densities at 1.65 V vs. RHE in the CV curves (Fig. 1c), as summarized in Fig. 1e. Notably, the OER current densities of CoCr$_2$O$_4$ remain nearly unchanged until 1000 CV cycles (Fig. 1e). A similar trend was observed in the chronopotentiometry measurements at a constant current density of 10 mA cm$^{-2}$ in 1.0 M KOH, during which CoCr$_2$O$_4$ is activated after ~2 h,

and it can maintain the low potential for ~100 h, while Co$_2$CrO$_4$ maintains the potential for only ~60 h (Fig. S3b). Additionally, Tafel slopes of CoCr$_2$O$_4$ and Co$_2$CrO$_4$, measured from the LSV curves in Fig. 1a, b and summarized in Fig. S4a, b, show that the charge transfer kinetics of CoCr$_2$O$_4$ (67 ± 1 mV dec$^{-1}$) is faster than that of Co$_2$CrO$_4$ (81 ± 1 mV mV dec$^{-1}$) at the onset of OER. After the activation (100 CV cycles), the Tafel slope of CoCr$_2$O$_4$ decreases slightly (65 ± 1 mV dec$^{-1}$) followed by a slight increase even after 2000 cycles (72 ± 1 mV dec$^{-1}$), while the Tafel slope of Co$_2$CrO$_4$ increases continuously to 94 ± 1 mV dec$^{-1}$ after 1000 cycles. These results indicate that the charge transfer kinetics of CoCr$_2$O$_4$ are retained after prolonged OER, whereas they decay rapidly in Co$_2$CrO$_4$. Overall, both CoCr$_2$O$_4$ and Co$_2$CrO$_4$ are active in OER, but activated CoCr$_2$O$_4$ outperforms Co$_2$CrO$_4$ by remaining active for significantly longer durations.

Intriguingly, the activation of CoCr$_2$O$_4$ during the first 100 cycles is closely associated with the changes in redox peaks in the CV curves (Fig. 1c, inset, with more CV curves in Fig. S5a). In the first 10 cycles (red curve), one pair of redox peaks (termed A2/C2 in Fig. 1c, inset) is observed at ~1.28 V vs. RHE, corresponding to the Cr(III)⟷Cr(IV) transition[23]. As the OER proceeds (blue curve, 100 cycles), the intensity of the A2/C2 redox peak decreases, accompanied by the appearance of

**Table 1 | Co or Cr redox peak positions and current of the CoCr$_2$O$_4$ sample during different CV cycles**

| Cycles | | | | | | | | | | | | |
|---|---|---|---|---|---|---|---|---|---|---|---|---|
| | A1 | | | | A2 | | | | A3 | | | |
| | Peak position $^a$ / V vs. RHE | Current density / mA cm$^{-2}$ | $\Delta S1_{A-C}$ | $\Delta Q1_{A-C}$ /C m$^{-2}$ | Peak position / V vs. RHE | Current density /mA cm$^{-2-2}$ | $\Delta S2_{A-C}$ | $\Delta Q2_{A-C}$ /C m$^{-2}$ | Peak position / V vs. RHE | Current density /mA cm$^{-2}$ | $\Delta S3_{A-C}$ | $\Delta Q3_{A-C}$ /C m$^{-2}$ |
| 1st | 1.17 | 0.54 | - | - | 1.27 | 1.51 | 0.015 | 2.9 | 1.45 | 0.86 | - | - |
| 10th | 1.14 | 0.63 | 0.004 | 0.7 | 1.26 | 1.36 | 0.012 | 2.5 | 1.45 | 0.87 | −0.001 | −0.3 |
| 100th | 1.06 | 2.25 | 0.013 | 2.7 | 1.24 | 1.03 | 0.003 | 0.5 | 1.46 | 1.08 | −0.004 | −0.8 |
| 500th | 1.06 | 1.89 | 0.016 | 3.1 | 1.24 | 0.80 | −0.002 | −0.3 | 1.46 | 1.37 | −0.006 | −1.2 |
| 1000th | 1.07 | 1.86 | 0.015 | 3.0 | - | - | −0.001 | −0.2 | 1.46 | 1.36 | −0.009 | −1.9 |
| 2000th | 1.08 | 1.23 | 0.010 | 2.1 | - | - | - | - | 1.46 | 1.36 | −0.014 | −2.8 |

$^a$The peak position determined by the potential at maximum current. The cut-off voltage for A1 is 1.0 ~ 1.18 V, A2 is 1.18 ~ 1.35, and A3 is 1.35 ~ 1.55, respectively.

two pairs of intensified redox peaks at ~1.06 V vs. RHE (A1/C1) and ~1.45 V vs. RHE (A3 and C3). These peaks often indicate Co(II)↔Co(II, III) and Co(II, III)↔Co(III) transitions, respectively[23,32,45]. The changes to these redox peaks suggest that the redox reaction at the Cr sites dominates during the first ten cycles, after which the Co sites take over as the primary drivers of the redox process in CoCr$_2$O$_4$. Notably, the A1/C1 redox peaks retain their high intensity, even after the 1000$^{th}$ cycle (Fig. 1c, purple curve). To further investigate the redox couples, we summarized the redox positions, current densities and enclosed areas as a function of CV cycles in Table 1. In addition, we analysed the charge difference between the anodic and cathodic peaks, based on the enclosed areas, as shown in Fig. S5b. Compared to A3/C3, the areas of the A1 and C1 peaks increase steadily during the first 100 cycles, after which they remain constant until after 2000 cycles, when the A1 peak area begins to decrease (Table 1 and Fig. 1c). The Co(II)↔Co(II, III) transition is likely associated with the Co$^{II}$(OH)$_2$ ↔ Co$^{III}$OOH transformation, as observed in our previous work on pure Co towards OER[46]. However, this transformation alone cannot account for the continuous increase in the A1 peak area[46]. This observation suggests that other processes are responsible for the pronounced A1/C1 redox peaks, such as ion intercalation. In addition to ion intercalation, the initial transformation likely involves the incorporation of hydroxide ions, as the A1 peak area increases continuously during the first 100 cycles of CoCr$_2$O$_4$ (Fig. 1c and Fig. S3a). To verify the ion intercalation processes, we performed the CV measurements on CoCr$_2$O$_4$ at various scan rates (Fig. S5c, e, f). We observed that the A1/C1 redox features are highly dependent on the scan rates, with a linear relationship between current densities and scan rates, confirming that the redox intensities are also induced by ion interaction and incorporation[47,48]. Thus, we hypothesize that this ion incorporation and intercalation is strongly associated with the Co$^{II}$(OH)$_2$ ↔ Co$^{III}$OOH transformation, possibly the α-Co$^{II}$(OH)$_2$ → γ-Co$^{III}$OOH transformation within the first 100 cycles (more experimental evidence in the following section), since the formation of γ-Co$^{III}$OOH requires the intercalation of hydroxide ions and water molecules into the layered α-Co$^{II}$(OH)$_2$ compared to β-Co$^{II}$(OH)$_2$. In comparison to CoCr$_2$O$_4$, the CV curves of Co$_2$CrO$_4$ mainly exhibit two pairs of redox peaks (A2/C2 and A3/C3) (Fig. 1d, inset). The intensity of A2/C2 remains constant during the first 500 cycles, but decreases slightly after 1000 cycles (Fig. 1d, purple curve). By relating these changes to the OER current densities (Fig. 1b, e), it appears that the redox activity of Co$_2$CrO$_4$ is dominated by Cr sites during the first 500 cycles. Thereafter, the surface becomes deactivated, likely due to deteriorated activity of the Cr sites. Additionally, the redox features of Co$_2$CrO$_4$ at varying scan rates are significantly less pronounced than those of CoCr$_2$O$_4$ (Fig. S5c, d), inferring that strong interaction/de-intercalation occurred in CoCr$_2$O$_4$ upon OER cycling.

Previous work found that surface activation is induced by Cr leaching, wherein newly generated vacancies and defect sites promote the formation of Co oxyhydroxide, which enhances the OER activity[23].

To examine the Cr leaching, ICP-MS was used to measure the cumulative cation masses in the electrolyte of both CoCr$_2$O$_4$ and Co$_2$CrO$_4$, before and after 5, 10, 20, 100, 500 and 1000 CV cycles (see Fig. 1f). For both CoCr$_2$O$_4$ and Co$_2$CrO$_4$, substantial Cr dissolution was observed compared to that of Co (Fig. 1f). Specifically, Cr leaching is four times greater in CoCr$_2$O$_4$ than in Co$_2$CrO$_4$ (Fig. 1f). The Cr dissolution rates in both systems are the highest during the first 100 cycles, after which they drop slowly. Cr dissolves at the applied potential and pH by forming soluble CrO$_4^{2-}$ in the aqueous electrolyte according to the Cr Pourbaix diagram[49]. Its solubility in KOH/H$_2$O is similar to that of K$_2$CrO$_4$ (~640 g/L at 20 °C). Thus, the solubility of Cr is ~171.5 g/L, which is significantly higher than the dissolved amount measured by ICP/MS (~1.2 mg/L). This suggests that Cr dissolution after 100 cycles is likely impeded by the in situ-formed surface Co-, Cr-based (oxy)hydroxides that may act as a barrier to inhibit rapid Cr leaching. Additionally, continuous Cr leaching requires its diffusion from the interior of the nanoparticle after surfaces are depleted of Cr in the first 100 cycles[41], which might further decrease the Cr dissolution rate. For Co, it dissolves in both materials upon OER cycling; its dissolution is slightly more pronounced in Co$_2$CrO$_4$, possibly due to its higher Co content (Fig. 1f). The stability number (S-number), defined as the molar ratio of generated oxygen to active cation loss during water electrolysis[50], was calculated to evaluate electrocatalyst stability. The S-numbers of CoCr$_2$O$_4$ and Co$_2$CrO$_4$ are ~4.3×10$^4$ and ~6.2×10$^3$, respectively (see Supplementary Note 1), indicating that CoCr$_2$O$_4$ has a better stability than Co$_2$CrO$_4$ towards OER. This finding contradicts the conventional view that stable electrocatalyst activity originates from high material stability. In our study, CoCr$_2$O$_4$, despite substantial Cr dissolution, was more stable than Co$_2$CrO$_4$. One might speculate that the considerable cation leaching induces substantial surface changes, such as an increase in surface area, which makes more electrocatalytically active surfaces accessible to electrolytes[51]. Our ECSA measurements, shown in Fig. S6a, b, confirm a considerable increase in ECSAs for both CoCr$_2$O$_4$ (from 13.5 ± 0.1 cm$^2$ to 52.0 ± 0.1 cm$^2$) and Co$_2$CrO$_4$ (from 13.5 ± 0.1 cm$^2$ to 32.5 ± 0.1 cm$^2$, error from fitting) during the rapid Cr leaching of the first 100 cycles. However, no activation was observed for Co$_2$CrO$_4$ in the first 100 cycles. These results suggest that the increase in ECSA induced by Cr cation leaching is unlikely to be a decisive factor in the activation of CoCr$_2$O$_4$ during OER cycling.

Notably, the activation of CoCr$_2$O$_4$ in the first 100 cycles is accompanied by the appearance of intensified A1/C1 and A3/C3 redox couples. The A3/C3 redox, which corresponds to the Co(II, III)↔Co(III) transition, also occurs during OER cycling of Co$_2$CrO$_4$, but no activation occurs. These results highlight the importance of the A1/C1 redox couple at ~1.06 V vs. RHE for the activation of CoCr$_2$O$_4$ during OER cycling. This was also noted in a previous study on CoCr$_2$O$_4$[23], where CoCr$_2$O$_4$ can be activated upon cycling from 0.9 V to 1.66 V vs. RHE, with the A1/C1 redox couple observed at ~1.1 V vs. RHE. This contrasts with a different set of CV measurements from 0.9 V to 1.52 V vs. RHE[23],

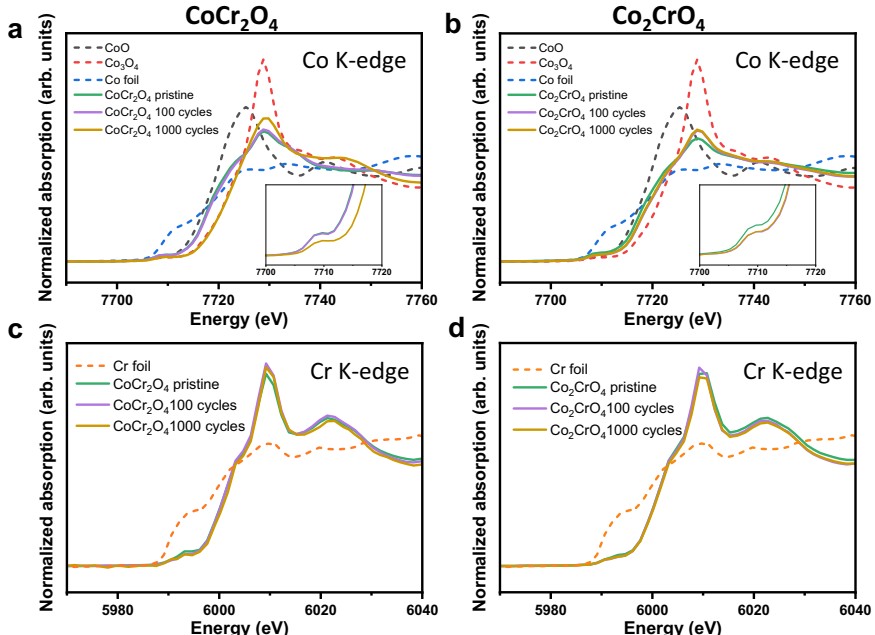

**Fig. 2 | Oxidation state of CoCr$_2$O$_4$ and Co$_2$CrO$_4$ during OER.** Normalized Co K-edge XANES spectra of **a** CoCr$_2$O$_4$ and **b** Co$_2$CrO$_4$ in the pristine state, after 100, and after 1000 CV cycles and the insets of the (**a** and **b**) are the pre-edge features of Co K-edge XANES; Cr K-edge XANES spectra of **c** CoCr$_2$O$_4$ and **d** Co$_2$CrO$_4$ in the pristine state, after 100, and after 1000 CV cycles.

during which no activation occurs for CoCr$_2$O$_4$ along with no pronounced A1/C1 redox peak. Those authors[23] speculated that the amount of Cr leaching at the potential before OER, i.e., 1.52 V vs. RHE, is insufficient to expose sufficient surface Co species for transformation into Co oxyhydroxides, as abundant Cr dissolution is thought to be critical for promoting surface reconstruction[23]. To further explore this hypothesis, we conducted additional CV measurements on CoCr$_2$O$_4$ from 1.20 V to 1.65 V vs. RHE (see Fig. S7a, b). Interestingly, CoCr$_2$O$_4$ OER activity drops continuously and rapidly (Fig. S7a), as indicated by the steadily increasing overpotentials (Fig. S7b) and decreasing OER current densities (Fig. S7a). Pronounced Cr dissolution, which is thought to expose sufficient surface Co sites for transformation into Co oxyhydroxides[23], is expected to occur at 1.65 V vs. RHE. However, CoCr$_2$O$_4$ cannot be activated upon cycling between 1.20 V and 1.65 V vs. RHE. These results suggest that CoCr$_2$O$_4$ activation requires not only sufficient Cr leaching, but also the occurrence of elementary processes at ~1.1 V vs. RHE (the A1/C1 redox couple) during which the Co(II)↔Co(II, III) transition and hydroxide ion (de)intercalation and incorporation occur. More importantly, the A1/C1 redox peak is also responsible for preserving the high OER activity of CoCr$_2$O$_4$, as evidenced by the more rapid drop in OER current density and overpotential when the A1/C1 transition is absent (Fig. S7a, b). Thus, we conclude that the occurrence of sufficient Cr leaching (at OER potential), Co$^{II}$(OH)$_2$ ↔ Co$^{III}$OOH transformation along with ion (de)intercalation and incorporation (at ~1.1 V vs. RHE), jointly improve the activity and stability of CoCr$_2$O$_4$ towards OER.

**Oxidation state and atomic coordination changes before and after OER cycling**

To further investigate the elementary processes during OER cycling and their effects on surface state evolution, we employed a multimodal characterization method to examine changes in the oxidation state, surface structure, composition and elemental distribution on the surfaces of CoCr$_2$O$_4$ and Co$_2$CrO$_4$ before and after OER. CoCr$_2$O$_4$ is known to have a normal spinel structure, where Co(II) occupies the tetrahedral sites and Cr(III) occupies the octahedral sites[52,53]. For Co$_2$CrO$_4$, Co(II) occupies the tetrahedral sites, and Co(III) and Cr(III)

the octahedral sites[54,55]. To assess the atomic coordination and oxidation state changes before and after OER, we performed XANES on CoCr$_2$O$_4$ and Co$_2$CrO$_4$ in the pristine state and after 100 and 1000 CV cycles under OER conditions (Fig. 2). The Co K-edge spectrum (Fig. 2a, purple curve) of 100-cycle CoCr$_2$O$_4$ nearly coincides with that of the pristine state (green curve), while after 1000 CV cycles (yellow curve), it shifts significantly toward higher energy values (by ~3 eV), indicating that the Co oxidation state increases after 1000 CV cycles. An approximate value of the oxidation state can be estimated by comparing the spectra with reference samples of similar metal coordination composition and known oxidation states (see Fig. 2a and Fig. S8a). The Co oxidation state of CoCr$_2$O$_4$ was estimated to change from ~2.0 in the pristine state and after 100 cycles to ~3.4 after 1000 cycles (Fig. S8a). This result agrees with our CV data (Fig. 1c), which reveals a broadening of the A1/C1 redox couple after 1000 cycles, suggesting that an irreversible oxidation process possibly occurs in CoCr$_2$O$_4$, leading to the presence of Co(III) species after OER cycling. Additionally, the intensity of the Co K pre-edge peak of CoCr$_2$O$_4$ at ~7710 eV decreases after 1000 cycles, along with an increase in the Co white line feature at ~7729 eV (Fig. 2a, yellow curve). This result suggests an increase in octahedrally coordinated Co after 1000 cycles, accompanied by a decrease in tetrahedrally coordinated Co, according to previous work[56]. The dissolution of Co(II) at the tetrahedral sites possibly results in the intensity decrease of the Co K pre-edge peak of CoCr$_2$O$_4$. The observed decrease in intensity of the tetrahedrally coordinated Co can also be attributed to the transformation of Co(II) at tetrahedral sites to the Co(III) oxyhydroxide after 1000 cycles, where Co is octahedrally coordinated[57].

The formation of oxyhydroxide on the surfaces of CoCr$_2$O$_4$ after OER is further confirmed by our XPS data from CoCr$_2$O$_4$ before and after 1000 CV cycles (Fig. 3a, c). Our peak fitting analysis of the Co 2$p_{3/2}$ spectra reveals that Co$^{III}$OOH is dominantly present on the CoCr$_2$O$_4$ surface after 1000 cycles (detailed peak fitting shown in Fig. 3a and Supplementary Note 2)[58]. The corresponding O 1$s$ spectra of CoCr$_2$O$_4$ (Fig. 3c) reveal two main components: (a) O1 at approximately 529.5 − 530.0 eV, corresponding to lattice oxygen in the metal oxide (Co−O or Cr−O); and (b) O2 at ~531.0 eV, indicating the presence of chemisorbed

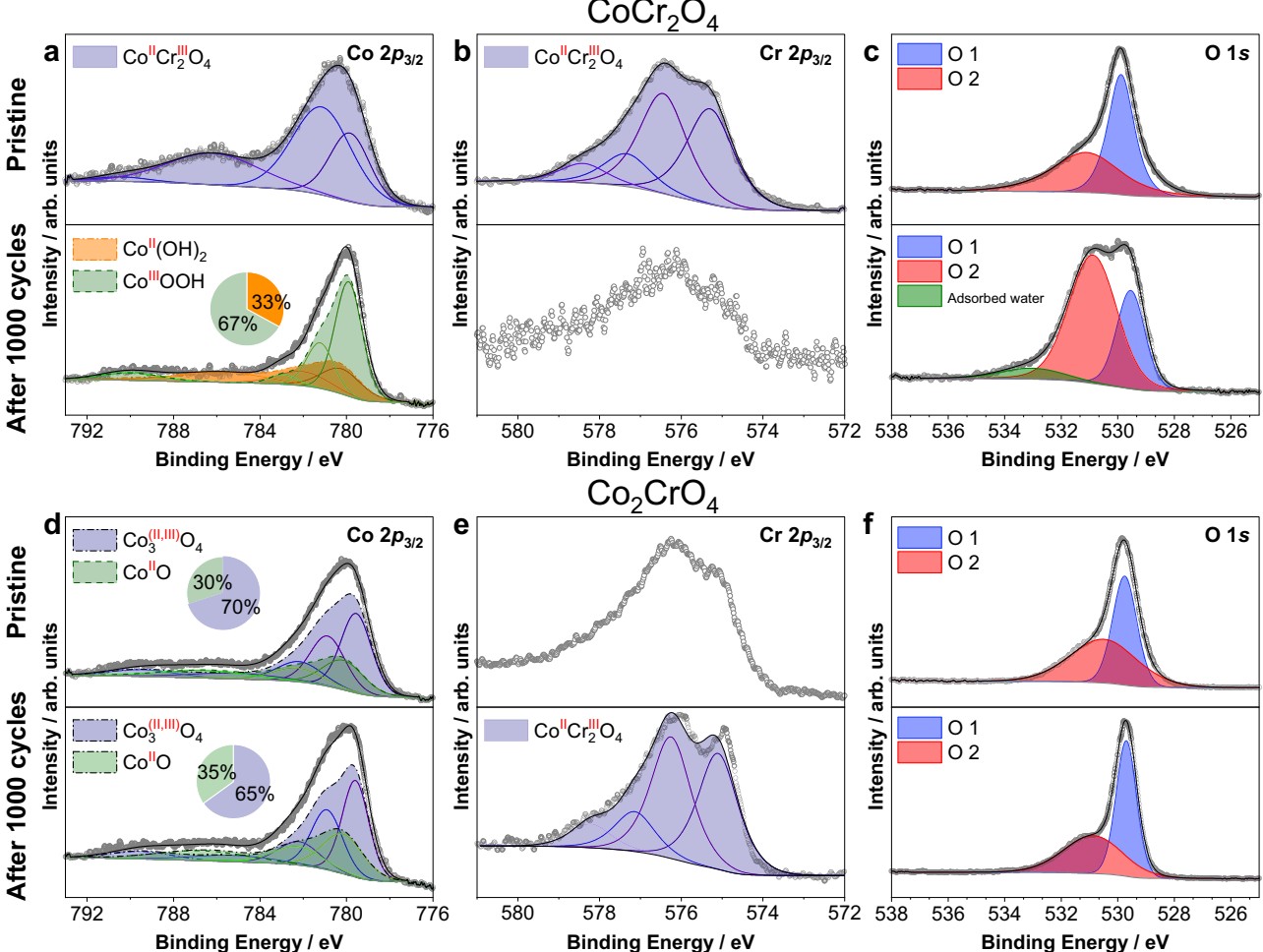

**Fig. 3 | Surface oxidation state and oxygen species of $CoCr_2O_4$ and $Co_2CrO_4$ before and after OER.** XPS of **a** Co 2p, **b** Cr 2p, and **c** O 1s levels of $CoCr_2O_4$; and **d** Co 2p, **e** Cr 2p, and **f** O 1s levels of $Co_2CrO_4$ in the pristine state and after 1000 cycles. A standard Shirley background was applied to all spectra. Models of Co(0), CoO, Co(OH)$_2$, CoOOH, Co$_3$O$_4$ were considered for the fitting of the Co 2p$_{3/2}$. The detailed information regarding the XPS measurements and peak fitting process are presented and discussed in Supplementary Note 2.

oxygen on the surface, such as $O_2^{2-}$ or $O^-$ associated with defective oxide and hydroxyl-like groups[59–62]. A considerable increase in the O2/O1 ratio of $CoCr_2O_4$ after 1000 cycles (Fig. 3c, bottom) indicates that more defective hydroxyl-like oxygen species were created through structural transformation, such as the formation of (oxy)hydroxides after 1000 cycles[59,60,63–66], consistent with our XANES data (Fig. 2a).

In situ generated Co(III) oxyhydroxide, as observed here, is generally thought to promote OER activity[31], indicating that it may already form during the activation of $CoCr_2O_4$ within the first 100 CV cycles. Interestingly, our electrochemical impedance spectroscopy (EIS) data, shown in Fig. S9a and Table S2, reveal that the electrical conductivity of $CoCr_2O_4$ increases after 100 cycles compared to its pristine state. This suggests that substantial transformation to defective (oxy)hydroxides likely occurs after 100 cycles, as this can enhance charge transfer kinetics[67]. Notably, purely cation vacancies may be insufficient to increase electrical conductivity, as the electrical conductivity of $Co_2CrO_4$ decreases after Cr leaching in the first 100 cycles (Fig. S9b). Thus, we speculate that although the transformation to defective Co(III) oxyhydroxide likely occurs in situ after 100 cycles, the transformation is most likely highly reversible, explaining why it was not detected by the ex situ XANES measurements.

In comparison to $CoCr_2O_4$, the Co oxidation state of $Co_2CrO_4$ increases from ~2.0 in the pristine state to ~2.4 after 100 cycles, and remains nearly constant after 100 and 1000 cycles, as indicated by the

near overlap of the Co K-edge spectra (Fig. 2b and Fig. S8b). This is consistent with our XPS data, which show negligible or no changes in the Co 2p$_{3/2}$ and O 1s spectra of $Co_2CrO_4$ after 1000 CV cycles (Fig. 3d, f). However, a slight decrease in the intensity of the Co K-pre-edge was observed after 100 cycles (Fig. 2b, inset), accompanied by an increase in the Co white line feature at ~7729 eV (Fig. 2b, purple and yellow curves). These results suggest a decrease in Co(II) at tetrahedral sites through Co leaching[19] and/or transformation to octahedrally coordinated Co(III) oxyhydroxides[57]. Given that the Co oxidation state of $Co_2CrO_4$ remains nearly unchanged after 100 and 1000 cycles (Fig. 3d), we speculate that the loss of tetrahedrally coordinated Co sites may arise from Co leaching. This is consistent with the ICP-MS data (Fig. 1f), which shows slightly more Co dissolution in $Co_2CrO_4$ during OER cycling.

Additionally, the Cr K-edge spectra of $CoCr_2O_4$ and $Co_2CrO_4$ remain nearly unchanged (Fig. 2c, d). XANES provides spectral information from the nanoparticle bulk, with a penetration depth of approximately ~3 µm[68], making it challenging to resolve minor changes on the nanometre-thick nanoparticle surfaces. In this regard, XPS is a more powerful technique for revealing surface details. Our XPS data (Fig. 3b, bottom) show that Cr located in the uppermost few nanometres of the $CoCr_2O_4$ surface possibly loses its atomic arrangement in the spinel structure due to substantial Cr dissolution and surface reconstruction after 1000 cycles. This is evidenced by the broadening

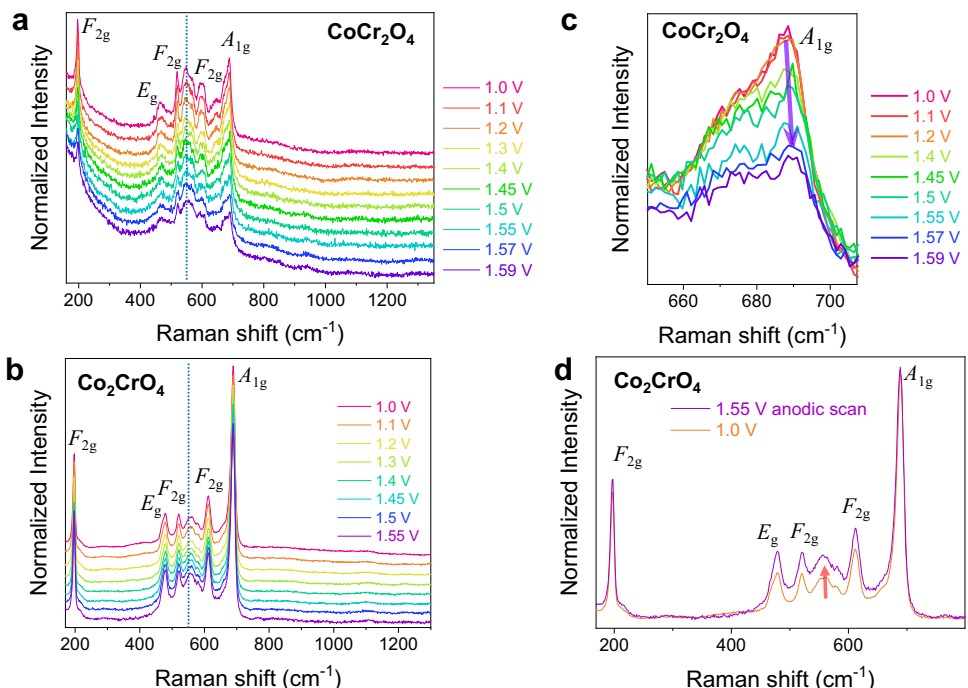

**Fig. 4 | Formation of intermediate species on $CoCr_2O_4$ and $Co_2CrO_4$ during OER.** In situ electrochemical Raman spectra of **a** 100-cycle $CoCr_2O_4$ measured stepwise in anodic scan 1 M KOH, **b** comparison overlaid spectra of $Co_2CrO_4$ on anodic and cathodic scans, **c** zoom in to $A_{1g}$ band of $CoCr_2O_4$, **d** $Co_2CrO_4$ measured stepwise in anodic scan.

of the Cr $2p_{3/2}$ spectra of $CoCr_2O_4$ after 1000 cycles, resulting in indistinct peaks that are unsuitable for peak fitting (Fig. 3b, bottom). For $Co_2CrO_4$, the Cr $2p_{3/2}$ spectra in the pristine state (Fig. 3e, top) could not be fitted to any reference samples due to the lack of a well-defined structure. The broadened spectra suggest a mixed chemical environment or state for Cr. After 1000 cycles, the Cr $2p_{3/2}$ spectra of $Co_2CrO_4$ become sharper and narrower, which fit well with those of pristine $CoCr_2O_4$, suggesting a similar Cr chemical and atomic coordination environment (Cr(III) are octahedrally coordinated with O in both $Co_2CrO_4$ and $CoCr_2O_4$). This might arise from the gradual exfoliation, erosion, or dissolution of the surface-reconstructed layer formed between 100 and 1000 cycles, exposing the well-defined bulk spinel oxide structure in the surface or near-surface regions after 1000 cycles (which will be discussed when describing TEM data in Fig. 5).

In brief, negligible changes to the Co oxidation state and coordination were detected by XANES and XPS for activated $CoCr_2O_4$ (after 100 cycles), while new species with a lower charge transfer resistance were formed, as revealed by EIS (Fig. S9a). After 1000 cycles, Co(II) at tetrahedral sites in $CoCr_2O_4$ is oxidized to form octahedrally coordinated Co(III) oxyhydroxides, which serve as active species for OER. In comparison, the Co oxidation state in $Co_2CrO_4$ remained almost unchanged ( + 2.4) after 100 and 1000 cycles of OER (Fig. 2c). As for Cr, its oxidation state in both $CoCr_2O_4$ remains the same before and after OER (Fig. 2d), while Cr dissolves substantially from the surfaces after 1000 cycles. Unlike substantial Cr dissolution in $CoCr_2O_4$, the Cr chemical environment in $Co_2CrO_4$ undergoes evident changes after extended OER cycling.

**In situ measurements of surface species by Raman spectroscopy**
XANES and XPS measurements were performed ex situ on the nanoparticles after OER cycling, which nevertheless show that the irreversible surface transformation and dissolution occur in $CoCr_2O_4$ and $Co_2CrO_4$ after OER cycling. To investigate the evolution of active surface species during OER cycling, in situ electrochemical Raman

spectroscopy was performed on activated $CoCr_2O_4$ and $Co_2CrO_4$ samples (after 100 cycles) between 1.0 and 1.59 V vs. RHE in 1 M purified KOH. Figure 4a, b shows that both samples contain five Raman modes of $A_{1g}$, $E_g$, and three $F_{2g}$, corresponding to structural features of the spinel phase[69]. The Raman peaks of $CoCr_2O_4$ appear less defined and possess asymmetric broadening on the $A_{1g}$ band compared to $Co_2CrO_4$ (Fig. 4a, b). The $A_{1g}$ band at ~687 cm⁻¹ belongs to the Raman mode of stretching vibration of M−O bond at octahedral sites of the spinel phase[69–71]. Given that Cr (III) occupies octahedral sites for both $CoCr_2O_4$ and $Co_2CrO_4$ samples, the intensity change in the $A_{1g}$ band can potentially provide insights into the Cr leaching during OER cycling. Interestingly, the intensity of the $A_{1g}$ band (at ~687 cm⁻¹) of 100-cycle $CoCr_2O_4$ decreases rapidly at above 1.40 V vs. RHE (Fig. 4c). Such a decrease in the intensity of this $A_{1g}$ band indicates bond distortion in the $MO_6$ sites, most likely due to Cr dissolution during the anodic sweep between 1.4 and 1.59 V vs. RHE. Notably, the $E_g$ band of CoOOH overlaps with $F_{2g}$ of Co spinel oxide phase at ~510 cm⁻¹, which makes it challenging to trace in situ formation of CoOOH on Co-based spinels by Raman spectroscopy. However, the $A_{1g}$ band of $CoCr_2O_4$ is slightly blue-shifted from 688 cm⁻¹ to 690 cm⁻¹ upon applied potential bias (Fig. 4c). This blue-shift in the $A_{1g}$ band indicates lattice contraction and charge redistribution due to the gradual formation of the Co-) based oxyhydroxide phase[72]. Thus, the gradual dissolution of Cr from the octahedral sites may lead to the formation of Cr and O vacancies, which promote the incorporation of hydroxide ions and the formation of the active oxyhydroxide phase[73].

In comparison, the $A_{1g}$ band of $Co_2CrO_4$ remains sharp at similar intensity, regardless of potentials (Fig. 4d), which indicates the bulk stability of octahedral-coordinated Co and Cr in 100-cycle $Co_2CrO_4$ during OER. In addition to the $A_{1g}$ band, a broad Raman peak centred at the wavenumber region of ~550 cm⁻¹, marked by blue-dashed line in Fig. 4a, b, is present on both samples from 1.0 to 1.55 V vs. RHE, which match with the main speak of $Cr(OH)_3$ reference material (see reference spectra at Fig. S10) from the bending vibration of Cr−O−H bond[74]. Upon the anodic sweep from 1.0 to 1.55 V vs. RHE on $Co_2CrO_4$,

a gradual increase of the broad peak at 430–630 cm$^{-1}$ was observed, which likely comes from the evolution of major peaks that belong to Cr(OH)$_3$ layer on the Co$_2$CrO$_4$ surfaces (Fig. 4d). Given that the redox activity of Co$_2$CrO$_4$ is thought to be dominated by Cr sites after 100 CV cycles (Fig. 1d), we speculate that Cr might transform to Cr(OH)$_3$, which contributes to the OER activity of Co$_2$CrO$_4$.

## Morphological, structural and compositional changes during activation and deactivation

To further examine the morphological, structural and compositional changes of CoCr$_2$O$_4$ and Co$_2$CrO$_4$, we performed TEM and APT in their pristine state and after 100 and 1000 CV cycles under OER conditions. The high-resolution TEM images (Fig. 5a, f) show that the lattice fringes in CoCr$_2$O$_4$ and Co$_2$CrO$_4$ nanoparticles correspond to a cubic spinel structure; d$_{111}$ in CoCr$_2$O$_4$ is slightly larger than that in Co$_2$CrO$_4$, as it has a slightly larger lattice constant, as indicated by our XRD data (Fig. S1a, b and selected area electron diffraction (SAED) pattern in Figs. S11a, S12a). To examine the atomistic structure on the nanoparticle surfaces, high resolution aberration-corrected high-angle annular dark-field scanning TEM (HAADF-STEM) was employed to analyse pristine CoCr$_2$O$_4$ and Co$_2$CrO$_4$ (Fig. 5b, g). We can see from Fig. 5b and the corresponding fast Fourier transform (FFT) images in Fig. 5c, viewed along the [110] zone axis, that the surface of pristine CoCr$_2$O$_4$ retains the spinel structure in its bulk. In addition to the surface atomistic structure, the composition and elemental distribution of pristine CoCr$_2$O$_4$ and Co$_2$CrO$_4$ nanoparticles were analysed by APT. The spinel oxides were detected in the form of O ions and Co- and Cr-containing complex molecular ions (see mass spectra in Fig. S13), represented as CoO$_x$ (in blue) and CrO$_y$ (in red), respectively, in Figs. 6 7. The cross-sectional atom map in Fig. 6a shows a uniform distribution of both Co and Cr in the pristine CoCr$_2$O$_4$ nanoparticle (additional APT data are provided in Figs. S14 and S15a). This is also confirmed by the 2D Cr compositional map shown in Fig. 6b. Notably, our TEM and EDX images in Fig. S1c reveal that most CoCr$_2$O$_4$ nanoparticles exhibit a triclinic shape, as confirmed in the atom map (Fig. 6a, b). To examine the surface compositions, 1D concentration profiles were plotted along the directions covering the vertices and flat surfaces of the triclinic-shaped nanoparticles, as indicated by the white and black arrows in Fig. 6a, b. The average Co/Cr and O/M (oxygen to (Co+Cr)) ratios measured from all the analysed nanoparticles are summarized in Tables 2 and 3. Figure 6c, d and Table 2 reveal that the Co/Cr atomic ratio is 0.56 ± 0.02 at the vertices, flat surfaces and bulk part of CoCr$_2$O$_4$, indicating a uniform distribution of Co and Cr across the CoCr$_2$O$_4$ nanoparticle.

After 100 cycles of OER, the activated CoCr$_2$O$_4$ nanoparticle surface becomes coarse, and the top 1–2 nm regions are amorphous, as shown in Fig. 5d (see more TEM images in Fig. S16a, b). The coarse surfaces of CoCr$_2$O$_4$ are likely induced by Cr dissolution, resulting in the formation of defective sites and material depletion from the nanoparticle surface. Previous work[23] reported the formation of amorphous (oxy)hydroxides on the surface of activated CoCr$_2$O$_4$. In our study, some surface regions (1–2 nm) become amorphous, and some do not become completely amorphous but exhibit a different atomic arrangement from the bulk part of the nanoparticles (Fig. 5d and Fig. S16a, b). Interestingly, additional reflection spots were observed in the SAED pattern of the 100-cycle CoCr$_2$O$_4$, as indicated by the red circles in Fig. S11b recorded from tens of nanoparticles, likely corresponding to the surface reconstructed layer. α- and β-Co(OH)$_2$[75], γ-CoOOH[76] and β-CoOOH[77] were fitted to the reflection spots, with α-Co(OH)$_2$, γ-CoOOH and β-CoOOH exhibiting different interplanar spacings (see Table S3). This suggests that CoCr$_2$O$_4$ nanoparticle surfaces might be partly transformed into α-Co(OH)$_2$, γ-CoOOH, and β-CoOOH. Additionally, APT data from the 100-cycle CoCr$_2$O$_4$ (Fig. 6e–h) reveals significant changes in composition and elemental distribution. Note that all the OER electrochemical measurements for APT specimen

preparation were conducted in a proton-free, deuterated electrolyte (1.0 M KOD in D$_2$O), to examine the elemental distribution of hydroxide ions (OD) after OER by APT[41,46,78]. Specifically, K and hydroxide ions are densely distributed in the upper 2–4 nm surface region of CoCr$_2$O$_4$ (Fig. 6f, with more data in Figs. S15b S17a), while some K and hydroxide ions are even present inside the CoCr$_2$O$_4$ nanoparticles (see detailed mass spectra in Fig. S18). By relating to the TEM/SAED results (Fig. S11b), the high amount of K and hydroxide ions is most likely associated with α-type Co-based hydroxide ↔ γ-type oxyhydroxide transition, which is in line with the intensified A1/C1 redox peak in Fig. 1c. The intercalated hydroxide and K ions balance the charges and stabilise the structure of γ-type oxyhydroxide, as observed during Ni α-hydroxide ↔ γ-oxyhydroxide transition[79]. Notably, the surface reconstructed region revealed by APT is 2–4 nm, which is thicker than that revealed by TEM (Fig. 4d). This could be caused by the high structural reversibility of the surface reconstructed layer. On the other hand, we cannot rule out the possibility that such a surface layer might partially undergo electron beam-induced recrystallization, especially at high magnification in the TEM.

The reconstructed 2–4 nm surface regions, where K and hydroxide ions are concentrated, are depleted of Cr and O, as indicated by an increased Co/Cr ratio of 0.69 ± 0.03 and a decreased O/M ratio of 1.1 ± 0.1 (lower than those in the bulk, see Fig. 6g and Table 2). Notably, the Co/Cr ratio in the core of 100-cycle CoCr$_2$O$_4$ is 0.59 ± 0.03 (Table 2), which is slightly higher than that of the pristine sample, suggesting that Cr leaches both at the surface and within the CoCr$_2$O$_4$ nanoparticles. Additionally, Cr and O are depleted to a lesser extent on flat surfaces compared to vertices, as indicated by the Co/Cr and O/M ratios revealed in Fig. 6g, h. Our observation of enhanced Cr dissolution at CoCr$_2$O$_4$ nanoparticle vertices is consistent with findings in other nanoparticle systems, where regions of high surface curvature exhibit increased dissolution rates[80,81]. The nanoparticles are thought to exhibit site- or facet-dependent dissolution behaviours due to the difference in surface energies, local potentials or adsorbed ligands[80,81]. Similarly, the reconstructed regions at the vertices contain higher amounts of K and hydroxide ions compared to those at flat surfaces (Fig. 6g and Fig. S15b). Notably, the K and hydroxide ions are mainly inside and underneath the nanoparticle surfaces. The K and hydroxide distribution is likely an indicator of the depth of the electrolytes that penetrate the nanoparticles and are accessible to the internal active sites. The pronounced increasing amount of hydroxide ions after 100 cycles (Table 1) is likely induced by ion incorporation and intercalation during OER cycling (as confirmed by Fig. S5c). Thus, our APT data demonstrate that the vertices of triclinic-shaped CoCr$_2$O$_4$ nanoparticles undergo more pronounced Cr dissolution, leading to the creation of more oxygen vacancies that promote the enhanced intercalation and incorporation of hydroxide ions in the reconstructed surface regions. This indicates that CoCr$_2$O$_4$ nanoparticles experience more surface transformation at their vertices than on their flat surfaces.

As OER proceeds after 1000 cycles, the surfaces of CoCr$_2$O$_4$ nanoparticles become increasingly rough (Fig. 5e). Additionally, ~2–3 nm nanoclusters form and attach to the CoCr$_2$O$_4$ nanoparticle surfaces (zoomed-in views in Fig. 5e and Fig. S16c, d). These nanoclusters exhibit a different crystal structure from spinel, likely α- or β-Co(OH)$_2$[75], γ-CoOOH[76] or β-CoOOH[77], as their structure matches well with three extra reflection spots in the SAED pattern of CoCr$_2$O$_4$, after 1000 CV cycles (Fig. S11c). Notably, β-CoOOH matches the best with all three additional reflection spots in the SAED pattern of CoCr$_2$O$_4$ after 1000 cycles, while the other three phases fit only one or two of the additional reflection spots, suggesting that β-CoOOH is most likely present in the 1000-cycle CoCr$_2$O$_4$ sample. This is also supported by our XPS finding that Co oxyhydroxides are formed after 1000 cycles (Fig. 3a, c). Additionally, the size of the CoCr$_2$O$_4$ nanoparticles decreases from 18.3 ± 5.5 nm to 14.5 ± 5.7 nm after 1000 cycles

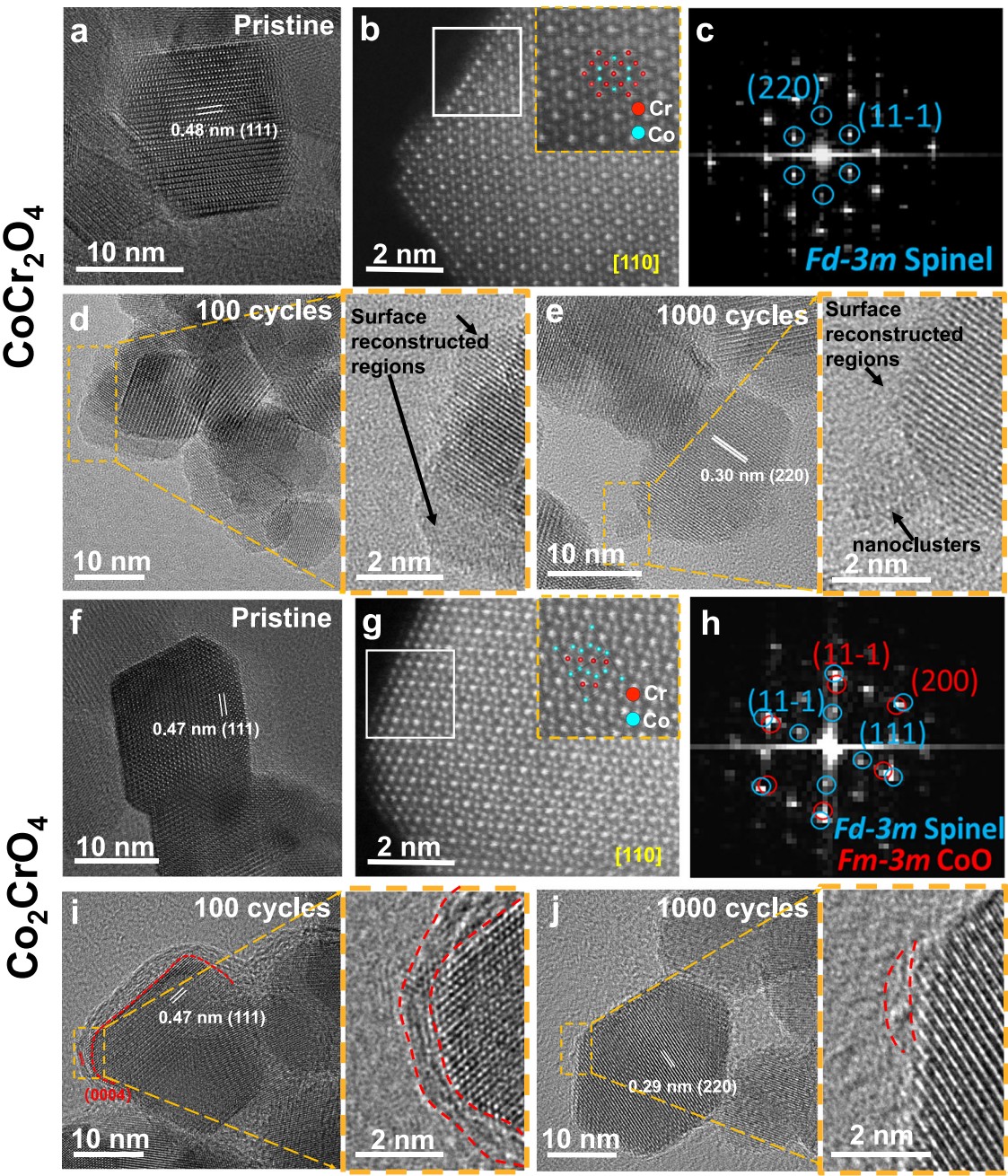

**Fig. 5 | Structural and morphological changes of the CoCr₂O₄ and Co₂CrO₄ nanoparticle after OER. a** High-resolution TEM and **b** aberration-corrected HAADF-STEM image of CoCr₂O₄ in the pristine state, observed in[110] orientation; **c** corresponding FFT image taken from the white square regions marked in (**b**); High-resolution TEM images of CoCr₂O₄ (**d**) after 100 cycles and (**e**) 1000 cycles with zoom-in surface regions show on the right that are taken from the yellow square region, respectively; **f** High-resolution TEM andaberration-corrected HAADF-STEM image of Co₂CrO₄ in pristine state viewed in[110] orientation; **h** corresponding FFT image taken from the white square regions marked in (**g**); High-resolution TEM images of Co₂CrO₄ (**i**) after 100 cycles and (**j**) 1000 cycles with zoom-in surface regions show on the right that are taken from the yellow square region, respectively. Simulated crystal structures are overlapped with the HAADF-STEM image in the insets of **b**, **g**, with Cr showing in red and Co in blue.

(see size histogram in Fig. S19). These results indicate that the reconstructed surface regions may undergo material depletion or structural collapse induced by continuous substantial Cr leaching. This is further evidenced by our APT data, which show that the thickness of the reconstructed surface regions decreases to 1–2 nm, as revealed by reduced amounts of K and hydroxide ions in the 2D Cr compositional maps and 1D concentration profiles (Fig. 6j–l, Fig. S15c, Fig. S17b and Table 2). These results suggest the surface reconstructed regions likely transform from hydrated α/γ (oxy)hydroxide (after 100 cycles) to β-CoOOH after 1000 cycles, which is typically less hydrated. The Co/Cr

ratio in the reconstructed surface regions increases to 0.98 ± 0.04 for both vertices and flat surfaces (Table 2 and Fig. 6k, l), suggesting that steady Cr dissolution occurs between 100 and 1000 cycles. Such continuous Cr leaching also occurs throughout almost the entire CoCr₂O₄ nanoparticle core, as the Co/Cr ratio decreases steadily to 0.79 ± 0.02 after 1000 cycles (Table 2). The OER activity of CoCr₂O₄ after 1000 cycles drops by ~34 mV compared to that after 100 cycles (Fig. 1a). The deterioration in activity of the 1000-cycle CoCr₂O₄ (compared to the most activated 100-cycle state) may be caused by the depletion of reconstructed surface regions and transformation to less

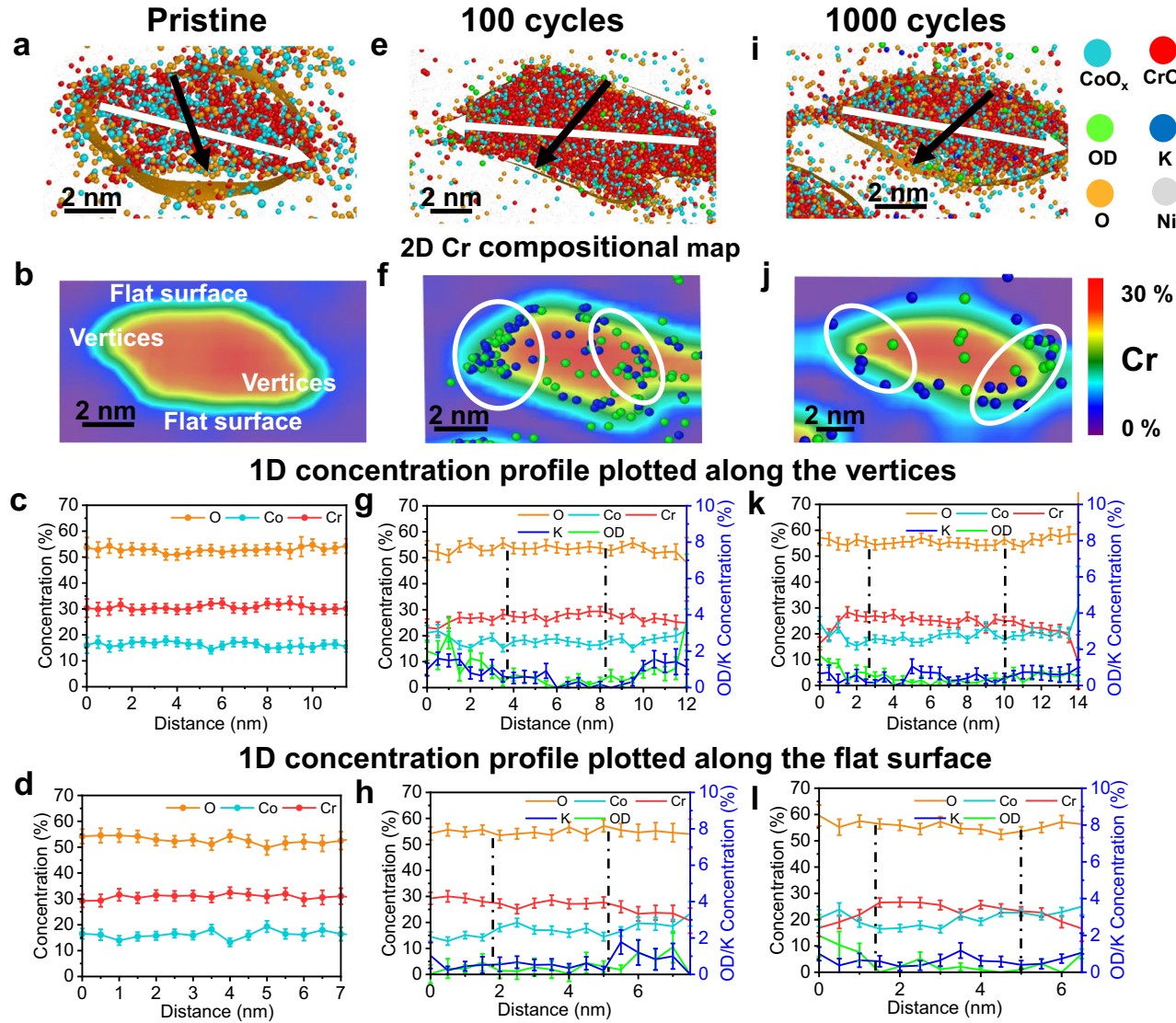

**Fig. 6 | 3D-APT reconstruction of CoCr$_2$O$_4$ nanoparticle showing the compositional evolution during OER. a, e,** and **i** 3D atom maps of CoCr$_2$O$_4$ nanoparticles in Ni matrix in the pristine state, after 100, and after 1000 CV cycles, respectively; 2D compositional maps of Cr in (**b, f** and **j**), the OD and K distribution at different states after OER are also shown with their rich region marked in the white circle; **c, g, k, d, h,** and **l** show the 1D concentration profiles of CoCr$_2$O$_4$, obtained by placing analysis cylinders with a diameter of 4 nm along the arrows with corresponding color marked in the 3D atom maps in (**a, e,** and **i**), in which white arrows from vertex to vertex and black from flat surface edge to edge, revealing the composition of metal composition changes. The error bars for the concentration were calculated from $\sqrt{\frac{(100-c)c}{N}}$, where $c$ is the concentration (in at.%) and N is the total number of atoms within the bin of the profile.

hydrated β-CoOOH. Although the reconstructed surface regions exfoliate gradually, they subsequently form small oxyhydroxide clusters that remain active toward OER, as the 1000-cycle CoCr$_2$O$_4$ still exhibits high OER current densities (Fig. 1e).

In comparison to CoCr$_2$O$_4$, the uppermost 4–5 atomic surface layers of pristine Co$_2$CrO$_4$ nanoparticles exhibit a different atomic arrangement compared to the core, as revealed by the aberration-corrected HAADF-STEM image in Fig. 5g. The FFT image, recorded from the surface region of the Co$_2$CrO$_4$ nanoparticle, shows additional spots (Fig. 5h, red circles) around the $(11\bar{1})$ and (200) reflections. To investigate the surface structure, the rock-salt structured CoO phase (Fm$\bar{3}$m[82]), corundum-structured Cr$_2$O$_3$ phase (R$\bar{3}$m[83]), and rutile-structured CoO$_2$ phase (P$\bar{3}$m1[84]) were fitted to the additional reflection spots. Only CoO matches the red colour reflection spots, indicating that the surface of Co$_2$CrO$_4$ spinel nanoparticles is covered by a rock-salt structure CoO phase. This secondary phase was not resolved by XRD (Fig. S1b), likely due to the thinness of the layer and its low

volume fraction. Additionally, our APT data reveal that some Co$_2$CrO$_4$ nanoparticles exhibit pronounced surface segregation of Cr, as more CrO$_x$ molecular ions (Fig. 7a, red dots) with a higher Cr concentration (Fig. 7b, red regions) are present on the surfaces (more APT data in Figs. S20 and S21a). This Cr surface enrichment is thought to be associated with a ~2 nm thick rock-salt structured CoO phase on the pristine Co$_2$CrO$_4$ nanoparticle surface. This explains why the Cr 2$p$ spectra cannot be fitted to any reference materials (Fig. 3e, top). The Co/Cr ratios in the Cr-rich region, i.e., the rock-salt structure CoO phase, are 1.2 ± 0.1 on the vertices and flat surface regions, which is lower than within the Co$_2$CrO$_4$ nanoparticle core (2.9 ± 0.1) (Fig. 7c, d). Thus, the uppermost ~2 nm rock-salt structure of Co$_2$CrO$_4$ spinel has a stoichiometry of Co$_{0.55\pm0.02}$Cr$_{0.45\pm0.02}$O; we term these as segregated Co$_2$CrO$_4$ nanoparticles. In contrast, the remaining Co$_2$CrO$_4$ nanoparticles exhibit uniformly distributed Co and Cr, with a Co/Cr ratio of 2.1 ± 0.1 (Fig. S22a–d, Table 2 and more APT data presented in Fig. S23a), which is closer to the nominal stoichiometric ratio of

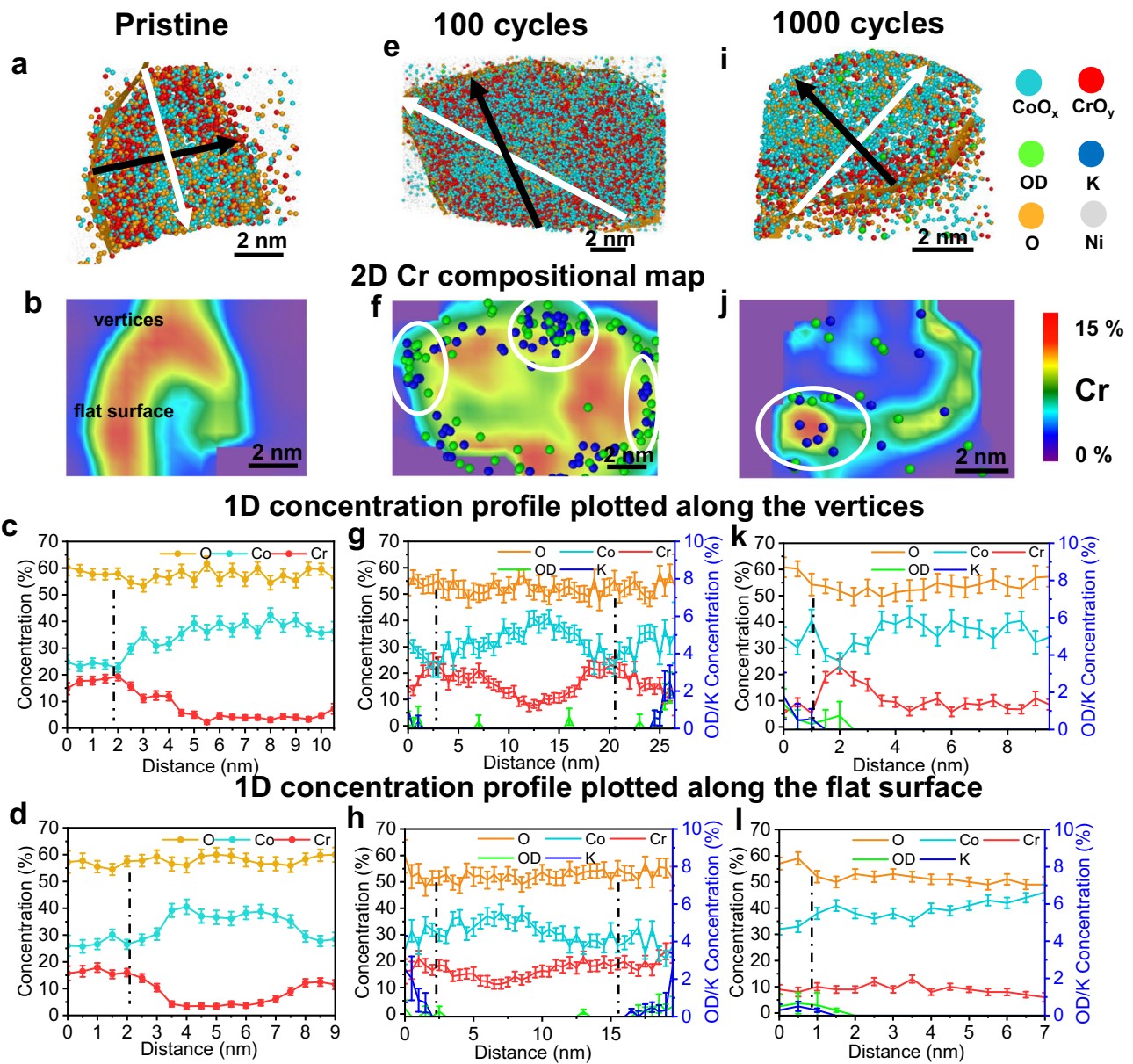

**Fig. 7 | 3D-APT reconstruction of segregated-Co₂CrO₄ nanoparticle showing the compositional evolution during OER. a, e**, and **i** 3D atom maps of segregated-Co₂CrO₄ nanoparticles in Ni matrix in the pristine state, after 100, and after 1000 CV cycles, respectively; 2D compositional maps of Cr in (**b, f** and **j**), the OD and K distribution at different states after OER are also shown with their rich region marked in the white circle; **c, g, k, d, h**, and **l** show the 1D concentration profiles of segregated-Co₂CrO₄, obtained by placing analysis cylinders with a diameter of 4 nm along the arrows with corresponding color marked in the 3D atom maps **a, e**, and **i**, in which white arrows from vertex to vertex and black from flat surface edge to edge, revealing the composition of metal composition changes. The error bars for the concentration were calculated from $\sqrt{\frac{(100-c)c}{N}}$, where $c$ is the concentration (in at.%) and N is the total number of atoms within the bin of the profile.

Co₂CrO₄. Essentially, Co(II) at the tetrahedral sites is thought to be responsible for the formation of active Co$^{III}$OOH species[85]. Co₂CrO₄ contains similar amounts of tetrahedrally coordinated Co(II) sites to that of CoCr₂O₄; however, no distinct Co(II)↔Co(II, III) redox couple was observed for Co₂CrO₄. This can be explained by the fact that tetrahedrally coordinated Co(II) is not exposed at the Co₂CrO₄ surface, as it is covered by the rock-salt structured (Co₀.₅₅Cr₀.₄₅)O phase, where Co(II) is octahedrally coordinated and thus cannot directly assist in the transformation to Co(III) oxyhydroxides. Therefore, Co₂CrO₄ cannot be activated upon OER cycling, as active Co-based oxyhydroxides cannot form.

After 100 cycles of OER, we observed that a 1–2 nm thick amorphous layer forms on the Co₂CrO₄ nanoparticles (as highlighted by the dashed lines in the TEM image in Fig. 5i and Fig. S24a, b). Additional reflection spots were observed in the SAED pattern of the 100-cycle Co₂CrO₄ samples (Fig. S12b), which has the best fit with Cr(OH)₃ since it matches all the additional reflection spots (Table S4). Given that our electrochemical data reveals that the Cr(III)↔Cr(IV) (A2/C2) and Co(II, III)↔Co(III) transitions (A3/C3), especially A2/C2, are responsible for the OER redox, we speculate that the amorphous layer is possibly composed of Cr-based (oxy)hydroxides that promote the OER activity of Co₂CrO₄. This is in line with the in situ Raman spectroscopy data (Fig. 4d), which shows that the intensity of Raman band corresponding to Cr(OH)₃ increases during the anodic sweep between 1.0 and 1.55 V vs. RHE. Additionally, our APT results in Fig. 7f reveal the presence of both hydroxide and K ions on the uppermost 1–2 nm surface of the Co₂CrO₄ nanoparticles. Notably, the hydroxide and K ions are present only on the surface of Co₂CrO₄ (Fig. 7f, Figs. S21b, S17c), unlike

**Table 2 | Average Co/Cr ratios in CoCr₂O₄ and Co₂CrO₄ nanoparticles were calculated by the total number of Co and Cr counts in all analyzed datasets, and average OD and K counts were obtained by calculating average OD/K counts from reconstruction data of all analyzed nanoparticles**

| Co/Cr | CoCr₂O₄ | | | | Co₂CrO₄ | | | | | | | |
| | | | | | Segregated | | | | Non-segregated | | | |
| | bulk | Reconstructed surface regions | OD counts per particle | K counts per particle | bulk | Cr-rich region | OD counts per particle | K counts per particle | bulk | Reconstructed surface regions | OD counts per particle | K counts per particle |
|---|---|---|---|---|---|---|---|---|---|---|---|---|
| Pristine | 0.56 ± 0.02 | - | - | - | 2.9 ± 0.1 | 1.2 ± 0.1 | - | - | 2.1 ± 0.1 | - | - | - |
| 100 cycles | 0.59 ± 0.03 | 0.69 ± 0.03 | 1114 ± 70 | 923 ± 54 | 2.6 ± 0.1 | 1.7 ± 0.1 | 644 ± 50 | 745 ± 36 | 2.1 ± 0.1 | 2.3 ± 0.1 | 506 ± 70 | 345 ± 42 |
| 1000 cycles | 0.79 ± 0.02 | 0.98 ± 0.04 | 598 ± 60 | 764 ± 32 | 2.7 ± 0.1 | 2.0 ± 0.1 | 362 ± 40 | 324 ± 30 | 2.1 ± 0.1 | 2.4 ± 0.1 | 337 ± 50 | 224 ± 45 |

The OD counts calculation is detailed in Supplementary Note 3.

$CoCr_2O_4$, where some K and hydroxide ions penetrate into the bulk. Additionally, the surface regions of the segregated $Co_2CrO_4$ nanoparticles, where hydroxide and K ions are located, show an increased Co/Cr ratio (~1.7, Fig. 7g). Beneath these surface regions, the Co/Cr ratio reaches ~1 (Fig. 7g), similar to the surface composition of pristine segregated $Co_2CrO_4$ nanoparticles. This result suggests that only the top half of the rock-salt structured $(Co_{0.55±0.02}Cr_{0.45±0.02})O$ layer transforms into amorphous Cr-based (oxy)hydroxides. Moreover, the Co/Cr ratio on the flat surfaces remains ~1.7 (Fig. 7h), lower than in the vertex region (Fig. 7g), suggesting that Cr dissolves more readily at the vertices than on the flat surface regions. After 100 cycles, the Co/Cr ratio in the $Co_2CrO_4$ nanoparticle core decreases compared to that in the bulk, indicating that Co also dissolves from the bulk. This finding is consistent with our XANES data of Co(II) loss and ICP-MS results. For the non-segregated $Co_2CrO_4$ nanoparticles, the top 1 nm is also covered by K and hydroxide ions and the Co/Cr ratio in the surface region increases to $2.3 ± 0.1$ (compared to $2.1 ± 0.1$ in the pristine state) (Fig. S22g and Table 2). This result demonstrates that the uppermost 1–2 nm surfaces of $Co_2CrO_4$ transform into amorphous Cr-based (oxy)hydroxides, accompanied by slight Cr leaching after 100 cycles.

After extended OER durations (1000 cycles), the ~1 nm amorphous surface layer on the $Co_2CrO_4$ nanoparticles almost completely disappears, resulting in rather rough surfaces, as revealed by TEM (Fig. 5j and Fig. S24c, d). Our APT data also confirm that the thickness of the Cr-rich surface layer (red-color regions) decreases to ~1 nm (Fig. 7j and Fig. S21c), which is thinner than in the pristine state (Fig. 7b) and after 100 cycles (Fig. 7f). Despite this, the $Co_2CrO_4$ nanoparticle size remains $18.4 ± 6.3$ nm (Fig. S25). Additionally, the uppermost 1 nm of the Cr-rich surface region shows an increased Co/Cr ratio ($2.0 ± 0.1$), suggesting continuous Cr dissolution during OER cycling. In contrast, although the Co/Cr ratio in the non-segregated $Co_2CrO_4$ nanoparticles remains the same as after 100 cycles, the amount of K and hydroxide ions decreases considerably (Figs. S17e, f, S22j–l and Fig. S23b, c), which indicates the disappearance of the active (oxy)hydroxide layers. The surface amorphous Cr-based hydroxides are most likely depleted, possibly due to the continuous dissolution of Cr and Co revealed by our ICP-MS data (Fig. 1f). Also, Cr-based hydroxides have a high solubility in alkaline conditions[86,87]. These results explain the significant drop in the OER activity of $Co_2CrO_4$ after 1000 cycles (Fig. 1d, e), and also why the XPS Cr 2$p$ spectra of $Co_2CrO_4$ become well-defined after 1000 cycles (compared to its pristine state, Fig. 3e) since the bulk spinel region is exposed on the surface due to dissolution of active Cr-based hydroxides.

### Activation mechanisms for CoCr2O4 and Co2CrO4, and roles of Cr and Co in the surface transformation

Overall, activated $CoCr_2O_4$ outperforms $Co_2CrO_4$, although both are active for OER (with an overpotential of $350 ± 2$ mV at 10 mA cm$^{-2}$, Fig. 1a, b). Previous studies[22,24] speculated that the addition of electron-deficient Cr increases the electrophilicity of Co(II) by increasing the Co-O covalency, which favours electron transfer between Co cations and oxygen adsorbates. Our study reveals that Cr at varying Co/Cr ratios induces the formation of different active species through various elementary processes during OER cycling, demonstrating that different spinel oxides facilitate distinct surface transformation mechanisms[88,89] (see schematic diagram in Fig. 8). For $CoCr_2O_4$, Cr dissolves steadily and substantially across nearly the entire nanoparticle (Fig. 8a), according to our ICP-MS and APT data (Fig. 1f, Table 2). This dissolution not only leads to an increased ECSA (Fig. S6a) but also creates vacancy sites, especially oxygen vacancies that facilitate quasi-bulk intercalation and incorporation of hydroxide ions into the $CoCr_2O_4$ nanoparticles during the first 100 cycles (Fig. 8a and Fig. S5c), as revealed by our APT data (Fig. 6f). The intercalation of hydroxide ions, especially within electrocatalyst materials after reactions, is difficult to identify using spectroscopic techniques, as

**Table 3 | Average O/M ratios in $CoCr_2O_4$ and $Co_2CrO_4$ nanoparticles calculated by the total number of O and M (Co + Cr) counts in all analyzed datasets**

| O/(Cr+Co) | $CoCr_2O_4$ | | | | $Co_2CrO_4$ | |
| | | | Segregated | | Non-segregated | |
| | bulk | Reconstructed surface regions | bulk | Cr-rich region | bulk | Reconstructed surface regions |
|---|---|---|---|---|---|---|
| Pristine | $1.2 \pm 0.1$ | - | $1.1 \pm 0.1$ | $1.2 \pm 0.1$ | $1.0 \pm 0.1$ | - |
| 100 cycles | $1.2 \pm 0.1$ | $1.1 \pm 0.1$ | $1.1 \pm 0.1$ | $1.1 \pm 0.1$ | $1.0 \pm 0.1$ | $1.1 \pm 0.1$ |
| 1000 cycles | $1.2 \pm 0.1$ | $1.2 \pm 0.1$ | $1.1 \pm 0.1$ | $1.2 \pm 0.1$ | $1.0 \pm 0.1$ | $1.3 \pm 0.1$ |

hydroxide ions have similar chemical and physical properties to hydroxyl groups in these materials. Our APT data provide unambiguous atomic-scale evidence that hydroxide and K ions are distributed on the activated $CoCr_2O_4$ nanoparticle surfaces, mainly at the vertices, as well as in the core (Fig. 6f), as illustrated in Fig. 8a. The continuous intercalation of hydroxide ions likely promotes the $\alpha$-$(Co^{II}, Cr^{III})$ $(OH)_2 \leftrightarrow \gamma$-$(Co^{III}, Cr^{III})OOH$[90,91] transformation, as evidenced by increasing amounts of hydroxide and K ions in $CoCr_2O_4$ nanoparticles (Table 2 and TEM/SAED pattern in Fig. S11b), Fig. 8b. The incorporation and intercalation of $K^+$ and $OH^-$ balances the charge and stabilises the structure of $\gamma$-type oxyhydroxide, as observed in the early Bode study[79]. Although the increased ECSA induced by Cr leaching contributes to the activation of $CoCr_2O_4$, it is not a decisive factor (as discussed in Section 2.1). Instead, the in situ transformation to highly hydrated $(Co^{III}, Cr^{III})OOH$ is thought to be responsible for the activation of $CoCr_2O_4$ upon OER cycling, where Cr plays an essential role in improving the OER activity of $(Co^{III}, Cr^{III})OOH$ grown on $CoCr_2O_4$. $Cr(III)$ in $(Co^{III}, Cr^{III})OOH$ can accelerate the charge transfer process due to the decreased charge transfer resistivity of the 100-cycle $CoCr_2O_4$ (EIS data in Fig. S9a, blue curve), thus increasing the OER current densities (Fig. 1c). This is consistent with the previous work on Co-Cr oxyhydroxide[92] or Co-Cr LDH[93], where their charge resistivity is lower than that of pure Co counterparts. Also, Cr lowers the oxidation potential of the Co(II)/Co(III) transition, as it occurs at ~1.06 V vs. RHE (Fig. 1c, inset), which is lower than that of $Co_3O_4$ (~1.1 V vs. RHE)[64], improving the activity.

More importantly, activated $CoCr_2O_4$ (after 100 cycles) can maintain its high OER activity for extended durations (~1000 CV cycles), albeit with substantial Cr dissolution. This is associated with a persistent and pronounced A1/C1 redox peak at ~1.06 V vs. RHE. This pronounced A1/C1 peak, like the redox peak in a battery or supercapacitor[94], arises from intercalation and incorporation of not only hydroxide and K ions, but is also due to the $(Co^{II}, Cr^{III})$ $(OH)_2 \leftrightarrow (Co^{III}, Cr^{III})OOH$ transition (Fig. 8b), which is driven dominantly by transformation from Co(II) at the tetrahedral sites to Co(III) oxyhydroxide in $CoCr_2O_4$, as indicated by our XANES and XPS results (Figs. 2a and 3a,c). Co(II) or Co(III) at the octahedral sites in Co-based spinels are thought to be the active sites for OER[95–98] since the 3 d orbitals overlap more with an oxygen 2p orbital, where electron transfer is favoured, compared to that of tetrahedrally coordinated Co(II). Our study demonstrates that the octahedrally coordinated Co(II) sites on the rock-salt structured surfaces of $Co_2CrO_4$ (revealed by HAADF-STEM image in Fig. 5g, h) do not yield the transformation to Co(III) oxyhydroxides (according to XANES and XPS results in Figs. 2b, 3d, f), whereby no pronounced A1/C1 peaks or activation are observed during OER cycling (Fig. 1d). This is consistent with a previous operando XANES study where tetrahedrally coordinated Co(II) was reported to be responsible for the formation of active $Co^{III}OOH$ species[85]. Intriguingly, these results suggest that the persistent and pronounced A1/C1 peak at 1.0-1.1 V vs. RHE exclusively indicates the transformation of tetrahedrally coordinated Co(II) to Co(III) oxyhydroxides along with pronounced ion intercalation and incorporation. Previous work on different pure Co oxides and oxyhydroxides[32] proposed that the redox

charge of Co(II)/Co(III) is a rational descriptor of OER active sites. Our study demonstrates that the persistent and prominent A1/C1 peak can be used as an indicator of both high OER activity and stability.

Adding Mn or Fe can also lower the Co(II)/Co(II, III) transition to $Co_3O_4$ or activated $Co_3O_4$[19,31,42]. However, the A1/C1 peak in most previous studies is comparatively subtle and the surfaces transform to $\beta$-type oxyhydroxide on CoMn- and CoFe-based spinel oxide[21,32,99–101]. The absence of prominent A1/C1 redox peaks in other studies[21,32,99–101] is possibly due to the limited incorporation and intercalation of hydroxide ions, which results from the absence of defect sites, e.g., vacancies that facilitate ion intercalation. In this regard, steady Cr dissolution from the surfaces and core of $CoCr_2O_4$ nanoparticles is critical for maintaining steady ion intercalation processes. On the other hand, Fe and Mn in $Co_2FeO_4$[31] or $Co_2MnO_4$[19,42] also dissolve during OER cycling but enhanced ion intercalation and incorporation were not observed for Fe- and Mn-doped $Co_3O_4$. This infers that, in addition to substantial dissolution, Cr may also promote ion intercalation and incorporation, especially hydroxide ions as they adsorb more easily on Cr than on Fe, Mn, Co, or Ni[102]. Additionally, Cr is speculated to be effective in stabilizing hydroxide ions on the adjacent Co sites[103]. When the ion intercalation process was intentionally avoided (Fig. S7), the OER activity drops rapidly (Fig. 1b and Fig. S7). These results lead us to draw the counterintuitive conclusion that steady Cr dissolution has a positive effect on OER activity and stability. Cr leaching (at OER potential) and its high ability for hydroxylation and stabilization of hydroxide ions (at ~1.0 V vs. RHE) enhance the ion incorporation and intercalation, promoting the highly reversible intercalation-assisted $(Co^{II}_{Td}, Cr^{III})$ $(OH)_2 \leftrightarrow (Co^{III}_{Oct}, Cr^{III})OOH$ transformation, which can retain the activity for extended OER cycling. This is reasonable, as intercalation is a reversible process that is responsible for high battery lifecycles and supercapacitor applications such as Co or Ni layered double hydroxides (LDHs)[65].

In comparison, phase separation occurs in pristine $Co_2CrO_4$ nanoparticles, where rock-salt structures (Cr, Co)O cover the surface (Fig. 8d), as revealed by our TEM and APT data (Figs. 5g, h and 6b). Such octahedrally coordinated Co(II) cannot assist the transformation of Co(III) oxyhydroxides, and Cr(III) sites therefore dominate the transition (Fig. 1d) by forming a 2–3 nm amorphous Cr-based hydroxide layer on the $Co_2CrO_4$ surfaces (Fig. 8e). An in situ transformation of Cr-based hydroxide to oxyhydroxide most likely occurs during OER, contributing to the activity of $Co_2CrO_4$, since Cr-based (oxy)hydroxides are also active for OER[104]. Despite this, the OER activity of Cr (oxy)hydroxides is lower than that of Co (oxyhydroxide)[105], explaining why the activity of $Co_2CrO_4$ is worse than that of $CoCr_2O_4$. Additionally, fewer intercalated ions are present for $Co_2CrO_4$, suggesting a lower extent of ion intercalation (as shown by the APT results in Table 2 and Fig. 7f).

Furthermore, the intercalated hydroxide and K ions might also contribute to the enhanced OER activity of $CoCr_2O_4$. For instance, the presence of intercalated $K^+$ in the interlayer of surface oxyhydroxide may improve OER activity, as it can promote and stabilize the formation of active intermediates, as reported for Co oxyhydroxide[106]. Also, we speculate that the enhanced intercalation and incorporation of

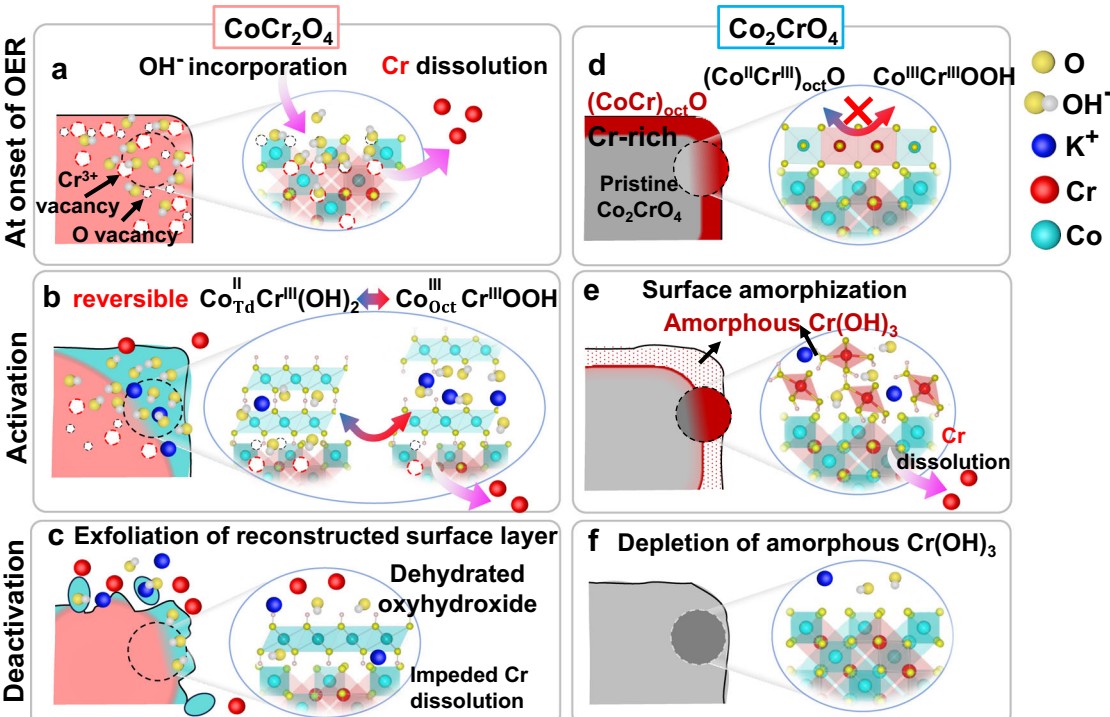

**Fig. 8 | Schematic diagram showing surface reconstruction and transformation of $CoCr_2O_4$ and $Co_2CrO_4$ during OER. a** Cr dissolves substantially across nearly the entire $CoCr_2O_4$ nanoparticles, leading the formation of Cr and O vacancies that promote the hydroxide ion incorporation; **b** the gradual hydroxide ion incorporation potentially facilitates reversible $(Co^{II}, Cr^{III})(OH)_2 \leftrightarrow (Co^{III}, Cr^{III})OOH$ transformation, giving rise to high activity and stability of $CoCr_2O_4$; as OER proceeds, **c** the (oxy)hydroxides are gradually exfoliated from $CoCr_2O_4$ surfaces. In contrast, **d** pristine $Co_2CrO_4$ surface is covered by rock-salt structured (Co,Cr)O which does not assist transformation of Co-based oxyhydroxide. Instead, **e** amorphous $Cr(OH)_3$ layer is formed on the $Co_2CrO_4$ surfaces, serving as active species for OER. Such self-limiting amorphous layer is gradually depleted from $Co_2CrO_4$ surfaces, decreasing the activity.

hydroxide ions also contribute to the increased OER activity of activated $CoCr_2O_4$ via the confinement effect[107,108], since the intercalated hydroxide ions have access to more active sites inside $CoCr_2O_4$ nanoparticles. A recent experimental coupled density functional theory study proposed that intercalated hydroxide ions can potentially deprotonate the hydroxyl groups of α-CoFe-LDHs, along with the oxidation of transition metals under the OER potential[36,109]. In other words, the intercalated hydroxide ions in the interlayers can serve either as proton acceptors or proton transfer reagents. The kinetics of such a process is generally regarded as highly dependent on the electrolyte pH value[110,111]. To investigate the pH-dependent OER activity characteristics, we performed CV measurements on 100-cycle $CoCr_2O_4$ and $Co_2CrO_4$ from 0.03 M KOH (pH = 12.5 ± 0.1) to 1.0 M KOH (pH = 14 ± 0.1) (Fig. S26). Interestingly, 100-cycle $CoCr_2O_4$ shows a pH-dependent OER activity, while 100-cycle $Co_2CrO_4$ displays a weaker pH-dependent OER activity (Fig. S26a, b). The pH dependence of OER activity indicates the presence of non-concerted proton-electron transfer steps during OER, where the rate-limiting step is a proton transfer step or is preceded by the acid-base equilibrium[110,111]. This result suggests that non-concerted proton-electron transfer steps likely occur for 100-cycle $CoCr_2O_4$ during OER, when the intercalated hydroxide ions promote the deprotonation of hydroxyl groups in (Co, Cr) oxyhydroxides by forming water molecules, thus enhancing OER activity. Additionally, one might speculate that the improved OER activity of $CoCr_2O_4$ is associated with the lattice oxygen mechanism[112]. However, this is challenging to assess as $CoCr_2O_4$ requires activation processes, wherein the surface lattice oxygen anions dissolve due to the formation of soluble $CrO_4^{2-}$ during the first 100 CV cycles. Essentially, the enhanced OER activity of activated $CoCr_2O_4$ is dominated by in situ surface reconstructed layer formed via (de)intercalation processes. Thus, we refrain from making conclusive statements, although efforts have been devoted to performing isotope-labelled operando

differential electrochemical mass spectrometry (DEMS) measurements on activated $CoCr_2O_4$ and $Co_2CrO_4$ (see discussion in Fig. S27–30 and Supplementary Note 4).

As OER proceeds for long durations, repeated ion intercalation, along with the $(Co^{II}, Cr)(OH)_2 \leftrightarrow (Co^{III}, Cr)OOH$ transformation and continuous Cr leaching in (oxy)hydroxides, possibly leads to the collapse and exfoliation of the (oxy)hydroxides, which gradually become depleted from the $CoCr_2O_4$ surfaces (Fig. 8c). This process is more pronounced at the vertices of $CoCr_2O_4$ nanoparticles, where more oxygen vacancies are created (as indicated by our APT data, Fig. 5g and Table 3). This results in a decrease in $CoCr_2O_4$ nanoparticle size (Fig. S19). The depleted active material takes the form of small nanoclusters (Fig. 4e), which can still contribute to the OER activity of $CoCr_2O_4$ (Fig. 8c). In contrast to $CoCr_2O_4$, the active amorphous Cr-based (oxy)hydroxides formed on $Co_2CrO_4$ are self-limiting. Upon OER cycling, this amorphous layer gradually disappears due to its high solubility[86,87], leading to a decline in $Co_2CrO_4$ OER activity (Fig. 8f). Thus, $Co_2CrO_4$ cannot retain the activity for longer OER durations due to gradual dissolution of Cr-based (oxy)hydroxide layer.

In summary, our multimodal approach provides a systematic correlation between the electrocatalyst surface states and the activity and stability of $CoCr_2O_4$ and $Co_2CrO_4$ spinels toward OER. The most important finding is that $CoCr_2O_4$ undergoes an activation process and subsequently exhibits enhanced stability. This is due to considerable Cr leaching, which promotes the intercalation and incorporation of hydroxide ions, yielding a highly reversible $(Co^{II}_{Td}, Cr^{III})(OH)_2 \leftrightarrow (Co^{III}_{Oct}, Cr^{III})OOH$ transformation, which is key to the high OER activity and stability of $CoCr_2O_4$. The persistent and prominent A1/C1 redox couple at -1.0–1.1 V vs. RHE in the CV curve is considered one descriptor of the high activity and stability of Co-based spinels toward OER. In comparison, an amorphous self-limiting Cr-based (oxy)hydroxide, with a thickness of 1–2 nm, forms on $Co_2CrO_4$ upon

OER cycling, contributing to its activity. As OER proceeds, the activity of $Co_2CrO_4$ drops rapidly when the (oxy)hydroxides disappear due to steady Cr leaching. Our study demonstrates that high OER activity and stability of Co-based spinels can be achieved by inducing the highly reversible $(Co_{Td}^{II})(OH)_2 \leftrightarrow (Co_{Oct}^{III})OOH$ transformation, assisted by the steady intercalation of hydroxide ions via approaches such as oxygen vacancy regulation[64]. These mechanistic insights will be potentially extended to develop next-generation OER electrocatalysts, such as Co-containing high-entropy spinels, to improve the efficiency of water electrolysers for hydrogen production.

## Methods

### Chemicals
$Co(NO_3)_2 \cdot 6H_2O$ (98%), $Cr(NO_3)_3 \cdot 9H_2O$ (99%), KOH (99.9%), Nafion® perfluorinated resin solution (D520, 5 wt.%), KOD (original content 40 wt.% in $D_2O$), $D_2O$ electrolyte (99.9%), $NiCl_2 \cdot 6H_2O$ (98%), $NiSO_4 \cdot 6H_2O$ (98%), and Boric acid (99.5%) are all from Sigma-Aldrich, Germany. The glassy carbon electrode, Hg/HgO (filled with 0.1 M KOH) reference electrode, and graphite carbon electrode were purchased from ALS, Tokyo, Japan.

### Synthesis of Co-Cr spinel-type oxide nanoparticles
In the synthesis of cobalt chromite nanoparticles, a coprecipitation approach was employed. Initially, a desired amount of $Co(NO_3)_2 \cdot 6H_2O$ and $Cr(NO_3)_2 \cdot 6H_2O$ (3 mmol in total, 1:2 in $CoCr_2O_4$ and 2:1 $Co_2CrO_4$ sample, respectively) was combined in 40 mL deionized water, within a 50 mL beaker. Subsequently, a 10 mL 3 M KOH solution was dropwise introduced into the reaction mixture with constant stirring until complete precipitation occurred. Then, the resultant precipitate was isolated via centrifugation, subjected to multiple washes with water and ethanol, and subsequently dried at 60 °C for twelve hours. The dried powder was annealed in air at 500 °C for three hours to obtain the final $CoCr_2O_4$ and $Co_2CrO_4$ nanoparticle powder.

### Electrochemical measurements
Electrochemical evaluations were conducted to assess the OER performance of synthesized catalysts, utilizing a three-electrode setup within a single chamber cell (SC), which is a plastic cell with a volume of 100 mL and a Teflon cap. The electrolyte (1 M KOH, pH = 14.00 ± 0.01 measured using pH meter) was freshly prepared and promptly utilized, which was prepared in the 1000 mL volumetric flask. The Chelex cation-exchange resin (Sigma Aldrich) was used to remove the metal impurities from the KOH solution[44]. A Bio-Logic SP-300 potentiostat was used for the measurements, with Hg/HgO serving as the reference and graphite carbon as the counter electrode. The reference electrode was calibrated against another unused Hg/HgO electrode in the same electrolyte by measuring the open-circuit potential (OCP). The working electrode was fabricated by depositing a suspension of the catalyst onto a 5 mm diameter glassy carbon electrode (0.196 cm²), and cleaned with alumina slurry and washed with ethanol and water using ultrasonography. This suspension was prepared by sonicating a mixture of the catalyst powder (4 mg), carbon black (0.8 mg), Nafion (4 mg, 5% wt. in water), and deionized water (1 mL). A 12.5 µL aliquot of this suspension was applied to the electrode surface and allowed to dry at ambient conditions, giving a mass loading of -0.25 mg cm⁻². Before OER measurements, the electrolyte was thoroughly degassed by purging it with oxygen ($O_2$) gas at a flow rate of 200 mL/min for 30 min, and the electrolyte volume was -80 mL for SC. The potential was calculated versus RHE scale, using the following equation:

$$E(RHE) = E(Hg/HgO) + 0.098V + 0.0592V \times pH$$

where E is denoted as the applied potential on the working electrode. LSV and CV were performed to determine the current density and to compensate for ohmic resistance, which was corrected using resistance values (obtained from EIS conducted at OCP) measured automatically by the potentiostat (90% $iR$ drop). The applied potential with $iR$ compensation is denoted as $E − iR$, and it is denoted directly as E without compensation. LSV curves were measured from 1 to 1.7 V (vs. RHE) at a slow scan rate of 10 mV s⁻¹ while the CV was recorded at a scan rate of 50 mV s⁻¹ with a potential window of 1 to 1.65 V (vs. RHE) at a constant rotation speed of 1600 rpm. The Tafel slope was determined using the LSV plots that covers the whole measuring range.

The dissolution of Cr and Co during the CV cycles was determined by measuring the probes of electrolyte taken after 5, 10, 20, 100, 500, and 1000 CV cycles (without refilling the electrolyte) using a PerkinElmer NexION 300x ICP-MS. ICP-MS was calibrated before the measurements by a four-point calibration slope prepared from standard solutions that contained ⁵⁹Co and ⁵²Cr. ⁷⁴Ge and ⁸⁹Y were used as internal standards, respectively. The internal standard solution was prepared in 2% $HNO_3$ electrolyte and introduced to the nebulizer of the ICP-MS via a Y-connector. Due to the presence of polyatomic interferences, the ICP-MS was operated in kinetic energy discrimination mode, utilizing He gas to minimize the impact of these interferences. To analyze the quantities of the dissolved elements, the probes in 1 M KOH electrolyte were diluted 100 times for analysis with ICP-MS. The dilution factors, as well as the original volumes of the provided analytes, were considered during the evaluation of the results.

The ECSA data were obtained based on $C_{dl}$ through CV. The CV curves were obtained from 0.80 to 0.90 V vs. RHE, where there was no Faradaic current, at scan rates of 10, 15, 25, 50, 75, and 100 mV dec⁻¹. The double-layer capacitance can be calculated as a formula, $C_{dl} = i/r$, where $i$ is the current density and r is the scan rate. The ECSA can be calculated as: $ECSA = S \cdot C_{dl}/C_s$, where $C_s$ is the specific capacitance value of an atomically smooth planar surface of the material per unit area (1 cm²) under identical electrolyte conditions, and S is generally equal to the geometric area of the electrode. The general value of $C_s$ ranges from 20–60 µF cm⁻². For our estimates of surface area, we used general specific capacitances of $C_s = 40$ µF cm⁻² in 1 M KOH based on typical reported values[113]. EIS data were collected at frequencies in the range $10^5$ to $10^{-1}$ Hz after equilibrating for 5 s in the potential of 1.6 V vs. RHE. The CP measurements were conducted at a constant current density of 10 mA cm⁻² in 1 M KOH without $iR$-compensation, using carbon paper as the working electrode, with a loading area of approximately 0.2 cm⁻². Before the CP measurement, 50 CV cycles recorded at a scan rate of 50 mV s⁻¹ with a potential window of 1 to 1.65 V (vs. RHE) were performed to activate the electrode materials. To quantify the pH-dependent OER kinetics, CV measurements in $O_2$-saturated 1 M, 0.3 M, 0.1 M, 0.03 M KOH were recorded at a scan rate of 20 mV s⁻¹ for both samples.

### Material characterization
The structural and compositional attributes of the catalysts were meticulously characterized. XRD patterns were acquired using a Bruker D8 instrument with Cu Kα radiation, providing critical insights into the crystalline structure. For XANES, XPS, TEM, and APT characterizations after various CV cycles, pure nanoparticle powder was used without the addition of carbon black or Nafion to prepare the suspension. TEM and STEM were performed on a JEOL JEM-ARM200F (NEOARM) operating at 200 kV, equipped with an Oxford X-max EDX detector for elemental mapping. XPS measurements were conducted using a setup with a HiPP-3 analyzer (Scienta Omicron) equipped with a monochromized Al $K_\alpha$ X-ray source (hv = 1486.6 eV) operating at 14.3 kV and 7.7 mA, with a 90° take-off angle. The spectra were collected under UHV conditions (analysis chamber: $10^{-8}$ mbar, analyzer: $10^{-9}$ mbar) and using a low energy electron flood gun. Spectral analysis was performed using CasaXPS software[114] version 2.3.14, and all spectra were calibrated to the C 1 s signal at a binding energy of 284.8 eV or 284.5 eV, depending on the nature of the carbon species. A standard Shirley background was used for the spectra. In situ electrochemical

Raman experiments were recorded in an inVia Renishaw Raman instrument equipped with a 50× objective microscope lens (Leica Microsystems). The OER reaction was performed in a custom-made three-electrode in situ electrochemical cell at room temperature, equipped with quartz windows, a reference hydrogen electrode (Gaskatel), platinum counter electrode, and the sample deposited on the gold foil as the working electrode. Prior to each measurement, the energy shift was calibrated using $520 \pm 0.5 \, cm^{-1}$ peak of silicon reference. A laser source with 785 nm wavenumber and $1200 \, l \, mm^{-1}$ grating was applied for the measurement. The Raman spectra were acquired at 1% laser power (~3.3 mW) with twenty consecutive scans at 3 s acquisition. $Cr(OH)_3$ and CoOOH reference materials were synthesized based on the method reported in the literature[86,115].

The DEMS measurements were performed on a home-built setup following the design principles outlined by Wolter and Heitbaum[116]. Before the measurement, the settings of the ion source of the quadrupole mass spectrometer (QMA 410, Pfeiffer Vacuum) were optimized. The used DEMS cell was the Dual Thin Layer Cell[117] and a porous PTFE membrane (Pall Inc., PTF002LH0P-SAMP, Pall Inc.) featuring an average pore size of 20 nm was used to create a vacuum/electrolyte interface, and the schematic figure are presented in Fig. S30. For the electrochemical measurements, a Hg/HgO electrode and a gold wire served as the reference and counter electrodes, respectively, while a glassy carbon electrode ($0.196 \, cm^2$) was used as the working electrode with the same mass loading of around $0.25 \, mg \, cm^{-2}$. The 100-cycle electrocatalytic activation pretreatments were carried out inside DEMS in the 1 M KOH electrolyte with normal $H_2O$. To measure the gas evolution, the electrode was held at a constant potential of 1.32 V vs. RHE until both electrochemical stability and a steady mass spectrometry baseline were achieved. Subsequently, the potential was cycled in total 15 times between 1.45 V and 1.6 V at a scan rate of 5 or $10 \, mV \, s^{-1}$ in an electrolyte containing 5% $H_2^{18}O$. After the final cycle, the potential was held at the lower limit (1.32 V vs. RHE) for 80-100 seconds, during which the electrolyte was replaced with normal water ($^{18}O$-lean electrolyte). The electrode was then subjected to repeated cycling under the same scan rate and potential window in the unlabelled electrolyte for five cycles. Throughout the entire electrochemical experiment, the mass spectrometer continuously recorded the ionic currents corresponding to m/z = 32 and m/z = 34. Multiple measurements were conducted and the error bars indicate the highest error observed in the entire measurement sequence. Hence, error bars indicate the upper limit of the error and not the standard deviation.

XAS measurements were conducted at an average nominal ring current of 300 mA in top-up and multi-bunch mode at KMC-2 beamline of the BESSY II synchrotron, operated by Helmholtz-Zentrum Berlin. The nanoparticles were measured on glassy carbon plates (loading of around $400 \, \mu g/cm^2$) and covered with Kapton tape in the pristine state and after OER conditioning. All samples were measured at room temperature in total electron yield mode, using a horizontally linearly polarized beam. The general setup was organized as follows: $I_0$ ionization chamber, sample, $I_1$ ionization chamber or fluorescence yield detector, energy reference, and $I_2$ ionization chamber. The double monochromator consisted of two Ge-graded Si(111) crystal substrates, and the beam polarization was linear horizontal. Samples were measured in fluorescence mode using a Bruker X-Flash 6/60 detector. Energy calibration of XANES was performed using the corresponding metal foil, with the inflection point set for Cr at 6989 eV. All spectra were normalized by subtracting a straight line fitted to the data before the K edge and dividing by a polynomial function fitted to the data after the K edge.

For APT specimen preparation, a bulk sample embedding the nanoparticles was fabricated by electrodeposition of Ni. Specifically, bulk nanoparticle films on glassy carbon were prepared by drop casting and dried overnight. Electrochemical CV experiments were carried out in 1 M KOD (original content 40 wt% in $D_2O$, Sigma-Aldrich) in $D_2O$ electrolyte (Sigma-Aldrich) as electrolyte at a scan rate of $50 \, mV \, s^{-1}$ from 1 V to 1.65 V vs. RHE for 100 and 1000 cycles, respectively. The glassy carbon electrode covered with nanoparticles was subsequently immersed in an electrolyte solution containing 1 g of nickel chloride, 6.0 g of nickel sulfate, and 1 g of boric acid in 20 mL of deionized water at a constant voltage of −1.0 V for 300 s for Ni electrodeposition[39]. Afterward, needle-shaped APT specimens were prepared from the nanoparticle-embedded Ni bulk sample via focused ion beam (FIB) lift-out procedures in FEI Helios G4 CX[31]. The APT experiments were conducted using a CAMECA LEAP 5000XR instrument in laser pulsing mode. APT data were acquired at a temperature of 60 K, with a laser energy of 30 pJ, a pulse frequency of 125 kHz, and detection rates of 0.5. The data were then reconstructed using AP suite 6.3.2 software. The mass spectra peaks were manually ranged by using the full-width at nine-tenths maximum method. The background was subtracted using the software's built-in global background subtraction function. The background was also subtracted from counts of individual elements for elemental composition analysis.

## Data availability
The source data and raw datasets generated and/or analysed during the current study are available in Figshare[118]. Source data are provided with this paper.

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

## Acknowledgements

M.M., K.T., T.L., and H.T. are grateful for funding from the Deutsche Forschungsgemeinschaft (DFG, German Research Foundation), Projektnummer 388390466 TRR 247 (A01, A09, B09, C01, and S projects). B.H. and T.L. acknowledge Zentrum für Grenzflächendominierte Höchstleistungswerkstoffe (ZGH) at Ruhr University Bochum for the access to infrastructure (Cameca LEAP 5000 XR, FEI Helios G4 CX SEM/FIB and JEOL JEM-ARM200F). B.H. and T.L. thank the BESSY II synchrotron and Dr. Götz Schuck. P.H. and K.T. acknowledge support from the Max Planck Fellowship Programme and the Deutsche Forschungsgemeinschaft under Germany's Excellence Strategy—EXC 2033–390677874-RESOLV. H.T. thanks the FUNCAT Centre of the Max Planck Society for the support and the Spanish Ministry of Science, Innovation and Universities for the ATRAE grant.

## Author contributions

T.L. designed, supervised, coordinated and conceptualised the project. B.H. synthesized the nanoparticles and performed the electrochemical tests. P.H., O.T., and K.T. performed the XPS measurements and analyzed the data. T.P. and S.C. performed and analysed the ICP-MS measurements. A.K. performed the TEM measurements, and A.K. and B.H. analysed the TEM data. E.B. and H.T. performed in situ Raman measurements and analysed the Raman data. C.B., P.H., and K.T. performed DEMS measurements and analysed the data. J.S. and M.M. provided XRD measurements. B.H. performed the APT experiments and analysed the APT data. T.L. and B.H. wrote the draft and all authors agreed on the contents and conclusion of the paper.

## Funding

## Competing interests

The authors declare no competing interests.

## Additional information

[1]Faculty of Mechanical Engineering, Atomic-scale Characterisation, Ruhr-Universität Bochum, Universitätsstraße 150, 44801 Bochum, Germany. [2]Max-Planck-Institut für Nachhaltige Materialien GmbH, Max-Planck-Straße 1, 40237 Düsseldorf, Germany. [3]Forschungszentrum Jülich GmbH, Helmholtz-Institute Erlangen-Nürnberg for Renewable Energy (IET-2), 91058 Erlangen, Germany. [4]Faculty of Chemistry and Biochemistry, Analytical Chemistry II, Ruhr-Universität Bochum, Universitätsstraße 150, 44801 Bochum, Germany. [5]Department of Heterogeneous Catalysis, Max-Planck-Institut für Kohlenforschung, Kaiser-Wilhelm-Platz 1, 45470 Mülheim an der Ruhr, Germany. [6]Faculty of Chemistry and Biochemistry, Laboratory of Industrial Chemistry, Ruhr-Universität Bochum, Universitätsstraße 150, 44801 Bochum, Germany. [7]Zentrum für Grenzflächendominierte Höchstleistungswerkstoffe (ZGH), Ruhr-Universität Bochum, Universitätsstraße 150, 44801 Bochum, Germany. [8]Catalysis and Energy Materials Group, IMDEA Materials Institute, Calle Eric Kandel 2, 28906 Getafe, Madrid, Spain. ✉e-mail: tong.li@rub.de

