## [Transparent Peer Review file · Nature Communications]

Atomic-scale insights into surface reconstruction and transformation in Co-Cr spinel oxides during the oxygen evolution reaction

Corresponding Author: Professor Tong Li

Version 0:

Reviewer comments:

Reviewer #1

(Remarks to the Author)

The authors state that this study explores the optimization of OER electrocatalysts by investigating the surface and subsurface reconstruction processes of the samples. Using a multimodal approach—including X-ray absorption fine structure, photoemission spectroscopy, transmission electron microscopy, atom probe tomography, and electrochemical measurements—the research examines the oxidation state, atomic coordination, structural, and compositional changes in CoCr_2O_4 and CrCo_2O_4 . In my point of view, the main novelty should be attributed to the mechanistic differences between CoCr_2O_4 and CrCo_2O_4 and the verification of these mechanisms. If this is not clearly stated, the significance of the material itself may not be as promising. Therefore, the main statement should focus on defining the role of Cr and evaluating whether the suggested mechanism is convincing or not.

(1) The ratio of $\text{Co}(\text{OH})_2$ and Co_3O_4 in Figure 3a does not match the graph.

(2) Within your TEM images, the authors mainly focus on the reconstruction layer and its material crystallinity. [1] Is it possible for performance to improve despite the minimal presence of a reconstruction layer in CoCr_2O_4 ? [2] Why does reconstruction not occur again in CoCr_2O_4 after the reconstruction layer is lost and CoCr_2O_4 is re-exposed? [3] Do you have any more detailed evidence regarding the stability of the reconstruction layer before and after ex-situ or in-situ analysis?

(3) The evidence for leached Cr forming an amorphous Cr oxyhydroxide layer on the surface seems insufficient. For example, it would be beneficial to clearly demonstrate the presence of amorphous Cr oxyhydroxide on the surface using other analytical techniques.

(4) What is the basis for specifying the exact voltage value of 1.65 V vs. RHE for significant Cr dissolution? Do you have any supporting evidence for this claim?

(5) The authors simply mentioned the intercalation of K^+ ions. What is the role or effect of K^+ ions in ion deintercalation and bonding? How does it influence the overall OER reaction?

Those questions should be clearly announced for the further evaluation.

Reviewer #2

(Remarks to the Author)

The authors employed multimodal method combining X-ray absorption fine structure and photoemission spectroscopy, transmission electron microscopy and atom probe tomography with electrochemical measurements to unveil how the changes in oxidation states, atomic coordination, structure and composition on ~20 nm CoCr₂O₄ and Co₂CrO₄ spinel nanoparticle surfaces affect OER activity and stability. The results seem interesting, but major changes have to be made before they are considered to be published anywhere.

1. Authors should use in situ Raman spectroscopy to demonstrate active phase for CoCr₂O₄ and Co₂CrO₄ spinel catalysts for OER.
2. More reaction intermediates should be monitored to prove the lattice oxygen mechanism or adsorbate evolution mechanism for activated CoCr₂O₄ and Co₂CrO₄ spinel catalysts.
3. The activation process of CoCr₂O₄ and Co₂CrO₄ spinel catalysts should be tested by using Chronopotentiometric curve.
4. The authors should discuss the variation of bond length and coordination number of Co-O and Cr-O bonds in CoCr₂O₄ and Co₂CrO₄ spinel catalysts during activation process for OER.
5. CoCr₂O₄ undergoes the activation process for OER. Why did the surface reconstruction not occur?
6. Aberration-corrected high-angle annular dark-field scanning TEM result should be consistent with theoretical atomic arrangement and FFT simulations, please refer to ACS Energy Lett. 2023, 8, 5136–5142; Chem. Engin. J. 2021, 419, 129568.
7. Changes in Tafel slope and active area during activation should be analyzed.
8. Whether Cr will be present in the surface active phase for reconstructed CoCr₂O₄ and Co₂CrO₄ spinel catalysts.
9. What is the activity of the activated CoCr₂O₄ and Co₂CrO₄ catalyst in the membrane electrode assembly?

Reviewer #3

(Remarks to the Author)

The manuscript titled "Atomic-scale insights into surface reconstruction and transformation in Co-Cr spinel oxides during the oxygen evolution reaction" has been reviewed. Although quite dense in terms of content, the manuscript is fairly well written and follows a logical flow, however there are some concerns around the references. Conclusions are well supported by experimental evidence. The paper presents analysis from a variety of complementary analytical techniques to help provide critical mechanistic understanding regarding the enhanced OER activity of CoCr₂O₄ over Co₂CrO₄ spinel nanoparticles. Specifically, the authors show that the enhanced OER activity of CoCr₂O₄ is induced by Cr dissolution, which in turn promotes the interaction of hydride to induce a surface chemical/structural transformation into the octahedral phase. It is my opinion that before this paper be considered for publication, the authors must explicitly address the critical concerns below:

1. There are some concerns about the references. For one, there are some references that are duplicated (e.g., Ref 31 and Ref 81 are the same). ALSO, there are some instances of what appear to be inappropriately used references. For example, Ref #60 seems not to be appropriate. Please thoroughly review ALL references for being appropriately used and make sure there are no duplicates.
2. ERROR BARS! There is a lack of experimental error reported. Other than APT composition profile results, all other data lacks reported error. On page 5, the authors describe "To ensure experimental reproducibility, each electrochemical measurement was conducted at least three times under the same conditions". Please incorporate these measurements into single plots with error, rather than rely on show examples of the same measurements in SI (but do keep the data in the SI still though). Small deviations in the LSV scans are reported; what is the error in these measurements to build confidence that deviations are significant to support your conclusions. Provide error descriptions throughout, including tables and reports of Co/Cr ratios and stoichiometries (e.g. Co_{0.55}Cr_{0.45}O). ALL data should have error.
3. Page 4: provide some explicit examples of what operando and in situ spectroscopy techniques lack in the sentence "Indeed, even operando and in situ spectroscopy techniques^{30,33-37} only provide few aspects of the required surface details"
4. Page 9 regarding sentence "The Cr dissolution rates in both systems are highest during the first 100 cycles, after which they drop, possibly due to Cr saturation in the electrolyte or Cr depletion from the materials". You could estimate this based on the composition within the electrolyte and then rule/confirm if in saturated state. Please do this. And if it is NOT saturated, then what does that say about "Cr depletion from the materials"?...maybe diffusion limited (i.e. as more and more Cr is leached, any more has to dissolve from further within the crystal. OR maybe there is a transient Cr/Co metal hydroxide salt forming at the surface which gets denser/thicker with cycles, lowering Cr dissolution rate? This would be more like a gel-like structure.
5. Page 10: "For Co₂CrO₄ inverse spinel, Co(II) occupies the tetrahedral sites"; the Co(II) should be Cr(II). ALSO since there are MANY instances of Co₂CrO₄ and CoCr₂O₄ throughout, please review throughout for correctness in reference to what is being described/explained (E.g., Page 19, "In comparison to Co₂CrO₄, the uppermost 4-5 atomic surface layers", should be CoCr₂O₄)
6. There is no description regarding APT based compositional analysis. How did you integrate peak counts, how do you deal with background? How did you deal background subtraction when Co-59 is flanked by Ni-58 and Ni-60 peaks from the electrodeposited Ni matrix?
7. Regarding conclusion that K and hydroxide ions are present INSIDE the CoCr₂O₄ NPs, how can you rule out aberrations leading to trajectory overlap misleading to interfaces that are artificially wide/mixed. ADDITIONALLY, related to this, how can you say with certainty that hydroxide is INSIDE, when there seems to be not many ions even counted (based on 3D distributions in Figures), meaning how can you distinguish/say that those ions shown to be INSIDE the particle are hydroxide and not background noise (WHAT IS THE S/N ratio?). One way you can do this is to isolate the INSIDE and show/analyze

those mass spectra. Please do this.

8. Additionally show APT mass spectra, at least in SI section

9. Regarding discussions around particle vertices showing enhanced Cr dissolution, please support this notion based on results from other material systems. Example: <https://doi.org/10.1016/j.seppur.2020.118014>;
<https://doi.org/10.1039/D4EN01004C>

10. Regarding the APT measurements, please discuss possible effects that the electrodeposition of the Ni matrix has on the surface composition. K is also present in the electrodeposition process.

11. What is the role of the electrolyte cation K having on the OER? Electrolyte counter ions like K can lower overpotentials by stabilizing reaction intermediates. You see it on the surface only of the Co₂CrO₄ particles but inside the CoCr₂O₄ particles. Seems like K may be playing an active role (not passive) that is being overlooked. See: <https://pubs.acs.org/doi/10.1021/acs.chemmater.7b0051760>

12. Be consistent with color coordination: E.g., K is shown in blue in all figures, except in Fig 7 where it is green

Version 1:

Reviewer comments:

Reviewer #1

(Remarks to the Author)

The current revised version of this manuscript is appropriate. Therefore, I would like to recommend the publication of this article.

Reviewer #2

(Remarks to the Author)

The authors have effectively strengthened the work and addressed the raised concerns through additional experiments. This convincingly demonstrates how the structure and composition of CoCr₂O₄ and Co₂CrO₄ spinel nanoparticles affect OER activity and stability. However, some responses still require further clarification. My specific comments are as follows:

1. During the OER process, both CoCr₂O₄ and Co₂CrO₄ catalysts form Cr-based (oxy)hydroxide layers. Why does the CoCr₂O₄ catalyst exhibit excellent stability, while the Co₂CrO₄ catalyst has poorer stability? Could it be that no highly reversible transformation of (Co²⁺ II, Cr)(OH)₂(Co³⁺ III, Cr)OOH occurs on the surface of the Co₂CrO₄ catalyst? What evidence supports this conclusion?

2. The author conducted OER under alkaline conditions. Why was there no mention of the standard catalysts under alkaline conditions in the introduction section, but instead, the commercial catalysts under acidic OER conditions were elaborated?

3. The author states that "As OER proceeds, the (oxy)hydroxide layers on Co₂CrO₄ are depleted from the surfaces, leading to deteriorating activity.". How did the author prove that the (oxy)hydroxide layers formed on the surface of CoCr₂O₄ disappeared during the OER? The author should provide high-magnification TEM images during the process of activity decrease.

4. XANES data needs to be fitted in order to understand the changes in coordination number and bond length for CoCr₂O₄ and Co₂CrO₄ catalysts. CoOOH should also be selected as the reference sample.

5. Why is there a significant shift in the O 1s XPS peak position in Figure 3c? The author should reveal the reason.

6. XPS spectra have confirmed that CoOOH is produced during the catalytic process. However, why did no CoOOH peak appear in the in-situ Raman test?

7. During the catalytic process for OER, CoOOH and CrOOH were in situ generated. What is the structural relationship between CoOOH and CrOOH on the surface of the restructured CoCr₂O₄ and Co₂CrO₄? CoOOH is the catalytically active phase. Then, does Cr exist in CoOOH? Atomic-level EDX/EELS analysis of the amorphous layer on the CoCr₂O₄ and Co₂CrO₄ catalyst surface is required.

8. In general, the original structure of pre-catalyst has an effect on the surface reconstruction degree (J. Am. Chem. Soc. 2020, 142, 12087–12095; J. Am. Chem. Soc. 2020, 142, 12087–12095; J. Am. Chem. Soc. 2024, 146, 11887–11896). The surface reconstruction degree of CoCr₂O₄ and Co₂CrO₄ catalysts should be compared. Is there a certain relationship between the original structure, reconstruction degree, and catalytic performance of CoCr₂O₄ and Co₂CrO₄ catalysts?

9. Is it the pure LOM pathway for the OER process for the CoCr₂O₄ and Co₂CrO₄ catalysts? However, the evidence is still insufficient. Here are many results that suggest that the LOM and AEM pathway are co-existent in the OER process. Did the author test the existence of AEM pathway? More reaction intermediates should be monitored to prove the lattice oxygen mechanism for activated CoCr₂O₄ and Co₂CrO₄ spinel catalysts using in situ Fourier transform infrared.

10. There are some obvious grammatical errors in this manuscript. The order of the diagrams is also quite chaotic, for example, the arrangement order of the impedance diagrams.

Reviewer #3

(Remarks to the Author)

Dear authors, I thank you for the thoroughness and thoughtfulness of your rebuttal to the specific comments of reviewer #3. I believe that your response to each concern and the edits you've made adequately address the initial comments. I have no further concerns about the manuscript.

Version 2:

Reviewer comments:

Reviewer #2

(Remarks to the Author)

The authors have addressed all the reviewers' comments, now I suggest to accept this work for publication.

Response to Reviewer's comments on Nature Communications Manuscript NCOMMS-25-05420

Reviewer #1:

The authors state that this study explores the optimization of OER electrocatalysts by investigating the surface and subsurface reconstruction processes of the samples. Using a multimodal approach—including X-ray absorption fine structure, photoemission spectroscopy, transmission electron microscopy, atom probe tomography, and electrochemical measurements—the research examines the oxidation state, atomic coordination, structural, and compositional changes in CoCr_2O_4 and CrCo_2O_4 . In my point of view, the main novelty should be attributed to the mechanistic differences between CoCr_2O_4 and CrCo_2O_4 and the verification of these mechanisms. If this is not clearly stated, the significance of the material itself may not be as promising. Therefore, the main statement should focus on defining the role of Cr and evaluating whether the suggested mechanism is convincing or not.

We appreciate Reviewer's insightful comments.

(1) The ratio of Co(OH)_2 and Co_3O_4 in Figure 3a does not match the graph.

We thank the Reviewer for this comment. That was a mistake in the colour scheme. In the revised manuscript, we conducted a thorough assessment of the XPS analysis by applying a multiplet peak-modelling strategy as described by Biesinger et al. (Applied Surface Science, 7(2011)2717). We have revised Figure 3 and added Supplementary Note 2 to explain how the XPS data were fitted.

Briefly, we fitted the Co $2p$ and Cr $2p$ spectra using peak models from literature that are based on reference spectra in which all peaks are constrained based on relative peak positions, full-width at half-maximum (FWHM) ratios, and area ratios against the first spectral component. Additionally, we constructed empirical peak models from the as-synthesized CoCr_2O_4 nanoparticles—assuming only the intended oxide species at the surface—and imposed the same position (± 0.2 eV) and FWHM (literature value to $+0.5$ eV) limits as those in the literature-based models, with the first peak's FWHM further restricted to ± 0.2 eV of the reference spectrum. Both literature-based and empirical models were then used to fit spectra acquired after 1000 CV cycles. Note that this approach cannot be used to construct peak models from the Co $2p$ and Cr $2p$ spectra of pristine Co_2CrO_4 since the surface of the particles contains rock-salt CoO structure, as indicated by our TEM data in Figure 5g-h. Because of this, we approximated the chemical/oxidation state of Co near the surface by fitting the Co $2p$ spectra using the peak model of CoO and Co_3O_4 . In addition to this approach, Monte Carlo simulations in CasaXPS were employed to compute quantitative standard deviations (σ) and assess the accuracy of all spectral fits.

We have modified the main text on page 14 and added Supplementary Note 2:

'For Co_2CrO_4 , the Cr $2p_{3/2}$ spectra in the pristine state (Figure 3e, top) could not be fitted to any reference samples due to the lack of a well-defined structure. The broadened spectra potentially suggest a mixed chemical environment or state for Cr. After 1000 cycles, the Cr $2p_{3/2}$ spectra of Co_2CrO_4 become sharper and narrower, which fit well with those of pristine CoCr_2O_4 , suggesting a similar Cr chemical

and atomic coordination environment. This might arise from the gradual exfoliation, erosion, or dissolution of the surface-reconstructed layer formed between 100 and 1000 cycles, exposing the well-defined bulk part of Co_2CrO_4 nanoparticles on the surface after 1000 cycles (which will be discussed when describing TEM data in Figure 5).'

Figure 3. Surface oxidation state and oxygen species of CoCr_2O_4 and Co_2CrO_4 before and after OER. XPS of a) Co 2p, b) Cr 2p, and c) O 1s levels of CoCr_2O_4 ; and d) Co 2p, e) Cr 2p, and f) O 1s levels of Co_2CrO_4 in the pristine state and after 1000 cycles. A standard Shirley background was applied to all spectra. Models of Co(0), CoO, Co(OH)₂, CoOOH, Co₃O₄ were considered for the fitting of the Co 2p_{3/2}. The detailed information regarding the XPS measurements and peak fitting process are presented and discussed in Supplementary Note 2.

Supplementary Note 2: XPS Section

The spectra were analyzed using the software CasaXPS version. A standard Shirley background was applied to all spectra. The spectra were charge calibrated by setting the C 1s peak to 284.8 eV or 284.5 eV depending on whether the carbon signal originated from adventitious carbon or the exposed glassy carbon substrate. To fit the Co 2p_{3/2}, Cr 2p_{3/2}, and O 1s, a Gaussian-Lorentzian product formula $GL(m)$ was used to describe the line shape of the peaks in CasaXPS with m describing the mixing and, for example, $m = 30$ being a 30% Lorentzian and 70% Gaussian line shape. All spectra were recorded with a pass energy of 200 eV in fixed transmission mode. To analyze the Co 2p and Cr 2p regions, an approach using peak models of reference models based on the work of Biesinger et al. was used to account for the multiplet splitting and assign chemical states. The peak models were constructed based on literature spectra of the Co 2p_{3/2} and Cr 2p_{3/2} of different cobalt and chromium species and the peaks were constrained regarding their relative peak positions, FWHM, and area ratios to the first peak in the respective model. This peak was additionally constrained in position (± 0.2 eV from literature model)

and FWHM (literature value to literature value +0.5 eV. Models of Co(0), CoO, Co(OH)₂, CoOOH, Co₃O₄ were considered for the fitting of the Co 2p_{3/2}.^{2,3} For the analysis of the Cr 2p_{3/2}, peak models of Cr(0), Cr₂O₃, Cr(OH)₃ and CrO₃ were considered.³ In addition, the Co 2p and Cr 2p spectra of the pristine CoCr₂O₄ nanoparticles were used to construct peak models based on a purely empirical fitting assuming that the surface of the nanoparticles consists only of the expected metal oxide species. The peak models were constrained similarly to the models based on literature, but the FWHM of the first peak was constrained to ±0.2 eV of the reference spectra. These were then employed together with the literature models to analyze the spectra of the CoCr₂O₄ sample after 1000 cycles. This approach could not be realized for the Co₂CrO₄ nanoparticles as other techniques revealed that their surface contained CoO and was Cr enriched. Instead, the state of cobalt was only approximated from the Co 2p using the models mentioned above. The best fit was achieved with a mixture of CoO and Co₃O₄. The share of CoO model in the fits exceeds the difference in Co^{II}/Co^{III} between Co₃O₄ (1:2) and Co₂CrO₄ (1:1) suggesting the presence of segregated CoO. Due to the surface enrichment of Cr on the Co₂CrO₄ nanoparticle surface, no reference model for the fitting of the Cr 2p after 1000 cycles could be constructed. In pristine Co₂CrO₄, the Cr 2p_{3/2} spectrum could not be fitted using any reference due to the lack of a well-defined structure, with the broad profile suggesting a mixed chemical environment. After 1000 cycles, the spectrum become narrower and can be fitted using the empirical model from CoCr₂O₄, indicating that Cr adopts a similar coordination environment.. Monte Carlo simulations as implemented in CasaXPS were used to calculate standard deviations (σ) of the quantification and ascertain the accuracy of the fit. After the extended electrochemical treatment of the samples, the sample with CoCr₂O₄ had a heterogeneous nanoparticle coverage presumably due to detachment during electrocatalysis. The survey spectra show no contamination aside from a K 2p signal that was observed on the electrochemically treated CoCr₂O₄ sample possibly due to residual KOH in the interlayer.

- (2) Within your TEM images, the authors mainly focus on the reconstruction layer and its material crystallinity. [1] Is it possible for performance to improve despite the minimal presence of a reconstruction layer in CoCr₂O₄? [2] Why does reconstruction not occur again in CoCr₂O₄ after the reconstruction layer is lost and CoCr₂O₄ is re-exposed? [3] Do you have any more detailed evidence regarding the stability of the reconstruction layer before and after ex-situ or in-situ analysis?

Thank you for raising these insightful questions.

Regarding [1] is it possible for the performance improvement despite minimal reconstruction in CoCr₂O₄: the surface reconstruction is **not** minimal during OER cycling since our selected area electron diffraction (SAED) pattern of 100-cycle CoCr₂O₄, shown in Figure R1a (i.e., Figure S10b), demonstrates the presence of additional phases in addition to spinel oxide. These additional phases fit well with α -Co(OH)₂, γ -CoOOH or β -CoOOH, thus, we speculate that 100-cycle CoCr₂O₄ nanoparticle surfaces might be partly transformed into α -Co(OH)₂, γ -CoOOH or β -CoOOH. More importantly, our atom probe data of 100-cycle CoCr₂O₄, shown in Figure R1c-d (i.e., Figure 6f-g, Figure S16a and Table 2), show pronounced compositional changes in the top 2-4 nm surface regions; Co/Cr ratio increases (from ~0.6 in the pristine state) to ~0.7 after 100 CV cycles and a significant amounts of hydroxyl ions and K ions are incorporated in the top 2-4 nm surface layers. By relating the TEM/SAED results with APT data, the high amount of K and hydroxide ions are most likely associated with the transition from α -type Co-based hydroxide to γ -type oxyhydroxide, wherein the incorporation and intercalation of K⁺ and OH⁻ balances the charge and stabilize the structure of the γ -type oxyhydroxide which will be discussed more clearly in Question 5.

Note that the top 2-4 nm surface-active regions can account for 60–85% of the volume of nanoparticles (with a diameter of ~18 nm), which is expected to significantly affect their activity,

as revealed by our LSV and CV data in Figure 1a, c. In addition, a considerable increase in ECSA (Figure R1e, i.e., Figure S6a) and a decrease in resistivity (Figure R1f, i.e., Figure S6c) were observed after 100 CV cycles. Our original high-resolution TEM image may not clearly show the surface reconstruction. Thus, we updated Figure 4d, as shown in Figure R1b, which now shows that the coarse surfaces of 100-cycle CoCr_2O_4 become amorphous. Additionally, the high-resolution TEM images in Figure S15a-b reveal that another surface-reconstructed layer is partially amorphous, exhibiting a different atomic arrangement from the bulk part of the nanoparticles.

Figure R1. Structural, compositional, electrical conductivity and ECSA changes of CoCr_2O_4 after 100 CV cycles: (a) Selected area electron diffraction (SAED) pattern of CoCr_2O_4 after 100 CV cycles, (b) high resolution TEM images of CoCr_2O_4 after 100 CV cycles, (c) 2D compositional maps of Cr in CoCr_2O_4 after 100 CV cycles, the OD and K distribution are also shown with their rich region marked in the white circle, (d) 1D composition profile along the vertices of the 100-cycle CoCr_2O_4 nanoparticle, (e) electrochemical impedance spectroscopy (EIS) data of CoCr_2O_4 and (f) electrochemical double-layer capacity (C_{dl}) in pristine, after 100 and after 1000 cycles for CoCr_2O_4 .

On the other hand, we notice that the thickness of the reconstructed region shown in TEM does not match that revealed by APT, which could be attributed to electron beam-induced recrystallization (especially for a high-magnification TEM image, such as Figure R1b, i.e., Figure 4d). Alternatively, the surface-reconstructed layer exhibits high structural reversibility during OER cycling. This underscores the importance of our multimodal approach. By combining the TEM, APT, EIS, and ECSA data, we reveal the drastic changes in composition, elemental distribution, ECSA, and electrical conductivity that contribute to the best OER activity among all conditions.

I have revised the text on page 20 and 21:

'After 100 cycles of OER, the activated CoCr_2O_4 nanoparticle surface becomes coarse, and the top 1-2 nm regions are amorphous, as shown in Figure 5d (see more TEM images in Figures S15a-b). The coarse surfaces of CoCr_2O_4 are likely induced by Cr dissolution, resulting in the formation of defective sites and material depletion from the nanoparticle surface. Previous work²² reported the formation of amorphous (oxy)hydroxides on the surface of activated CoCr_2O_4 . In our study, some surface regions (1-2 nm) become amorphous, and some do not become completely amorphous but exhibit a different atomic arrangement from the bulk part of the nanoparticles (Figure 5d and Figure S15a-b). Interestingly, additional reflection spots were observed in the SAED pattern of the 100-cycle CoCr_2O_4 , as indicated by the red circles in Figure S10b recorded from tens of nanoparticles, likely corresponding to the surface reconstructed layer.'

'By relating to the TEM/SAED results (Figure S10b), the high amount of K and hydroxide ions is most likely associated with α -type Co-based hydroxide \leftrightarrow γ -type oxyhydroxide transition, which is in line with the intensified Al/Cl redox peak in Figure 1c. The intercalated hydroxide and K ions balances the charges and stabilises the structure of γ -type oxyhydroxide, as observed during Ni α -hydroxide \leftrightarrow γ -oxyhydroxide transition.⁸⁰ Notably, the surface reconstructed region revealed by APT is 2-4 nm, which is thicker than that revealed by TEM (Figure 4d). This could be caused by the high structural reversibility of the surface reconstructed layer. On the other hand, we cannot eliminate the possibility that such surface layer might partially undergo the electron beam-induced recrystallization, especially at a high magnification in TEM.'

Regarding the reconstruction not occur again in CoCr_2O_4 after the reconstruction layer is lost and CoCr_2O_4 is re-exposed: the surfaces of 1000-cycle CoCr_2O_4 are not re-exposed by CoCr_2O_4 spinel. The reconstructed surface regions between 100 and 1000 CV cycles undergo materials depletion or structural collapse induced by continuous Cr leaching. This leads to the formation of 2-3 nm nanoclusters attached to the CoCr_2O_4 nanoparticle surfaces (Figure R2b, i.e., Figure 5e, and Figure S15c-d). Because of this, the thickness of the surface reconstructed regions is reduced from 2-4 nm (formed after 100 CV cycles) to 1-2 nm after 1000 cycles, as revealed by the high-resolution TEM image in Figure R2b and APT data in Figure R2c-d (i.e., Figure 6j-k). This means the surfaces of CoCr_2O_4 nanoparticles are still covered by a 1-2 nm (oxy)hydroxide, likely β - CoOOH (as indicated by our selected area electron diffraction pattern analysis in Figure R2a, i.e., Figure S10c). This β - CoOOH layer is less hydrated as it contains a considerably lower amount of hydroxide and K ions (as indicated by APT data in Figure R2c and Table 2). Despite this, the remaining thin β - CoOOH layer and oxyhydroxide clusters are still active for OER, as the 1000-cycle CoCr_2O_4 still exhibits high OER current densities (Figure 1e).

Figure R2. Structural and compositional of CoCr_2O_4 after 1000 CV cycles: (a) SAED pattern of CoCr_2O_4 after 1000 CV cycles. (b) HRTEM images of CoCr_2O_4 after 1000 CV cycles. (c) 2D compositional maps of Cr in CoCr_2O_4 after 1000 CV cycles, the OD and K distribution are also shown with their rich region marked in the white circle. (d) 1D concentration profiles of CoCr_2O_4 nanoparticle, obtained by placing analysis cylinders with a diameter of 4 nm along the direction to vertices.

To clarify this, we have amended the text on page 22:

*'This is further evidenced by our APT data, which show that the thickness of the reconstructed surface regions decreases to 1-2 nm, as revealed by reduced amounts of K and hydroxide ions in the 2D Cr compositional maps and 1D concentration profiles (Figures 6j-l, S14c, S16b and Table 2). **These results suggest the surface reconstructed regions likely transform from hydrated α/γ (oxy)hydroxide (after 100 cycles) to β -CoOOH after 1000 cycles which is typically less hydrated.'***

*'The deterioration in activity of the 1000-cycle CoCr_2O_4 (compared to the most activated 100-cycle state) may be caused by the depletion of reconstructed surface regions and **transformation to less hydrated β -CoOOH**. Although the reconstructed surface regions exfoliate gradually, they subsequently form small oxyhydroxide clusters that remain active toward OER, as the 1000-cycle CoCr_2O_4 still exhibits high OER current densities (Figure 1e).'*

Regarding additional evidence on reconstruction layer stability from ex situ/in situ analyses: Firstly, I want to stress that the surface reconstructed layer is highly reversible and undergoes dynamic changes during OER cycling, which is the novelty of this study. Specifically, Cr leaching leads to the creation of cation and oxygen vacancies, resulting in the formation of the 2-4 nm reconstructed layer on the surfaces of CoCr_2O_4 (revealed by our APT

data in Figure 6f-h) after 100 cycles, as schematically shown in Figure 8a-b (see below). This surface-reconstructed layer is likely a hydrated gamma-type $(\text{Co}^{\text{III}}, \text{Cr}^{\text{III}})\text{OOH}$ (Figure S10b), which can be reversibly formed during cycling in the first 1000 cycles (as indicated by the intensified A1/C1 redox couple in Figure 1c). After 1000 cycles, this surface-reconstructed layer undergoes exfoliation and reformation as ~ 2 nm nanoclusters (Figure 5e), which remain active for OER (Figure 1c, Figure 8c). In addition, the thickness of the surface-reconstructed regions of CoCr_2O_4 is reduced from 2-4 nm (after 100 CV cycles, Figure 6g) to 1-2 nm after 1000 cycles (Figure 6k), likely due to the formation of β - CoOOH (Figure S10c). This β - CoOOH layer is comparatively less hydrated as it contains a considerably lower amount of hydroxide and K ions (Table 2).

Thus, to answer this question, the surfaces of CoCr_2O_4 undergo highly reversible intercalation-assisted transition from alpha-type $(\text{Co}^{\text{II}}, \text{Cr}^{\text{III}})(\text{OH})_2$ to gamma-type $(\text{Co}^{\text{III}}, \text{Cr}^{\text{III}})\text{OOH}$ in the first 1000 cycles, after which the gamma-type oxyhydroxides likely lose its structural stability and transform to less-hydrated beta-type oxyhydroxide via the exit of hydroxide and K ions. Despite this, beta-type oxyhydroxide is still active for OER.

Figure 8. Schematic diagram showing surface reconstruction and transformation of CoCr_2O_4 and Co_2CrO_4 during OER. a) Cr dissolves substantially across nearly the entire CoCr_2O_4 nanoparticles, leading the formation of Cr and O vacancies that promote the hydroxide ion incorporation; b) the gradual hydroxide ion incorporation potentially facilitates reversible $(\text{Co}^{\text{II}}, \text{Cr}^{\text{III}})(\text{OH})_2 \leftrightarrow (\text{Co}^{\text{III}}, \text{Cr}^{\text{III}})\text{OOH}$ transformation, giving rise to high activity and stability of CoCr_2O_4 ; as OER proceeds, (c) the (oxy)hydroxides are gradually exfoliated from CoCr_2O_4 surfaces. In contrast, d) pristine Co_2CrO_4 surface is covered by rock-salt structured $(\text{Co}, \text{Cr})\text{O}$ which does not assist transformation of Co-based oxyhydroxide. Instead, e) amorphous $\text{Cr}(\text{OH})_3$ layer is formed on the Co_2CrO_4 surfaces, serving as active species for OER. Such self-limiting amorphous layer is gradually depleted from Co_2CrO_4 surfaces, decreasing the activity.

- (3) The evidence for leached Cr forming an amorphous Cr oxyhydroxide layer on the surface seems insufficient. For example, it would be beneficial to clearly demonstrate the presence of amorphous Cr oxyhydroxide on the surface using other analytical techniques.

We thank Reviewer's suggestion. We have performed in situ Raman spectroscopy on 100-cycle CoCr_2O_4 and Co_2CrO_4 (new Figure 4 and Section 2.3). After 100-cycle activation, the amorphous layer is expected to form on the surfaces of Co_2CrO_4 (as revealed by our TEM image in Figure 5i). In Figure 4b (see below), we observed a broad Raman peak centered at the wavenumber region of $\sim 550\text{ cm}^{-1}$, marked by blue-dashed line, is present on both samples from 1.0 to 1.55 V vs. RHE, which match with the main speak of $\text{Cr}(\text{OH})_3$ reference material (Figure S9) from the bending vibration of $\text{Cr}-\text{O}-\text{H}$ bond (Journal of Raman Spectroscopy, 48(2017)1256). Upon the anodic sweep from 1.0 to 1.55 V vs. RHE on Co_2CrO_4 , a gradual increase of the broad peak at $430 - 630\text{ cm}^{-1}$ was observed in Figure 4d, which likely comes from the evolution of major peaks that belong to $\text{Cr}(\text{OH})_3$ layer on the Co_2CrO_4 surfaces. Thus, the new in situ Raman spectroscopy confirms the formation of $\text{Cr}(\text{OH})_3$ layer during the anodic sweep on 100-cycle Co_2CrO_4 .

Figure 4. Formation of intermediate species on CoCr_2O_4 and Co_2CrO_4 during OER. In situ electrochemical Raman spectra of (a) 100-cycle CoCr_2O_4 measured stepwise in anodic scan 1 M KOH, (b) comparison overlaid spectra of Co_2CrO_4 on anodic and cathodic scans, (c) zoom in to A_{1g} band of CoCr_2O_4 , (d) CoCr_2O_4 measured stepwise in anodic scan.

Figure S9. Raman spectra of as-synthesized reference $\text{Cr}(\text{OH})_3$ and CoOOH materials.

We have incorporated a new Section 2.3 on page 4 and page 15-16:

'In situ Raman spectroscopy reveals the formation of intermediate species on the bulk electrocatalysts during the electrolytic reaction.'

'XANES and XPS measurements were performed ex situ on the nanoparticles after OER cycling, which nevertheless show that the irreversible surface transformation and dissolution occur in CoCr_2O_4 and Co_2CrO_4 after OER cycling. To investigate the evolution of active surface species during OER cycling, in situ electrochemical Raman spectroscopy was performed on activated CoCr_2O_4 and Co_2CrO_4 samples (after 100 cycles) between 1.0 and 1.59 V vs. RHE in 1 M purified KOH. Figure 4a-b shows that both samples contain five Raman modes of A_{1g} , E_g , and three F_{2g} , corresponding to structural features of the spinel phase.⁶⁹ The Raman peaks of CoCr_2O_4 appear less defined and possess asymmetric broadening on the A_{1g} band compared to Co_2CrO_4 (Figure 4a-b). The A_{1g} band at $\sim 687 \text{ cm}^{-1}$ belongs to the Raman mode of stretching vibration of $M\text{-O}$ bond at octahedral sites of the spinel phase.⁶⁹⁻⁷¹ Given that Cr (III) occupies octahedral sites for both CoCr_2O_4 and Co_2CrO_4 samples, the intensity change in the A_{1g} band can potentially provide insights into the Cr leaching during OER cycling. Interestingly, the intensity of the A_{1g} band (at $\sim 687 \text{ cm}^{-1}$) of 100-cycle CoCr_2O_4 decreases rapidly at above 1.40 V vs. RHE (Figure 4c). Such a decrease in the intensity of this A_{1g} band indicates bond distortion in the MO_6 sites, most likely due to Cr dissolution during the anodic sweep between 1.4 and 1.59 V vs. RHE. Notably, the A_{1g} band of CoCr_2O_4 is slightly blue-shifted from 688 cm^{-1} to 690 cm^{-1} upon applied potential bias (Figure 4c). This blue-shift in the A_{1g} band indicates lattice contraction and charge redistribution due to the gradual formation of the Co-based oxyhydroxide phase.⁷² Thus, the gradual dissolution of Cr from the octahedral sites may lead to the formation of Cr and O vacancies, which promote the incorporation of hydroxide ions and the formation of the active oxyhydroxide phase.⁷³

In comparison, the A_{1g} band of Co_2CrO_4 remains sharp at similar intensity, regardless of potentials (Figure 4d), which indicates the bulk stability of octahedral-coordinated Co and Cr in 100-cycle Co_2CrO_4 during OER. In addition to the A_{1g} band, a broad Raman peak centered at the wavenumber region of $\sim 550 \text{ cm}^{-1}$, marked by blue-dashed line in Figure 4a-b, is present on both samples from 1.0 to 1.55 V vs. RHE, which match with the main speak of $\text{Cr}(\text{OH})_3$ reference material (see reference spectra at Figure S9) from the bending vibration of Cr-O-H bond.⁷⁴ Upon the anodic sweep from 1.0 to 1.55 V vs. RHE on Co_2CrO_4 , a gradual increase of the broad peak at $430 - 630 \text{ cm}^{-1}$ was observed, which likely comes from the evolution of major peaks that belong to $\text{Cr}(\text{OH})_3$ layer on the Co_2CrO_4 surfaces (Figure 4d). Given that the redox activity of Co_2CrO_4 is thought to be dominated by

Cr sites after 100 CV cycles (Figure 1d), we speculate that Cr might transform to Cr(OH)₃, which contributes to the OER activity of Co₂CrO₄.

- (4) What is the basis for specifying the exact voltage value of 1.65 V vs. RHE for significant Cr dissolution? Do you have any supporting evidence for this claim?

1.65 V vs. RHE is selected for the OER occurrence. Higher than 1.65 V vs. RHE induces bursting amounts of oxygen bubbles that affect the electrochemical measurements.

At 1.65 V vs. RHE, Cr dissolution is indeed significant, as indicated by previous work (Angew. Chem., 60(2021)7418-7425. Figure 2b of this work shows that the cumulative mass of leached Cr in CoCr₂O₄ increases from ~0.02 µg at 1.56 V vs. RHE to ~0.08 µg at 1.66 V vs. RHE during chronoamperometry measurements in 1 M KOH. We have cited this work across our main manuscript (note that they stressed the importance of the highest OER potential, as substantial Cr leaching is essential for the activation. In our study, we propose that the low potential range 1.0 V vs. RHE is more critical as the transition Co(+2)/Co(+2, +3) must be triggered to promote the alpha/gamma (oxy)hydroxide transformation, as evidenced by our comparison CV/LSV measurements between 1.2 V – 1.65 V vs. RHE in Figure S7 where CoCr₂O₄ cannot be activated).

In addition, our new in-situ Raman spectroscopy data (Figure 4c) also reveal that the decrease in intensity of this A_{1g} band, typically attributed to bond distortion in the octahedrally coordinated MO₆ sites, decreases significantly at >1.4 V vs. RHE. This confirms the occurrence of pronounced Cr leaching (at OER potential) as the voltage increases above 1.4 V vs. RHE.

- (5) The authors simply mentioned the intercalation of K⁺ ions. What is the role or effect of K⁺ ions in ion deintercalation and bonding? How does it influence the overall OER reaction?

We thank Reviewer's comment as this was not discussed in the original manuscript.

The enhanced OER activity and stability of CoCr₂O₄ rely on the highly reversible intercalation-assisted α-type hydroxide ↔ γ-type oxyhydroxide. According to Bode's early study (Electrochimica Acta, 11(1966)1079) on the Ni electrocatalysts, the transformation from alpha hydroxide to hydrated gamma oxyhydroxide requires the intercalation of hydroxide and K ions to balance the charges during anodic sweep, according to the equation below. Upon cathodic sweep, K and hydroxide ions de-intercalate from the gamma oxyhydroxide and transform back to the alpha hydroxide.

Thus, K⁺ serves as a charge-balancing ion. Our APT data (Table 2 and Figure 16a) show that a significant amount of K⁺ is present on the surfaces, and some is even present in the interior of 100-cycle CoCr₂O₄ nanoparticles. By relating to the TEM/SAED results (Figure S10b), the surfaces of 100-cycle CoCr₂O₄ most likely undergo the intercalation-assisted α-type hydroxide ↔ γ-type oxyhydroxide transition. The incorporation and intercalation of K⁺ and OH⁻ balances the charge of γ-type oxyhydroxide, which is subsequently transformed back to alpha-type hydroxide via deintercalation of K⁺ and OH⁻ upon cathodic sweep. Thus, ion intercalation and deintercalation are essential for maintaining the reversible surface transformation, which is key for the activity and stability.

In brief, the K presence can stabilize the negatively charged hydroxide-rich environment, stabilizing the structure of gemma-type oxyhydroxide. In addition, a recent study (Angew. Chem. Int. Ed. 2023, 135, e202313886) also speculated that the presence of K^+ might stabilize or promote the formation of the higher oxidation state of Co-containing active species. Another study on alkali-ion effects (e.g., J. Phys. Chem. C 2020, 124, 27, 14734–14741) suggests that larger cations like K^+ modulate the local electric field and proton transfer dynamics, and can stabilize the intermediates more effectively, thereby indirectly affecting the OER activity.

Based on this, we modified the text on page 21:

'By relating to the TEM/SAED results (Figure S10b), the high amount of K and hydroxide ions is most likely associated with α -type Co-based hydroxide \leftrightarrow γ -type oxyhydroxide transition, which is in line with the intensified Al/Cl redox peak in Figure 1c. The intercalated hydroxide and K ions balances the charges and stabilises the structure of γ -type oxyhydroxide, as observed during Ni α -hydroxide \leftrightarrow γ -oxyhydroxide transition.⁸⁰ Notably, the surface reconstructed region revealed by APT is 2-4 nm, which is thicker than that revealed by TEM (Figure 4d). This could be caused by the high structural reversibility of the surface reconstructed layer. On the other hand, we cannot eliminate the possibility that such surface layer might partially undergo the electron beam-induced recrystallization, especially at a high magnification in TEM.'

Reviewer #2:

The authors employed multimodal method combining X-ray absorption fine structure and photoemission spectroscopy, transmission electron microscopy and atom probe tomography with electrochemical measurements to unveil how the changes in oxidation states, atomic coordination, structure and composition on ~20 nm CoCr_2O_4 and Co_2CrO_4 spinel nanoparticle surfaces affect OER activity and stability. The results seem interesting, but major changes have to be made before they are considered to be published anywhere.

1. Authors should use in situ Raman spectroscopy to demonstrate active phase for CoCr_2O_4 and Co_2CrO_4 spinel catalysts for OER.

We thank the reviewer for this suggestion. To address this comment, we have performed in situ Raman spectroscopy measurements on the activated CoCr_2O_4 and Co_2CrO_4 (after 100 CV cycles).

We have included a new Section 2.3 on pages 4 and 15-16, a new Figure 4, Figure S9, and added text in the Experimental Section.

'In situ Raman spectroscopy reveals the formation of intermediate species on the bulk electrocatalysts during the electrolytic reaction.'

'XANES and XPS measurements were performed ex situ on the nanoparticles after OER cycling, which nevertheless show that the irreversible surface transformation and dissolution occur in CoCr_2O_4 and Co_2CrO_4 after OER cycling. To investigate the evolution of active surface species during OER cycling, in situ electrochemical Raman spectroscopy was performed on activated CoCr_2O_4 and Co_2CrO_4 samples (after 100 cycles) between 1.0 and 1.59 V vs. RHE in 1 M purified KOH. Figure 4a-b shows that both samples contain five Raman modes of A_{1g} , E_g , and three F_{2g} , corresponding to structural features of the spinel phase.⁶⁹ The Raman peaks of CoCr_2O_4 appear less defined and possess asymmetric broadening on the A_{1g} band compared to Co_2CrO_4 (Figure 4a-b). The A_{1g} band at $\sim 687\text{ cm}^{-1}$ belongs to the Raman mode of stretching vibration of M–O bond at octahedral sites of the spinel phase.⁶⁹⁻⁷¹ Given that Cr (III) occupies octahedral sites for both CoCr_2O_4 and Co_2CrO_4 samples, the intensity change in the A_{1g} band can potentially provide insights into the Cr leaching during OER cycling. Interestingly, the intensity of the A_{1g} band (at $\sim 687\text{ cm}^{-1}$) of 100-cycle CoCr_2O_4 decreases rapidly at above 1.40 V vs. RHE (Figure 4c). Such a decrease in the intensity of this A_{1g} band indicates bond distortion in the MO_6 sites, most likely due to Cr dissolution during the anodic sweep between 1.4 and 1.59 V vs. RHE. Notably, the A_{1g} band of CoCr_2O_4 is slightly blue-shifted from 688 cm^{-1} to 690 cm^{-1} upon applied potential bias (Figure 4c). This blue-shift in the A_{1g} band indicates lattice contraction and charge redistribution due to the gradual formation of the Co-based oxyhydroxide phase.⁷² Thus, the gradual dissolution of Cr from the octahedral sites may lead to the formation of Cr and O vacancies, which promote the incorporation of hydroxide ions and the formation of the active oxyhydroxide phase.⁷³

In comparison, the A_{1g} band of Co_2CrO_4 remains sharp at similar intensity, regardless of potentials (Figure 4d), which indicates the bulk stability of octahedral-coordinated Co and Cr in 100-cycle Co_2CrO_4 during OER. In addition to the A_{1g} band, a broad Raman peak centered at the wavenumber region of $\sim 550\text{ cm}^{-1}$, marked by blue-dashed line in Figure 4a-b, is present on both samples from 1.0 to 1.55 V vs. RHE, which match with the main speak of $\text{Cr}(\text{OH})_3$ reference material (see reference spectra at Figure S9) from the bending vibration of Cr–O–H bond.⁷⁴ Upon the anodic sweep from 1.0 to 1.55 V vs. RHE on Co_2CrO_4 , a gradual increase of the broad peak at $430 - 630\text{ cm}^{-1}$ was observed, which likely comes from the evolution of major peaks that belong to $\text{Cr}(\text{OH})_3$ layer on the Co_2CrO_4 surfaces (Figure 4d). Given that the redox activity of Co_2CrO_4 is thought to be dominated by

Cr sites after 100 CV cycles (Figure 1d), we speculate that Cr might transform to $\text{Cr}(\text{OH})_3$, which contributes to the OER activity of Co_2CrO_4 .

Figure 4. Formation of intermediate species on CoCr_2O_4 and Co_2CrO_4 during OER. In situ electrochemical Raman spectra of (a) 100-cycle CoCr_2O_4 measured stepwise in anodic scan 1 M KOH, (b) comparison overlaid spectra of Co_2CrO_4 on anodic and cathodic scans, (c) zoom in to A_{1g} band of CoCr_2O_4 , (d) CoCr_2O_4 measured stepwise in anodic scan.

Figure S9. Raman spectra of as-synthesized reference $\text{Cr}(\text{OH})_3$ and CoOOH materials.

'In situ electrochemical Raman experiments were recorded in an inVia Renishaw Raman instrument equipped with a 50× objective microscope lens (Leica Microsystems). The OER reaction was performed in a custom-made three-electrode in situ electrochemical cell at room temperature, equipped with quartz windows, a reference hydrogen electrode (Gaskatel), platinum counter electrode, and sample deposited on the gold foil as the working electrode. Prior to each measurement, the energy shift was calibrated using $520 \pm 0.5 \text{ cm}^{-1}$ peak of silicon reference. A laser source with 785 nm wavenumber and 1200 l mm^{-1} grating was applied for the measurement. The Raman spectra were acquired at 1% laser power ($\sim 3.3 \text{ mW}$) with twenty consecutive scans at 3 s acquisition. $\text{Cr}(\text{OH})_3$ and CoOOH reference materials were synthesized based on the method reported in the literature.'

2. More reaction intermediates should be monitored to prove the lattice oxygen mechanism or adsorbate evolution mechanism for activated CoCr_2O_4 and Co_2CrO_4 spinel catalysts.

We appreciate the reviewer's constructive suggestion. To address this point, we performed isotope-labelled differential electrochemical mass spectrometry (DEMS) measurements to interrogate the lattice oxygen mechanism or adsorbate evolution mechanisms on activated CoCr_2O_4 and Co_2CrO_4 . Specifically, 100-cycle electrocatalytic activation pretreatments were performed inside DEMS using the normal electrolyte (1 M KOH in H_2O). DEMS measurements were subsequently carried out, first in 5% H_2^{18}O in H_2O (^{18}O -rich electrolyte) for 15 cycles, followed by another six cycles in normal electrolyte (1 M KOH in H_2O). $^{18}\text{O}^{16}\text{O}$ mass spectrometric signals were not measured as they would yield a χ of only $\sim 0.25\%$ in 5% H_2^{18}O in H_2O , which is too low for reliable evaluation (Figure S26-S27). Thus, Figure S28 shows how the fraction of $\chi(^{16}\text{O}^{18}\text{O})$ changes when the electrolyte is changed from ^{18}O -rich (5% H_2^{18}O) to normal electrolyte at various cycles. $\chi(^{16}\text{O}^{18}\text{O})$ is calculated from the following equation, $\chi(^{16}\text{O}^{18}\text{O}) = ^{18}\text{O}^{16}\text{O} / (^{18}\text{O}^{16}\text{O} + ^{16}\text{O}^{16}\text{O})$.

Our DEMS data in Figure S28 (see below) show, after activation in the 5% H_2^{18}O isotope labelled electrolyte, the molar fraction $\chi(^{16}\text{O}^{18}\text{O})$ ($m/z = 34$) is $\sim 0.9\%$ in 100-cycle Co_2CrO_4 and $\sim 0.6\%$ for CoCr_2O_4 in the normal electrolyte (^{18}O -lean electrolyte, blue-coloured regions), which is slightly higher than ~ 0.4 , which is the $\chi(^{16}\text{O}^{18}\text{O})$ theoretical value from the normal water with 0.2% H_2^{18}O natural abundance (Energy & Environmental Science, 15(2022)1988, Electrochemistry Communications, 9(2007)1969). This suggests that lattice oxygen mechanisms might occur in both samples. Because ^{18}O -containing species would cover the surfaces of electrocatalysts in 5% H_2^{18}O isotope-labelled electrolyte before cycling in normal water. Such ^{18}O -containing species would generate a molar fraction of $^{18}\text{O}^{16}\text{O}$ ($m/z = 34$) larger than ~ 0.4 . However, it is difficult to make a conclusive statement as $\chi(^{16}\text{O}^{18}\text{O})$ for CoCr_2O_4 is close to ~ 0.4 with error bars. However, such low $\chi(^{16}\text{O}^{18}\text{O})$ values could be induced by the exchange in the electrolyte, whereby the sudden change in electrolyte concentration and local pH at the electrode/electrolyte interfaces may alter the stability of the active species formed on the electrocatalyst surfaces. In particular, for 100-cycle CoCr_2O_4 , the transition or dynamics from hydroxide to active oxyhydroxide, assisted by intercalation, could be interrupted, or some of the active species might be removed during the exchange of electrolyte. Therefore, we refrain from making conclusive statements based solely on DEMS measurements. Our pH-dependent OER activity measurement (Figure S25) indicates that the activated 100-cycle CoCr_2O_4 is highly pH-dependent, suggesting non-concerted photo-electron transfer steps.

To address this comment, we have added text on page 30, Supplementary Note 4 and Figure S26-28 and text in the Experimental section:

'Additionally, one might speculate that the improved OER activity of CoCr_2O_4 is associated with the lattice oxygen mechanism.¹¹¹ However, this is challenging to assess as CoCr_2O_4 requires activation processes, wherein the surface lattice oxygen anions dissolve due to the formation of soluble CrO_4^{2-} during the first 100 CV cycles. Essentially, the enhanced OER activity of activated CoCr_2O_4 is dominated by in situ surface reconstructed layer formed via (de)intercalation processes. Thus, we refrain from making conclusive statements, although efforts have been devoted to performing isotope-labelled operando differential electrochemical mass spectrometry (DEMS) measurements on activated CoCr_2O_4 and Co_2CrO_4 (see discussion in Figure S26-28 and Supplementary Note 4).'

'Supplementary Note 4: DEMS measurements

Isotope-labeled operando DEMS measurements were performed on CoCr_2O_4 and Co_2CrO_4 to investigate the reaction mechanisms regarding the lattice oxygen involvement (see Figure S26-S28). The 100-cycle electrocatalytic activation pretreatments were carried out inside DEMS in 1 M KOH electrolyte (in normal H_2O). Figures S26 and S27 show the results of the DEMS measurement that were combined with an exchange from a 5% H_2^{18}O isotope-labelled electrolyte (^{18}O -rich electrolyte) to the normal water with 0.2% H_2^{18}O natural isotope abundance (^{18}O -lean electrolyte, normal water). The upper panels in Figure S26 and S27 show the faradaic current measured by the potentiostat. 15 cycles were recorded in the electrolyte featuring 5% H_2^{18}O (^{18}O -rich electrolyte). The left side of Figures S26 and S27 shows only the last 5 cycles before the exchange to the ^{18}O -lean electrolyte, and the right side shows the results in normal water (^{18}O -lean electrolyte). The panels in the middle show the mass spectroscopic response (i.e. ionic current) for mass 32, which is proportional to the formation rate of $^{16}\text{O}_2$ and the bottom panels show the mass spectroscopic response for mass 34, which is proportional to the formation rate of $^{16}\text{O}^{18}\text{O}$. Figure S28 summarizes the molar fraction of $\chi(^{16}\text{O}^{18}\text{O})$ changes as a function of the cycle number when the electrolyte was changed from ^{18}O -rich (5% H_2^{18}O) to ^{18}O -lean (normal H_2O) electrolyte at various cycles. To this end the mass spectroscopic signals for mass 32 and 34 were integrated. $\chi(^{16}\text{O}^{18}\text{O})$ was then calculated by dividing the integral for the mass spectroscopic signal for mass 34 by the sum of mass 34 and 32. Mass 36 ($^{18}\text{O}^{18}\text{O}$) was not measured as it yields $\sim 0.25\%$ molar fraction using 5% H_2^{18}O isotope-labelled electrolyte, which is insufficient for reliable evaluation due to high statistical error.

In Figure S28, the molar fraction $\chi(^{16}\text{O}^{18}\text{O})$ for 100-cycle Co_2CrO_4 is $\sim 0.9\%$ and $\sim 0.6\%$ for 100-cycle CoCr_2O_4 , slightly higher than ~ 0.4 , which is the theoretical value of $\chi(^{16}\text{O}^{18}\text{O})$ from the normal water with 0.2% H_2^{18}O natural abundance.^{5,6} This suggests that lattice oxygen mechanisms might occur in both samples. Because ^{18}O -containing species would cover the surfaces of electrocatalysts in 5% H_2^{18}O isotope-labelled electrolyte before cycling in normal water. Such ^{18}O -containing species would generate a molar fraction of $^{18}\text{O}^{16}\text{O}$ ($m/z = 34$) larger than ~ 0.4 .⁵ As OER proceeds in the normal water, the amount of oxygen that contains ^{18}O is expected to decrease due to the gradual consumption of the ^{18}O -containing species on the electrocatalyst surfaces. However, the values of $\chi(^{16}\text{O}^{18}\text{O})$ do not decrease steadily in the normal electrolyte and it drops suddenly from 5% H_2^{18}O electrolyte to normal water (0.2% H_2^{18}O), inferring that lattice oxygen might not be involved dominantly. Also, the $\chi(^{16}\text{O}^{18}\text{O})$ for 100-cycle CoCr_2O_4 is close $\sim 0.4\%$ with the error bars. This makes us refrain from concluding that lattice oxygen mechanisms occur. One possible explanation is that 100-cycle CoCr_2O_4 is activated via the (de)intercalation processes. In this case, the oscillations exceeding 0.4% could also likely be attributed to the disproportionate release of residual hydroxide ions and water molecules trapped within the interlayer during the labelling process in ^{18}O -rich electrolyte. Additionally, the transition or dynamics from hydroxide to active oxyhydroxide on 100-cycle CoCr_2O_4 , facilitated by intercalation, could be interrupted, or some of the active species may be removed during electrolyte exchange. Therefore, we refrain from making conclusive statements based solely on DEMS measurements.'

Figure S26. DEMS results of the electrolyte exchange experiment conducted at Co_2CrO_4 . Top Panels: Faradaic current measured by the potentiostat. Centre: Mass spectrometric signal for mass 32. Bottom: Mass spectrometric signal for mass 34. The mass spectrometer measures the signal in current, but Faraday's laws are not applicable. Left: measured in 1 M KOH featuring 5% H_2^{18}O . Right: measured in 1 M KOH in normal water featuring with natural isotope abundance. Conditions: 5 mV/s, electrolyte flow rate 5 $\mu\text{L/s}$. The electrolyte exchange was performed at 0.53 V vs. Hg/HgO and took 78 seconds. Prior to the experiment Co_2CrO_4 was activated by performing 100 cycles in 1 M KOH electrolyte with normal water.

Figure S27. DEMS results of the electrolyte exchange experiment conducted at CoCr_2O_4 . Top Panels: Faradaic current measured by the potentiostat. Centre: Mass spectrometric signal for mass 32. Bottom: Mass spectrometric signal for mass 34. The mass spectrometer measures the signal in current, but Faraday's laws are not applicable. Left: measured in 1 M KOH featuring 5% H_2^{18}O . Right: measured in 1 M KOH in normal water featuring with natural isotope abundance. Conditions: 10 mV/s, electrolyte flow rate 5 $\mu\text{L/s}$. The electrolyte exchange was conducted at 0.54 V vs. Hg/HgO and took 98 seconds. Prior to the experiment CoCr_2O_4 was activated by performing 100 cycles in the 1 M KOH electrolyte with normal water.

Figure S28. $\chi(^{16}\text{O}^{18}\text{O})$ changes with cycles, which is defined as the fraction of $^{18}\text{O}^{16}\text{O} \cdot 100 / (^{18}\text{O}^{16}\text{O} + ^{16}\text{O}^{16}\text{O})$ versus the cycle number. Between the 5th and the 6th the exchange from the electrolyte featuring 5% H₂¹⁸O to the ¹⁸O-lean electrolyte with natural abundance is performed. $\chi(^{16}\text{O}^{18}\text{O})$ was determined from the experimental values shown in Figure S26 and Figure S27, respectively.

The DEMS measurements were performed on a setup that was homebuilt following the design principles outlined by Wolter and Heitbaum.¹¹⁴ Prior to the measurement the settings of ion source of the quadrupole mass spectrometer (QMA 410, Pfeiffer Vacuum) were optimized. The used DEMS cell was the Dual Thin Layer Cell¹¹⁵ and a porous PTFE membrane (Pall Inc., PTF002LHOP-SAMP) was used to create a vacuum/electrolyte interface. For the electrochemical measurements, a Hg/HgO electrode and a gold wire served as the reference and counter electrodes, respectively, while a glassy carbon electrode was used as the working electrode. The 100-cycle electrocatalytic activation pretreatments were carried out inside DEMS in the 1 M KOH electrolyte with normal H₂O. To measure the gas evolution, the electrode was held at a constant potential of 1.32 V vs. RHE until both electrochemical stability and a steady mass spectrometry baseline were achieved. Subsequently, the potential was cycled in total 15 times between 1.45 V and 1.6 V at a scan rate of 5 or 10 mV s⁻¹ in an electrolyte containing 5% H₂¹⁸O. After the final cycle, the potential was held at the lower limit (1.32 V vs. RHE) for 80-100 seconds, during which the electrolyte was replaced with normal water (¹⁸O-lean electrolyte). The electrode was then subjected to repeated cycling under the same scan rate and potential window in the unlabelled electrolyte for 5 cycles. Throughout the entire electrochemical experiment, the mass spectrometer continuously recorded the ionic currents corresponding to m/z = 32 and m/z = 34. Multiple measurements were conducted and the error bars indicate the highest error observed in the entire measurement sequence. Hence, error bars indicate the upper limit of the error and not the standard deviation.'

3. The activation process of CoCr₂O₄ and Co₂CrO₄ spinel catalysts should be tested by using Chronopotentiometric curve.

Chronopotentiometric measurements have been performed on both samples at a constant current density of 10 mA cm⁻² in 1.0 M KOH, Figure S3b (see below). CoCr₂O₄ (black curve) shows clear activation behaviour during the initial 2 hours, as evidenced by a gradual decrease

in overpotential, consistent with the redox activation and intercalation processes discussed in Section 2.1 and Figure 1. After activation, CoCr_2O_4 maintains a low and stable overpotential for nearly 100 hours, indicating excellent operational durability. In contrast, Co_2CrO_4 maintains a stable potential only for ~60 hours, after which the potential rises, suggesting activity deterioration and relatively lower stability.

Figure S3. (a) Linear sweep voltammetry (LSV) plots of pristine states for both samples that are normalized by electrochemical active surface area (ECSA) with corresponding overpotential at $100 \mu\text{A cm}^{-2}$. (b) Long-term stabilities of both CoCr_2O_4 and Co_2CrO_4 . The chronopotentiometry (CP) measurements were employed at a constant current density of 10 mA cm^{-2} in 1 M KOH with carbon paper as the working electrode and the loading area is around 0.2 cm^2 . Before the measurement, 50 cyclic voltammetry (CV) cycles were performed to allow the electrolyte penetration and activate the electrode materials.

- The authors should discuss the variation of bond length and coordination number of Co-O and Cr-O bonds in CoCr_2O_4 and Co_2CrO_4 spinel catalysts during activation process for OER.

We agree with the Reviewer that an EXAFS analysis of these data would be valuable; however, beamtime use is limited and requires a lengthy process, from initial beamtime application to scheduling and finally measurement. Unfortunately, the data we could collect for these samples does not allow for high-quality EXAFS analysis. The signal from the electrode was weak, and the available beamtime did not yield a spectrum of sufficient quality for a reliable

EXAFS analysis. Although EXAFS analysis could still be performed using our current data, we refrained from including this data in the Supporting Information or main text to avoid overclaiming the structural changes.

Additionally, when EXAFS is collected in fluorescence mode—as required for samples deposited on electrodes—self-absorption effects are common. Additionally, the measurement time for high-quality EXAFS analysis is significantly longer than that for conventional XANES measurements. Such self-adsorption effects for long measurement time can potentially alter the chemical state and bond length, etc., of the samples. Additionally, the species of interest are loosely bound to the electrocatalyst surfaces, which can be detected on the surface due to the self-adsorption effect. This increases the uncertainty in the EXAFS analysis and limits the reliability of determining the coordination number.

Although EXAFS fitting was not explicitly included in this manuscript, we provide qualitative insights based on our Co and Cr K-edge XANES and pre-edge feature analysis in Figure 2. We are confident in our pre-edge analysis, which is extremely sensitive to changes in the first coordination sphere of the absorbing metal ion, as amply discussed in numerous XAS textbooks and review articles, like [XAFS for Everyone, by Scott Calvin, CRC Press, 2013].

5. CoCr₂O₄ undergoes the activation process for OER. Why did the surface reconstruction not occur?

We thank Reviewer for raising this point. Surface reconstruction occurs, as evidenced by the following results summarized in Figure R1. The TEM/selected area electron diffraction (SAED) pattern of 100-cycle CoCr₂O₄ nanoparticles, shown in Figure R1a (i.e., Figure S10b), reveals the presence of additional phase(s) in addition to spinel oxide. These additional phases fit well with α -Co(OH)₂, γ -CoOOH or β -CoOOH, thus, we speculate that 100-cycle CoCr₂O₄ nanoparticle surfaces might be partly transformed into α -Co(OH)₂, γ -CoOOH or β -CoOOH. More importantly, our APT data of 100-cycle CoCr₂O₄, shown in Figure R1c-d (i.e., Figure 6f-g, Figure S14b, Figure S16a and Table 2), show pronounced compositional changes in the top 2-4 nm surface regions; Co/Cr ratio increases (from ~0.6 in the pristine state) to ~0.7 after 100 CV cycles and a significant amounts of hydroxyl ions and K ions are incorporated in the top 2-4 nm surface layers. By relating the TEM/SAED results with APT data, the high amount of K and hydroxide ions is most likely associated with the transition from α -type Co-based hydroxide to β -type oxyhydroxide. In addition, a considerable increase in ECSA (Figure R1e, i.e., Figure S6a) and a decrease in resistivity (Figure R1f, i.e., Figure S6c) were observed. Thus, surface reconstruction occurs, resulting in changes to the composition and elemental distribution.

Our original high-resolution TEM image may not clearly show the surface reconstruction. Thus, we updated Figure 4d, as shown in Figure R1b, which now shows that the coarse surfaces of 100-cycle CoCr₂O₄ become amorphous. Additionally, the high-resolution TEM images in Figure S15a-b reveal a partially amorphous surface-reconstructed layer, exhibiting a distinct atomic arrangement from the bulk part of the nanoparticles. We do notice that the thickness of the reconstructed region shown in TEM (Figure R1b) does not match that revealed by APT (Figure R1c-d), which could be induced by electron beam-induced recrystallization (especially for high-magnification TEM images, like Figure R1b). Alternatively, the surface reconstructed layer has high structural reversibility during OER cycling, which explains why no oxidation change was observed in our XANES data (Figure 2a).

Figure R1. Structural, compositional, electrical conductivity and ECSA changes of CoCr_2O_4 after 100 CV cycles: (a) Selected area electron diffraction (SAED) pattern of CoCr_2O_4 after 100 CV cycles, (b) high resolution TEM images of CoCr_2O_4 after 100 CV cycles, (c) 2D compositional maps of Cr in CoCr_2O_4 after 100 CV cycles, the OD and K distribution are also shown with their rich region marked in the white circle, (d) 1D composition profile along the vertices of the 100-cycle CoCr_2O_4 nanoparticle, (e) electrochemical impedance spectroscopy (EIS) data of CoCr_2O_4 and (f) electrochemical double-layer capacity (C_{dl}) in pristine, after 100 and after 1000 cycles for CoCr_2O_4 .

We have revised the text on page 20 and 21:

'After 100 cycles of OER, the activated CoCr_2O_4 nanoparticle surface becomes coarse, and the top 1-2 nm regions are amorphous, as shown in Figure 5d (see more TEM images in Figures S15a-b). The coarse surfaces of CoCr_2O_4 are likely induced by Cr dissolution, resulting in the formation of defective sites and material depletion from the nanoparticle surface. Previous work²² reported the formation of amorphous (oxy)hydroxides on the surface of activated CoCr_2O_4 . In our study, some surface regions (1-2 nm) become amorphous, and some do not become completely amorphous but exhibit a different atomic arrangement from the bulk part of the nanoparticles (Figure 5d and Figure S15a-b). Interestingly,

additional reflection spots were observed in the SAED pattern of the 100-cycle CoCr_2O_4 , as indicated by the red circles in Figure S10b recorded from tens of nanoparticles, likely corresponding to the surface reconstructed layer.'

'By relating to the TEM/SAED results (Figure S10b), the high amount of K and hydroxide ions is most likely associated with α -type Co-based hydroxide \leftrightarrow γ -type oxyhydroxide transition, which is in line with the intensified Al/C1 redox peak in Figure 1c. The intercalated hydroxide and K ions balances the charges and stabilises the structure of γ -type oxyhydroxide, as observed during Ni α -hydroxide \leftrightarrow γ -oxyhydroxide transition.⁸⁰ Notably, the surface reconstructed region revealed by APT is 2-4 nm, which is thicker than that revealed by TEM (Figure 4d). This could be caused by the high structural reversibility of the surface reconstructed layer. On the other hand, we cannot eliminate the possibility that such surface layer might partially undergo the electron beam-induced recrystallization, especially at a high magnification in TEM.'

- Aberration-corrected high-angle annular dark-field scanning TEM result should be consistent with theoretical atomic arrangement and FFT simulations, please refer to ACS Energy Lett. 2023, 8, 5136–5142; Chem. Engin. J. 2021, 419, 129568.

We thank Reviewer's suggestion. We have performed the theoretical atomic arrangement and FFT simulations in Figure R3 along [110] axis. The HAADF-STEM images were cropped from Figure 5b and Figure 5g, respectively. Figure R3 shows the overlay of the simulated structure with HAADF-STEM images. Note that oxygen anions are not detectable in HAADF-STEM mode. The simulated structures only showing the cations and their corresponding simulated FFTs, shown in Figure R3, fit perfectly with the HAADF-STEM images and FFT images in Figure 5b-c, 5g-h.

Figure R3. HAADF-STEM image cropped from (a) Figure 5b for pristine CoCr_2O_4 and (b) Figure 5g for pristine Co_2CrO_4 overlapped with theoretical atomic arrangement with or without oxygen anions along [110] zone axis and bottom images showing FFT simulations.

We have added insets in Figure 5b and 5g to show the theoretical arrangement (see below).

Figure 5. Structural and morphological changes of the nanoparticle. a) High-resolution TEM images of CoCr_2O_4 in pristine state, b) corresponding experimental HAADF-STEM images observed in $[110]$ orientation; The corresponding FFT images in c) are taken from the white cubic regions marked in b); High-resolution TEM images of CoCr_2O_4 in d) after 100 cycles and e) after 1000 cycles with zoom-in surface regions show on the right that are taken from the yellow cubic region, respectively; f) High-resolution TEM images of Co_2CrO_4 in pristine state and g) corresponding experimental HAADF-

STEM images observed in [110] orientation; The corresponding FFT images in h) are taken from the white cubic regions marked in g); High-resolution TEM images of Co_2CrO_4 in i) after 100 cycles and j) after 1000 cycles with zoom-in surface regions show on the right that are taken from the yellow cubic region, respectively. **Simulated crystal structures are overlapped with the HAADF-STEM image in the insets of Figure 5b and 5g, with Cr showing in red and Co in blue.**

7. Changes in Tafel slope and active area during activation should be analyzed.

We agree that evaluating the evolution of the Tafel slope and electrochemically active surface area (ECSA) during the activation process provides valuable insights into both the kinetics and surface accessibility of the electrocatalysts. We have added new Figure S4 and ECSA changes were incorporated in Figure S6a-b. The Tafel slopes in Figure S6a-b show that the charge transfer kinetics of CoCr_2O_4 (~67 mV/dec) is faster than that of Co_2CrO_4 (~81 mV/dec) at the onset of OER. After the activation (100 CV cycles), the Tafel slope of CoCr_2O_4 decreases slightly (~65 mV/dec), followed by a slight increase even after 2000 cycles (~72 mV/dec), while the Tafel slope of Co_2CrO_4 increases to ~94 mV/dec after 1000 cycles.

Regarding ECSA, its value for CoCr_2O_4 increases significantly from 13.5 cm^2 in the pristine state to 52 cm^2 during the first 100 CV cycles, consistent with the observed activation behaviour (Figure 1a, 1c). In contrast, although Co_2CrO_4 also shows a moderate ECSA increase (from 13.5 cm^2 to 32.5 cm^2), it does not exhibit OER activation, suggesting that surface area increase alone is insufficient to reflect activation, as already discussed in our main text on page 9.

We have added text to describe the Tafel slope results on page 6:

‘Additionally, Tafel slopes of CoCr_2O_4 and Co_2CrO_4 , measured from the LSV curves in Figure 1a-b and summarized in Figure S4a-b, show that charge transfer kinetics of CoCr_2O_4 (~67 mV/dec) is faster than that of Co_2CrO_4 (~81 mV/dec) at the onset of OER. After the activation (100 CV cycles), the Tafel slope CoCr_2O_4 decreases slightly (~65 mV/dec) followed by a slight increase even after 2000 cycles (~72 mV/dec), while the Tafel slope of Co_2CrO_4 increases to ~94 mV/dec after 1000 cycles. These results indicate that the charge transfer kinetics CoCr_2O_4 is retained after long durations of OER, while it decays rapidly in Co_2CrO_4 . Overall, both CoCr_2O_4 and Co_2CrO_4 are active in OER, but activated CoCr_2O_4 outperforms Co_2CrO_4 by remaining active for significantly longer durations.’

Figure S4. Tafel slope of (a) CoCr_2O_4 and (b) Co_2CrO_4 nanoparticles that are obtained from LSV data at different stages. For CoCr_2O_4 : The Tafel slope decreases slightly from ~67 to ~65 mV/dec after 100 cycles, indicating improved OER kinetics due to formation of conductive, active (Co,Cr)-based (oxy)hydroxides, which is in line with most Co-based catalyst with a Tafel

plot of around 60 mV/dec. The slope remains relatively stable afterward, consistent with the long-term durability as ~ 72 mV/dec after 2000 cycles was obtained. For Co_2CrO_4 , The Tafel slope remains high and increases slightly with cycling, from ~ 81 mV/dec in pristine to 94 mV/dec after 1000 cycles, reflecting a lack of kinetic improvement and suggesting a degradation during long cycles.

Figure S6. Electrochemical double-layer capacity (C_{dl}) in pristine, after 100 and after 1000 cycles for (a) CoCr_2O_4 and (b) Co_2CrO_4 nanoparticles that are obtained from CV. After 100 CV cycles, the CoCr_2O_4 nanoparticle shows an increase in the electrochemical surface area (ECSA), while its value is slightly decreased after 1000 cycles. For the Co_2CrO_4 nanoparticle, the ECSA also increased before 100 cycles and then decreased after 1000 cycles. Electrochemical impedance spectroscopy (EIS) data of (c) CoCr_2O_4 and (d) Co_2CrO_4 during OER at 1.60 V vs. RHE. The equivalent circuit model for CoCr_2O_4 in series includes an ohmic resistor, which is predominantly defined by the electrolyte resistance between the working and reference electrodes. This setup is considered an electric double-layer capacitor connected in parallel to the faradaic resistance of the oxygen evolution reaction (OER) as indicated in the insets. Despite the OER mechanism involving multiple electron transfer stages, the associated serial resistances are parallel to the same capacitor, resulting in only one semicircle. The EIS simulation results obtained by Zviwer software are presented in Supplementary Table 2

- Whether Cr will be present in the surface active phase for reconstructed CoCr_2O_4 and Co_2CrO_4 spinel catalysts.

Our APT results (Figure 6-7, Table 2, Figure S14 and Figure S21-22) clearly show that ~ 25 at.% Cr is present on the surface and near-surface regions (2-4 nm) in the activated 100-cycle CoCr_2O_4 (as exemplified by Figure 6e-h) and ~ 20 at.% Cr Co_2CrO_4 (e.g., Figure 7e-h). These results highlight the novelty of our study – we provide unprecedented atomic-scale compositional changes, including Cr, Co, O, hydroxide, and K ions, on the surfaces of CoCr-based spinel-type oxides in both the pristine state and after various OER cycles (Table 2).

9. What is the activity of the activated CoCr_2O_4 and Co_2CrO_4 catalyst in the membrane electrode assembly?

We thank the reviewer for raising this application-oriented question. The activity and stability of these CoCr_2O_4 and Co_2CrO_4 spinel-type oxide nanoparticles in the membrane electrode assembly are important, but this study does not aim for device-based applications. Instead, our study aims to improve the fundamental understanding of surface reconstruction and transformation mechanisms on CoCr_2O_4 and Co_2CrO_4 spinel-type oxide nanoparticles during OER. We combined CV, LSV, EIS, ECSA, ICP/MS, XANES, XPS, TEM, APT with in situ Raman spectroscopy and in situ DEMS measurements to interrogate the mechanisms responsible for activation and deactivation of electrocatalysts. The electrochemical activation behaviour is relatively feasible, controllable, and comparable in a three-electrode half-cell experimental setup. However, the activity measured in the membrane electrode assembly may vary due to factors such as specific catalyst loading, ionomer content, mass transport conditions, and membrane properties, which can influence the device setup. Thus, the membrane electrode performance is beyond the scope of our study.

Reviewer #3:

The manuscript titled “Atomic-scale insights into surface reconstruction and transformation in Co-Cr spinel oxides during the oxygen evolution reaction” has been reviewed. Although quite dense in terms of content, the manuscript is fairly well written and follows a logical flow, however there are some concerns around the references. Conclusions are well supported by experimental evidence. The paper presents analysis from a variety of complementary analytical techniques to help provide critical mechanistic understanding regarding the enhanced OER activity of CoCr₂O₄ over Co₂CrO₄ spinel nanoparticles. Specifically, the authors show that the enhanced OER activity of CoCr₂O₄ is induced by Cr dissolution, which in turn promotes the interaction of hydride to induce a surface chemical/structural transformation into the octahedral phase. It is my opinion that before this paper be considered for publication, the authors must explicitly address the critical concerns below:

1. There are some concerns about the references. For one, there are some references that are duplicated (e.g., Ref 31 and Ref 81 are the same). ALSO, there are some instances of what appear to be inappropriately used references. For example, Ref #60 seems not to be appropriate. Please thoroughly review ALL references for being appropriately used and make sure there are no duplicates.

We apologize for this. We have checked the entire manuscript and removed the duplicated references.

Regarding reference 60, the paper is cited for clarifying the deconvolution of the O 1s spectra into four peaks, which contain surface hydroxyl groups. This citation directly supports our statement. Thus, this was kept in the manuscript.

2. ERROR BARS! There is a lack of experimental error reported. Other than APT composition profile results, all other data lacks reported error. On page 5, the authors describe “To ensure experimental reproducibility, each electrochemical measurement was conducted at least three times under the same conditions”. Please incorporate these measurements into single plots with error, rather than rely on show examples of the same measurements in SI (but do keep the data in the SI still though). Small deviations in the LSV scans are reported; what is the error in these measurements to build confidence that deviations are significant to support your conclusions. Provide error descriptions throughout, including tables and reports of Co/Cr ratios and stoichiometries (e.g. Co_{0.55}Cr_{0.45}O). ALL data should have error.

We thank the Reviewer for the constructive comment. We have now combined LSV results from three independent experiments and plotted them as mean ± standard deviation in the new Figure S2 (see below) and amended the text on page 5:

‘To ensure experimental reproducibility, each electrochemical measurement was conducted at least three times under the same conditions; the deviation from each LSV measurements were plotted in Figure S2.’

Figure S2. Linear sweep voltammetry curves after the first, 100th and 2000th cyclic voltammetry cycles with error bars that obtained from the three measurements

In addition, we have added EDX data with error bars and all ratios with error bars (in Table S1 and the text highlighted in red).

- Page 4: provide some explicit examples of what operando and in situ spectroscopy techniques lack in the sentence “Indeed, even operando and in situ spectroscopy techniques^{30,33-37} only provide few aspects of the required surface details”

We have revised the sentence on page 4 as follows :

‘It is nearly impossible to obtain complete information regarding the surface state of the electrocatalyst using only a single technique. Indeed, even operando and in situ spectroscopy techniques^{31,33-35} only provide few aspects of the required surface details, such as surface oxidation state from X-ray photoelectron spectroscopy (XPS) or chemical species from Infrared spectroscopy.’

- Page 9 regarding sentence “The Cr dissolution rates in both systems are highest during the first 100 cycles, after which they drop, possibly due to Cr saturation in the electrolyte or Cr depletion from the materials”. You could estimate this based on the composition within the electrolyte and then rule/confirm if in saturated state. Please do this. And if it is NOT saturated, then what does that say about “Cr depletion from the materials”?...maybe diffusion limited (i.e. as more and more Cr is leached, any more has to dissolve from further within the crystal. OR maybe there is a transient Cr/Co metal hydroxide salt forming at the surface which gets denser/thicker with cycles, lowering Cr dissolution rate? This would be more like a gel-like structure.

Thank you for this insightful comment. Cr dissolves at our OER cycling potential and pH (14) by forming soluble CrO_4^{2-} in the aqueous electrolyte according to the Cr Pourbaix diagram (Corrosion Science, 39(1997)43). The solubility of CrO_4^{2-} in 1 M KOH/ H_2O is most likely similar to that of K_2CrO_4 (640 g/L at 20 °C). Hence, the solubility of Cr is $\sim 171.52\text{g/L}$ at 20 °C in KOH/ H_2O , which is significantly higher than our ICP/MS measured values ($\sim 1.2\text{ mg/L}$). It is thus highly likely that the in situ formation of surface Co-, Cr-based (oxy)hydroxides after 100 CV cycles acts as a barrier to impede rapid Cr leaching. In addition, we fully agree with the Reviewer that the diffusion process plays a critical role (as reported in our work in ACS

Catalysis, 14 (2024), 12704). More Cr needs to diffuse from the interior part of the nanoparticle to the surface regions when the surfaces are gradually depleted of Cr, which also slows down the Cr dissolution rate.

To address this point, we added the following text on page 9:

‘Cr dissolves at the applied potential and pH by forming soluble CrO_4^{2-} in the aqueous electrolyte according to the Cr Pourbaix diagram.⁴⁸ Its solubility in $\text{KOH}/\text{H}_2\text{O}$ is similar to that of K_2CrO_4 (~640 g/L at 20 °C). Thus, the solubility of Cr is ~171.5g/L, which is significantly higher than the dissolved amount measured by ICP/MS (~1.2 mg/L). This suggests that the Cr dissolution after 100 cycles is likely impeded by the in situ-formed surface Co-, Cr-based (oxy)hydroxides that may act as a barrier to inhibit rapid Cr leaching. Additionally, continuous Cr leaching requires its diffusion from the interior of the nanoparticle after surfaces are depleted of Cr in the first 100 cycles,⁴⁹ which might further decrease the Cr dissolution rate.’

5. Page 10: “For Co_2CrO_4 inverse spinel, Co(II) occupies the tetrahedral sites”; the Co(II) should be Cr(II). ALSO since there are MANY instances of Co_2CrO_4 and CoCr_2O_4 throughout, please review throughout for correctness in reference to what is being described/explained (E.g., Page 19, “In comparison to Co_2CrO_4 , the uppermost 4-5 atomic surface layers”, should be CoCr_2O_4)

We thank Reviewer for this comment. The sentence on page 10 ‘For Co_2CrO_4 inverse spinel, Co(II) occupies the tetrahedral sites, and Co(III) and Cr(III) the octahedral sites’ is correct, which has been reported in Journal of Inorganic and Nuclear Chemistry, 33(1971)63).

Cr does not exist in divalent state. On the other hand, we completely understand why this sentence confused the Reviewer, as the spinel structure is defined by divalent cations occupying tetrahedral sites and trivalent cations occupying octahedral sites, $\text{tet}(\text{A}^{\text{II}})_{\text{oct}}[\text{B}^{\text{III}}]_2\text{O}_4$. In contrast, in the inverse spinel structure, the divalent cations occupy octahedral sites and trivalent cations take both tetrahedral and octahedral sites, $\text{tet}(\text{B}^{\text{III}})_{\text{oct}}[\text{A}^{\text{II}}\text{B}^{\text{III}}]\text{O}_4$. Or mixed spinel means divalent and trivalent cations occupy both tetrahedral and octahedral sites.

However, one possible spinel structure is $\text{tet}(\text{A}^{\text{II}})_{\text{oct}}[\text{A}^{\text{III}}\text{B}^{\text{III}}]_2\text{O}_4$, where A(II) occupy the tetrahedral sites, A(III) and B(III) occupy the octahedral sites. When one strictly follows the definition of inverse spinel, in which trivalent cations occupy tetrahedral sites, our sentence would be incorrect. However, this $\text{tet}(\text{A}^{\text{II}})_{\text{oct}}[\text{A}^{\text{III}}\text{B}^{\text{III}}]_2\text{O}_4$ spinel belongs to neither normal spinel nor mixed spinel based on the conventional definition. Therefore, a more rational definition of inverse spinel could be AB_2O_4 , which could be defined by A cations occupying the octahedral sites and B cations occupying both tetrahedral and octahedral sites.

To avoid confusion for readers, we nevertheless updated this sentence on page 10:

‘For Co_2CrO_4 , Co(II) occupies the tetrahedral sites, and Co(III) and Cr(III) the octahedral sites.’

The text on page 22 (previously page 19) was amended to ‘In comparison to CoCr_2O_4 , the uppermost 4-5 atomic surface layers of pristine Co_2CrO_4 ’. We have carefully checked this point across the manuscript and the Supplementary file.

6. There is no description regarding APT based compositional analysis. How did you integrate peak counts, how do you deal with background? How did you deal

background subtraction when Co-59 is flanked by Ni-58 and Ni-60 peaks from the electrodeposited Ni matrix?

We thank the Reviewer for raising this critical point. The APT data analysis was performed in the AP suite 6.3 software. All peaks were manually ranged by using the full-width at nine tenths maximum method. The background was subtracted using software with a built-in global background subtraction function. The background was subtracted from the counts of individual elements.

Our atom probe is LEAP 5000 XR, and experiments were conducted in the laser mode. This yields a high mass-resolving power with a full-width half maximum of ~ 650 . The peak at 59 Da (Co) is sharp, which overlaps with the shoulder of the peak at 58 Da or produces a peak shoulder that overlaps with the peak at 59 Da, as shown in Figure R4, a zoomed-in view of Figure S12.

Figure R4. Zoomed-in view of mass spectra shown in Figure S12 (100-cycle CoCr_2O_4).

We have incorporated additional text on page 35 in Method Section:

'The mass spectra peaks were manually ranged by using the full-width at nine tenths maximum method. The background was subtracted using the software built-in global background subtraction function. The background was also subtracted from counts of individual elements for elemental composition analysis.'

- Regarding conclusion that K and hydroxide ions are present INSIDE the CoCr_2O_4 NPs, how can you rule out aberrations leading to trajectory overlap misleading to interfaces that are artificially wide/mixed. ADDITIONALLY, related to this, how can you say with certainty that hydroxide is INSIDE, when there seems to be not many ions even counted (based on 3D distributions in Figures), meaning how can you distinguish/say that those ions shown to be INSIDE the particle are hydroxide and not background noise (WHAT IS THE S/N ratio?). One way you can do this is to isolate the INSIDE and show/analyze those mass spectra. Please do this.

We appreciate Reviewer's insightful comment. We have compared the mass spectra on the surface top 2 nm and the interior of 100-cycle CoCr_2O_4 in the new Figure S17a-b. The signal-to-noise ratio of the K and OD peaks is ≥ 10 in Figures 17a and 17 b. Figure S17b clearly shows that the interior region of the nanoparticle contains K (at 39 Da), hydroxide ions (OD at 18 Da), along with D_2O (at 20 Da), although their amounts are significantly less than those in the near-surface regions shown in Figure S17a.

Also, it is rational to speculate that the presence of K and hydroxide ions (OD) originates from trajectory aberration from the matrix (Ni). To clarify this, the mass spectra in the nearby Ni matrix were also listed in Figure S17c. We can see that the nearby Ni matrix does not contain any K (at 39 Da) or OD (at 18 Da) peaks, Figure S17c. Thus, such speculation can be ruled out. Our APT data reveal that the K and hydroxide ions are genuinely intercalated ions on the near-surfaces and in the interior of the CoCr_2O_4 nanoparticles after 100 CV cycles.

Figure S17. Localized mass spectrum extracted from (a) surface of the nanoparticle, (b) inner of the nanoparticle, and (c) Ni substrate in no nanoparticle regions of CoCr_2O_4 after 100 cycles within the range of 0-40 Da, all the regions marked in the 2D Cr map of 100-cycle CoCr_2O_4 nanoparticles on the

top. Figure S17b clearly shows the interior region of the nanoparticles contains some amounts of K (at 39 Da), hydroxide ions (OD at 18 Da) and water molecules (D_2O at 20 Da), albeit much less than those in the surface/near-surface regions (Figure S17a). Also, the Ni matrix contains nearly no K and OD (Figure S17c), which excludes the possibility the presence of K and hydroxide ions (OD) originate from their trajectory aberration (APT artefact¹) from the matrix materials (Ni).

To clarify this point, we added the text on page 21 and new Figure S17 in supplementary file:

'Specifically, K and hydroxide ions are densely distributed in the upper 2-4 nm surface region of $CoCr_2O_4$ (Figure 6f, with more data in Figures S14b and S16a), while some K and hydroxide ions are even present inside the $CoCr_2O_4$ nanoparticles (see detailed mass spectra in Figure S17).'

8. Additionally show APT mass spectra, at least in SI section

We fully agree with the reviewer's comment and have included the representative mass spectra in Figure S12 and added the text on page 17.

'The spinel oxides were detected in the form of O ions and Co- and Cr-containing complex molecular ions (see mass spectra in Figure S12), represented as CoO_x (in blue) and CrO_y (in red), respectively, in Figures 6 and 7.'

Figure S12. Full mass spectrum collected from a representative $CoCr_2O_4$ nanoparticle after OER.

9. Regarding discussions around particle vertices showing enhanced Cr dissolution, please support this notion based on results from other material systems. Example: <https://doi.org/10.1016/j.seppur.2020.118014>; <https://doi.org/10.1039/D4EN01004C>

We appreciate the reviewer's suggestion. We fully support this notion. We have incorporated these two references and added few sentences to support the notion on page 21:

'Our observation of enhanced Cr dissolution at $CoCr_2O_4$ nanoparticle vertices is consistent with findings in other nanoparticle systems, where regions of high surface curvature exhibit increased

dissolution rates.^{79,80} The nanoparticles are thought to exhibit site- or facet-dependent dissolution behaviours due to the difference in surface energies, local potentials or adsorbed ligands.^{79,80}

10. Regarding the APT measurements, please discuss possible effects that the electrodeposition of the Ni matrix has on the surface composition. K is also present in the electrodeposition process.

We apologize that the method section was not clear enough. The process of APT specimen preparation of post-OER nanoparticles is as follows. We performed CV measurements on nanoparticles that were drop-cast on a glassy carbon electrode under OER conditions in 1 M KOH. Afterward, the glassy carbon electrode was removed from the KOH aqueous electrolyte and immersed in another electrolyte solution containing 1 g of nickel chloride, 6.0 g of nickel sulphate, and 1 g of boric acid in 20 mL of deionized water for Ni electrodeposition. We then used FIB lift-out to prepare APT specimens from the bulk Ni samples where nanoparticles are embedded. K is not present in the Ni electrodeposition process, which is also confirmed by the mass spectra in the Ni matrix (see Figure S17c in Reviewer Question 7).

To clarify this point, we have amended the text in Method Section on page 35:

‘For APT specimen preparation, a bulk sample embedding the nanoparticles was fabricated by electrodeposition of Ni. Specifically, bulk nanoparticle film on glassy carbon were prepared by drop cast and dried overnight. Electrochemical CV experiments used 1 M KOD (original content 40 wt% in D₂O, Sigma-Aldrich) in D₂O electrolyte (Sigma-Aldrich) as electrolyte at a scan rate of 50 mV/s from 1 V to 1.65 V vs. RHE for 100 and 1000 cycles, respectively. The glassy carbon electrode covered by nanoparticles was subsequently immersed in another electrolyte mixed with 1 g nickel chloride, 6.0 g nickel sulphate, and 1 g boric acid in 20 ml DI water at a constant voltage of -1.0 V for 300 s for Ni electrodeposition.³⁸ Afterward, needle-shape APT specimens were prepared from the nanoparticle-embedded Ni bulk sample via focused ion beam (FIB) lift out procedures in FEI Helios G4 CX.³⁰

Regarding the possible effects of Ni deposition on the surface composition of nanoparticles, the ‘thickness’ of the surface reconstructed layer, particularly elemental distribution, is affected due to the trajectory aberration, as mentioned by Reviewer’s Question 7. On the other hand, this trajectory aberration artefact affects all the datasets similarly. Thus, the trend or difference in the composition and elemental distribution on the surfaces or the bulk of the nanoparticles in the pristine state, after 100 and 1000 CV cycles, is thought to be genuinely induced during electrochemical measurements. Additionally, such artifacts would impact other nanoparticle systems published by my group. For example, we did not observe such high K and OD counts in CoFe-based and CoMn-based spinel oxide nanoparticles (Nature Communication, 13 (2022)279), Advanced Energy Materials, 15(2024)2403096). Thus, the pronounced intercalation of ions leads us to propose intercalated-assisted alpha-type hydroxide to gemma-type oxyhydroxide surface transformation mechanisms for CoCr-based spinel oxides.

11. What is the role of the electrolyte cation K having on the OER? Electrolyte counter ions like K can lower overpotentials by stabilizing reaction intermediates. You see it on the surface only of the Co₂CrO₄ particles but inside the CoCr₂O₄ particles. Seems like K may be playing an active role (not passive) that is being overlooked. See: <https://pubs.acs.org/doi/10.1021/acs.chemmater.7b0051760>

We appreciate the Reviewer for mentioning the effect of K. Essentially, according to early Bode’s study (Electrochimica Acta, 11(1966)1079) on the Ni systems, the transformation from

alpha hydroxide to hydrated gamma oxyhydroxide requires the intercalation of hydroxide and K ions to balance the charges according to the equation below.

Thus, K⁺ serves as a charge-balancing ion. Their presence can stabilize the negatively charged hydroxide-rich environment, stabilizing the structure of gemma-type oxyhydroxide. Our APT data (Table 2 and Figure 16a) show that a significant amount of K⁺ is present on the surfaces, and some is even present in the interior of 100-cycle CoCr₂O₄ nanoparticles. Such pronounced amounts of K and hydroxides were not observed in our CoFe- and CoMn-based spinels (Nature Communication, 13 (2022)279), Advanced Energy Materials, 15(2024)2403096). Therefore, we believe that this indicates the surface transformation of CoCr₂O₄ is significantly different from that occurred on the surfaces of CoFe- and CoMn-based spinels where beta oxyhydroxide is formed after OER. By relating to the TEM/SAED results (Figure S10b), the surfaces of 100-cycle CoCr₂O₄ most likely undergo the transition from α-type Co-based hydroxide to β-type oxyhydroxide. The incorporation and intercalation of K⁺ and OH⁻ balances the charge of the type oxyhydroxide.

In addition, a recent study (Angew. Chem. Int. Ed. 2023, 135, e202313886) also speculated that the presence of K⁺ might stabilize or promote the formation of the higher oxidation state of Co-containing active species. Another study on alkali-ion effects (e.g., J. Phys. Chem. C 2020, 124, 27, 14734–14741) suggests that larger cations like K⁺ modulate the local electric field and proton transfer dynamics, and can stabilize the intermediates more effectively, thereby indirectly affecting the OER activity.

Based on this, we modified the text on page 21:

'By relating to the TEM/SAED results (Figure S10b), the high amount of K and hydroxide ions is most likely associated with α-type Co-based hydroxide ↔ γ-type oxyhydroxide transition, which is in line with the intensified Al/Cl redox peak in Figure 1c. The intercalated hydroxide and K ions balances the charges and stabilises the structure of γ-type oxyhydroxide, as observed during Ni α-hydroxide ↔ γ-oxyhydroxide transition.⁸⁰ Notably, the surface reconstructed region revealed by APT is 2-4 nm, which is thicker than that revealed by TEM (Figure 4d). This could be caused by the high structural reversibility of the surface reconstructed layer. On the other hand, we cannot eliminate the possibility that such surface layer might partially undergo the electron beam-induced recrystallization, especially at a high magnification in TEM.'

12. Be consistent with color coordination: E.g., K is shown in blue in all figures, except in Fig 7 where it is green

We thank the reviewer for noticing this and helping us to improve the manuscript. We have changed the colour of K⁺ in Figure 8 (Figure 7 in the original manuscript) to dark blue (see below).

Figure 8. Schematic diagram showing surface reconstruction and transformation of CoCr₂O₄ and Co₂CrO₄ during OER. a) Cr dissolves substantially across nearly the entire CoCr₂O₄ nanoparticles, leading the formation of Cr and O vacancies that promote the hydroxide ion incorporation; b) the gradual hydroxide ion incorporation potentially facilitates reversible (Co^{II}, Cr^{III})(OH)₂ ↔ (Co^{III}, Cr^{III})OOH transformation, giving rise to high activity and stability of CoCr₂O₄; as OER proceeds, (c) the (oxy)hydroxides are gradually exfoliated from CoCr₂O₄ surfaces. In contrast, d) pristine Co₂CrO₄ surface is covered by rock-salt structured (Co,Cr)O which does not assist transformation of Co-based oxyhydroxide. Instead, e) amorphous Cr(OH)₃ layer is formed on the Co₂CrO₄ surfaces, serving as active species for OER. Such self-limiting amorphous layer is gradually depleted from Co₂CrO₄ surfaces, decreasing the activity.

Response to Reviewer's comments on Nature Communications Manuscript NCOMMS-25-05420A

Reviewer #1:

The current revised version of this manuscript is appropriate. Therefore, I would like to recommend the publication of this article.

Thank you for the constructive comments.

Reviewer #2:

The authors have effectively strengthened the work and addressed the raised concerns through additional experiments. This convincingly demonstrates how the structure and composition of CoCr₂O₄ and Co₂CrO₄ spinel nanoparticles affect OER activity and stability. However, some responses still require further clarification. My specific comments are as follows:

1. During the OER process, both CoCr₂O₄ and Co₂CrO₄ catalysts form Cr-based (oxy)hydroxide layers. Why does the CoCr₂O₄ catalyst exhibit excellent stability, while the Co₂CrO₄ catalyst has poorer stability? Could it be that no highly reversible transformation of (Co_{Td}^{II}, Cr)(OH)₂ ↔ (Co_{Oct}^{III}, Cr)OOH occurs on the surface of the Co₂CrO₄ catalyst? What evidence supports this conclusion?

Our study demonstrates two distinct activation mechanisms on CoCr₂O₄ and Co₂CrO₄ during OER cycling. Specifically, the activation of **CoCr₂O₄** is induced by a steady and substantial Cr dissolution that facilitates bulk incorporation and intercalation of hydroxide ions, coupled with the highly reversible (Co_{Td}^{II}, Cr)(OH)₂ ↔ (Co_{Oct}^{III}, Cr)OOH transformation, which enhances OER activity and stability. In comparison, a ~2 nm thick amorphous self-limiting Cr-based (oxy)hydroxide forms on **Co₂CrO₄** upon cycling, contributing to OER activity. This is evidenced by our new Raman spectroscopy measurement in Figure 4b (which was requested by the Reviewer). Such active amorphous Cr-based (oxy)hydroxides formed on Co₂CrO₄ are self-limiting. Upon OER cycling, this amorphous layer gradually disappears due to its high solubility, leading to a decline in Co₂CrO₄ OER activity (as evidenced by our TEM images in Figure 5i-j after 100 and 1000 cycles). Thus, Co₂CrO₄ cannot retain the activity for longer OER durations due to the gradual dissolution of the Cr-based (oxy)hydroxide layer.

The two distinct activation mechanisms are illustrated in the schematic diagram in Figure 8 and further clarified in the abstract. Additionally, the last paragraph on page 30 was revised as follows:

'In contrast to CoCr₂O₄, the active amorphous Cr-based (oxy)hydroxides formed on Co₂CrO₄ are self-limiting. Upon OER cycling, this amorphous layer gradually disappears due to its high solubility^{86,87}, leading to a decline in Co₂CrO₄ OER activity (Figure 8f). Thus, Co₂CrO₄ cannot retain the activity for longer OER durations due to gradual dissolution of Cr-based (oxy)hydroxide layer.'

‘ CoCr_2O_4 is found to undergo an activation process and subsequently retains high OER activity for extended durations. The activation of CoCr_2O_4 is induced by a steady and substantial Cr dissolution that facilitates bulk incorporation and intercalation of hydroxide ions, coupled with the highly reversible $(\text{Co}_{\text{Td}}^{\text{II}}, \text{Cr})(\text{OH})_2 \leftrightarrow (\text{Co}_{\text{Oct}}^{\text{III}}, \text{Cr})\text{OOH}$ transformation, which enhances OER activity and stability. In comparison, a ~ 2 nm thick amorphous self-limiting Cr-based (oxy)hydroxide forms on Co_2CrO_4 upon cycling, contributing to OER activity. As OER proceeds, the Cr-based (oxy)hydroxide layers on Co_2CrO_4 are depleted from the surfaces, leading to deteriorating activity.’

Figure 8. Schematic diagram showing surface reconstruction and transformation of CoCr_2O_4 and Co_2CrO_4 during OER. a) Cr dissolves substantially across nearly the entire CoCr_2O_4 nanoparticles, leading the formation of Cr and O vacancies that promote the hydroxide ion incorporation; b) the gradual hydroxide ion incorporation potentially facilitates reversible $(\text{Co}^{\text{II}}, \text{Cr}^{\text{III}})(\text{OH})_2 \leftrightarrow (\text{Co}^{\text{III}}, \text{Cr}^{\text{III}})\text{OOH}$ transformation, giving rise to high activity and stability of CoCr_2O_4 ; as OER proceeds, (c) the (oxy)hydroxides are gradually exfoliated from CoCr_2O_4 surfaces. In contrast, d) pristine Co_2CrO_4 surface is covered by rock-salt structured $(\text{Co,Cr})\text{O}$ which does *not* assist transformation of Co-based oxyhydroxide. Instead, e) amorphous $\text{Cr}(\text{OH})_3$ layer is formed on the Co_2CrO_4 surfaces, serving as active species for OER. Such self-limiting amorphous layer is gradually depleted from Co_2CrO_4 surfaces, decreasing the activity.

- The author conducted OER under alkaline conditions. Why was there no mention of the standard catalysts under alkaline conditions in the introduction section, but instead, the commercial catalysts under acidic OER conditions were elaborated?

Thanks for this comment. We have revised the text in the introduction on page 3:

'Ni-based electrocatalysts show promising OER stability in alkaline media, but the activity is limited.⁶ To meet global energy demands, it is thus essential to reduce the cost of electrolyzers and develop more affordable, sustainable and efficient OER electrocatalysts.'

- The author states that "As OER proceeds, the (oxy)hydroxide layers on Co_2CrO_4 are depleted from the surfaces, leading to deteriorating activity.". How did the author prove that the (oxy)hydroxide layers formed on the surface of CoCr_2O_4 disappeared during the OER? The author should provide high-magnification TEM images during the process of activity decrease.

The zoomed-in views in Figure 5i-j and Figure S24 are high-resolution TEM images at high magnification (see Figure R1 summarizing three examples of Co_2CrO_4 nanoparticles after 100 and 1000 CV cycles). We can see that a 1-2 nm thick amorphous layer forms on the Co_2CrO_4 nanoparticles after 100 cycles (as delineated by the red dashed lines in zoomed-in views of Figure R1a-c). After 1000 cycles, such amorphous layer almost completely disappears (Figure R1d-f). These results suggest that the amorphous layer undergoes a gradual dissolution process, which explains the drop in activity after prolonged OER cycling. To reveal the surface amorphous layer, we have drawn red dashed lines on the surfaces in the new Figure 5.

Figure R1 High-resolution TEM images and zoomed-in views of Co_2CrO_4 nanoparticle surfaces after (a-c) 100 and (d-f) 1000 CV cycles under the OER conditions.

4. XANES data needs to be fitted in order to understand the changes in coordination number and bond length for CoCr_2O_4 and Co_2CrO_4 catalysts. CoOOH should also be selected as the reference sample.

The Co and Cr XANES data were fitted to reveal the coordination number and bond length, as shown in Figure R2, and detailed fitting parameters are shown in Tables R1 and R2 (see below). The Fourier transform (FT) of the EXAFS was calculated between 3.0 to 10.3 \AA^{-1} above the K edge ($E_0 = 6989$ eV for Cr and $E_0 = 7709$ eV for Co). Each scan in EXAFS analyses was meticulously calibrated with the reference foil to correct for beam energy fluctuations during the experiment. Background removal and normalization were achieved using the Athena program. The normalized EXAFS data were subsequently processed in the Artemis program, applying a k^3 -weighting and a Hanning window within the k range of 3-11 \AA^{-1} and an R range of 1.0-3.2 \AA . The optimization of the fits was done by minimizing the sum of squared deviations between the experimental and fitted values.

According to the previous study (J. Am. Chem. Soc. 2016, 138, 36–39), the peak at about 1.5 \AA in Figure R1a, c belongs to single scattering paths of the Co cation to the nearest neighbouring oxygen anion (Co–O) and two separate peaks at ~2.5 and 3.1 \AA correspond to the radial distances of Co_{Oh} (Co in octahedral site) and Co_{Td} (Co in tetrahedral site) to their neighbour metal cations, respectively. The two peaks in Cr spectra in Figure R1b,d show Cr–O at ~1.5 \AA and Cr_{Oh} at ~2.5 \AA .

We can see from Figure R1a a slight reduction of Co_{Td} peak intensity at ~3.1 \AA in CoCr_2O_4 sample and an increase in Co_{Oh} at ~2.5 \AA . After 1000 cycles, the Co_{Td} peak diminishes completely with a pronounced Co_{Oh} peak (Figure R1a). Additionally, a higher O coordination number (N) of the Co–O, from 3.69 in the pristine state to 3.78 after 100 cycles and 5.89 after 1000 cycles (Table R1). The bond length (R) of Co–O is reduced from 2 to 1.93 \AA after 100 cycles and 1.88 \AA after 1000 cycles, suggesting the possible formation of a more disordered structure on the surface. In addition, a slight change in the coordination number and bond length of Cr was observed (Figure R2b). These results suggest that Co tetrahedral sites undergo pronounced changes, most likely contributing to the formation of CoOOH (consistent with our XPS and XANES data). In comparison, the Co–O coordination number and bond length change in Co_2CrO_4 marginally (Table R1 and R2).

Figure R2. Fourier transforms of the (a) Co K-edge and (b) Cr K-edge for CoCr₂O₄ in the pristine state (black curves) and after 100 (red curves) and 1000 (blue curves) CV cycles, and (c) Co K-edge and (d) Cr K-edge for Co₂CrO₄ in the pristine state (black curve), after 100 CV (red curve), and after 1000 (blue curve) CV cycles.

Table R1. Structural fit parameters of Co K-edge EXAFS.

Sample ^a	Path	N ^b	R / Å ^c	$\sigma^2 / 10^{-3} \text{ \AA}^d$	R-factor
CoCr ₂ O ₄ in pristine	Co ₁ -O ₁	3.69	2.0	4.3	2.4 %
	Co ₁ -M ₁	11.57	3.35	6.8	
	Co ₁ -O ₂	11.67	3.43	12.1	
	Co ₁ -M ₂	3.72	3.65	11.2	
CoCr ₂ O ₄ after 100 cycles	Co ₁ -O ₁	3.78	1.93	4.8	3.8 %
	Co ₁ -M ₁	11.81	3.23	8.1	
	Co ₁ -O ₂	11.81	3.52	11.2	
	Co ₁ -M ₂	3.72	3.53	11.5	
	Co ₂ -O ₂	5.82	1.93	14.3	
	Co ₂ -M ₂	5.82	2.72	17.7	
CoCr ₂ O ₄ after 1000 cycles	Co ₂ -O ₂	5.89	1.88	11.2	4.4 %
	Co ₂ -M ₂	5.89	2.73	15.2	
	Co ₂ -M ₁	5.89	3.33	17.2	
Co ₂ CrO ₄ in pristine	Co ₁ -O ₁	3.74	1.98	6.7	1.7 %
	Co ₁ -M ₁	11.81	3.35	3.5	
	Co ₁ -O ₂	11.81	3.47	10.5	

	Co ₁ -M ₂	3.74	3.57	13.4	
	Co ₂ -O ₂	5.84	1.98	11.2	
	Co ₂ -M ₂	5.84	2.85	15.2	
	Co ₂ -M ₁	5.84	3.43	17.2	
Co ₂ CrO ₄ after 100 cycles	Co ₁ -O ₁	3.76	1.96	7.3	2.4 %
	Co ₁ -M ₁	11.83	3.33	4.9	
	Co ₁ -O ₂	11.83	3.44	11.2	
	Co ₁ -M ₂	3.73	3.54	16.3	
	Co ₂ -O ₂	5.87	1.96	17.6	
	Co ₂ -M ₂	5.87	2.83	16.2	
	Co ₂ -M ₁	5.87	3.42	19.2	
Co ₂ CrO ₄ after 1000 cycles	Co ₁ -O ₁	3.76	1.95	6.2	2.3 %
	Co ₁ -M ₁	11.87	3.34	3.3	
	Co ₁ -O ₂	11.87	3.44	11.4	
	Co ₁ -M ₂	3.73	3.56	12.5	
	Co ₂ -O ₂	5.87	1.95	13.8	
	Co ₂ -M ₂	5.87	2.80	19.4	
	Co ₂ -M ₁	5.87	3.40	16.2	

Notes: *a* S_0 was fixed as 1.0. Data ranges: $3.0 < k < 11.0 \text{ \AA}^{-1}$, $1.0 < R < 3.2 \text{ \AA}$. *b* N is the coordination number. *c* R is the distance between absorber and backscatter atoms. *d* σ^2 is the Debye-Waller factor. The R-factor is the residual factor. In the Path column, Co₁, O₁, and M₁ represent tetrahedral sites, Co₂, O₂, and M₂ represent octahedral sites, and M can be Cr or Co.

Table R2. Structural fit parameters of Cr K-edge EXAFS.

Sample ^a	Path	N^b	$R / \text{\AA}^c$	$\sigma^2 / 10^{-3} \text{ \AA}^d$	R-factor
CoCr ₂ O ₄ in pristine	Cr ₂ -O ₂	5.84	1.98	3.4	1.3 %
	Cr ₂ -M ₂	5.84	2.93	8.5	
	Cr ₂ -M ₁	5.84	3.39	9.1	
CoCr ₂ O ₄ after 100 cycles	Cr ₂ -O ₂	5.86	1.96	3.2	3.2 %
	Cr ₂ -M ₂	5.86	2.92	8.5	
	Cr ₂ -M ₁	5.86	3.34	8.8	
CoCr ₂ O ₄ after 1000 cycles	Cr ₂ -O ₂	5.85	1.96	4.2	3.2 %
	Cr ₂ -M ₂	5.85	2.91	7.4	
	Cr ₂ -M ₁	5.85	3.39	8.2	
Co ₂ CrO ₄ in pristine	Cr ₂ -O ₂	5.66	1.98	5.2	2.1 %
	Cr ₂ -M ₂	5.66	2.93	6.7	
	Cr ₂ -M ₁	5.66	3.42	11.1	
Co ₂ CrO ₄ after 100 cycles	Cr ₂ -O ₂	5.68	1.97	4.2	2.4 %
	Cr ₂ -M ₂	5.68	2.92	5.7	
	Cr ₂ -M ₁	5.68	3.41	11.3	
Co ₂ CrO ₄ after 1000 cycles	Cr ₂ -O ₂	5.69	1.97	3.8	2.7 %
	Cr ₂ -M ₂	5.69	2.90	4.7	
	Cr ₂ -M ₁	5.69	3.40	12.3	

Notes: *a* S_0 was fixed as 1.0. Data ranges: $3.0 < k < 11.0 \text{ \AA}^{-1}$, $1.0 < R < 3.5 \text{ \AA}$. *b* N is the coordination number. *c* R is the distance between absorber and backscatter atoms. *d* σ^2 is the Debye-Waller factor. The R-factor is the residual factor. In the Path column, M₁ represents tetrahedral sites, Cr₂, O₂, and M₂ represent octahedral sites, and M can be Cr or Co.

Despite these analyses, we emphasize that the signal from the electrode was weak, and the allocated beamtime did not yield spectra of sufficient data quality. Additionally, the amount of post-OER nanoparticles decreased, especially after 1000 cycles, further reducing the spectra intensity. Therefore, we refrain from including the results in the manuscript and supplementary file.

Although EXAFS fitting was not explicitly included in this manuscript, we have provided qualitative insights based on our Co and Cr K-edge XANES and pre-edge feature analysis in Figure 2. We are confident in our pre-edge analysis, which is extremely sensitive to changes in the first coordination sphere of the absorbing metal ion, as amply discussed in numerous XAS textbooks and review articles, like [XAFS for Everyone, by Scott Calvin, CRC Press, 2013]. These results demonstrate the changes in the Co sites of CoCr_2O_4 before and after OER.

5. Why is there a significant shift in the O 1s XPS peak position in Figure 3c? The author should reveal the reason.

We apologize that there was an error in the legend of O 1s XPS peaks in the original Figure 3c. We have revised the legend colour of O1 and O2 in new Figure 3, which shows no O 1s peak shift. The new Figure 3 has been added in the manuscript.

Figure 3. Surface oxidation state and oxygen species of CoCr_2O_4 and Co_2CrO_4 before and after OER. XPS of a) Co 2p, b) Cr 2p, and c) O 1s levels of CoCr_2O_4 ; and d) Co 2p, e) Cr 2p, and f) O 1s levels of Co_2CrO_4 in the pristine state and after 1000 cycles. A standard Shirley background was applied to all spectra. Models of Co(0), CoO, Co(OH)₂, CoOOH, Co₃O₄ were considered for the fitting of the Co 2p_{3/2}. The detailed information regarding the XPS measurements and peak fitting process is presented and discussed in Supplementary Note 2.

6. XPS spectra have confirmed that CoOOH is produced during the catalytic process. However, why did no CoOOH peak appear in the in-situ Raman test?

We appreciate the Reviewer's insightful question. The E_g band of the CoOOH overlaps with one of the F_{2g} peaks of the Co spinel oxide at $\sim 510\text{ cm}^{-1}$ (as seen from Figure S10 and Figure 4). This makes it difficult to trace the in situ formation of CoOOH on the Co spinel oxide using in situ Raman spectroscopy. On the other hand, we observed that the A_{1g} band of CoCr_2O_4 is slightly blue-shifted from 688 cm^{-1} to 690 cm^{-1} upon application of a potential bias (Figure 4c). This blue shift in the A_{1g} band indicates lattice contraction and charge redistribution due to the gradual formation of the Co-based oxyhydroxide phase (according to Nature Communications, 12 (2021), 3036).

Additionally, we stress the importance of our multimodal approach as different techniques provide complementary experimental details. The absence of a CoOOH peak in the in-situ Raman spectra can also be attributed primarily to the difference in surface sensitivity between XPS and Raman spectroscopy. XPS is an inherently surface-sensitive technique, probing the top few nanometers of the material, making it highly effective at detecting thin surface layers such as the CoOOH formed from OER. In contrast, Raman laser has a significantly greater penetration depth—typically on the order of micrometers—especially when applied to powdered samples. This means that Raman signals predominantly originate from the bulk material beneath the surface rather than from the ultra-thin surface layers. Furthermore, the relatively thick deposition of catalyst powder for the in-situ Raman measurement further hinders the detection of surface-specific features. Importantly, the CoOOH species formed from OER exists only as a few atomic layers at the catalyst surface (see the HRTEM image in Figure 5d and corresponding SAED pattern in Figure S11). Given these ultra-thin surface layers and the limited surface sensitivity of Raman under our experimental conditions, it is reasonable that the CoOOH signal was not observable in the in-situ Raman spectra.

To clarify this point, we added the text on page 15:

'Notably, the E_g band of CoOOH overlaps with F_{2g} of Co spinel oxide phase at $\sim 510\text{ cm}^{-1}$, which makes it challenging to trace in situ formation of CoOOH on Co-based spinels by Raman spectroscopy. However, the A_{1g} band of CoCr_2O_4 is slightly blue-shifted from 688 cm^{-1} to 690 cm^{-1} upon applied potential bias (Figure 4c). This blue-shift in the A_{1g} band indicates lattice contraction and charge redistribution due to the gradual formation of the Co-based oxyhydroxide phase.⁷²'

7. During the catalytic process for OER, CoOOH and CrOOH were in situ generated. What is the structural relationship between CoOOH and CrOOH on the surface of the restructured CoCr_2O_4 and Co_2CrO_4 ? CoOOH is the catalytically active phase. Then, does Cr exist in CoOOH? Atomic-level EDX/EELS analysis of the amorphous layer on the CoCr_2O_4 and Co_2CrO_4 catalyst surface is required.

The in situ formed (oxy)hydroxides on CoCr_2O_4 and Co_2CrO_4 after 100 cycles are amorphous (Figure R3a-b, which is Figure 5d, 5i in the manuscript). This is impossible for us to elucidate the growth orientation relationship of the amorphous (oxy)hydroxide layers on CoCr_2O_4 and Co_2CrO_4 in this study. Such an amorphous layer cannot be resolved by scanning transmission electron microscopy (STEM) imaging mode, as indicated by our high-angle annular dark-field/STEM images in Figure R3c-d. Thus, it is nearly impossible to perform atomic-level STEM/EELS imaging as the (oxy)hydroxide layers have lost the crystalline atomistic structure after OER cycling.

Figure R3. Structural and elemental changes CoCr₂O₄ and Co₂CrO₄ nanoparticles after 100 CV cycles. High-resolution TEM images of (a) CoCr₂O₄ and (b) Co₂CrO₄ with zoomed-in surface regions, taken from the yellow cubic regions, are shown on the right. High-resolution HAADF-STEM images of (c) CoCr₂O₄ and (d) Co₂CrO₄. 3D atom maps of (e) CoCr₂O₄ and (f) Co₂CrO₄ nanoparticles. 2D Cr compositional profiles of (g) CoCr₂O₄ and (h) Co₂CrO₄

nanoparticles overlaid with atom maps showing only hydroxide and K ions. 1D compositional profiles of (i) CoCr_2O_4 and (j) Co_2CrO_4 plotted along the vertices obtained by placing analysis cylinders with a diameter of 4 nm along the white arrow in the corresponding atomic elemental distribution map, while 1D concentration profiles of (k) CoCr_2O_4 and (l) Co_2CrO_4 are plotted along the flat surfaces by placing analysis cylinders with a black arrow in atomic elemental distribution map.

In addition, we stress here that APT can provide three-dimensional compositional details and elemental distribution with sub-nanometre spatial resolution, which is well suited to resolve the surface compositional details for catalyst nanoparticles; our APT data with extended analysis has already show the Cr concentration on the nanoparticle surfaces, as revealed in Table 2, Figure 6-7, Figure S15, S17, S21-23. In comparison, EDX and EELS, as these maps provide ambiguous surface compositional details of nanoparticles, since the inherently few-nanometre scale of electrocatalyst surfaces renders them difficult to analyse in a 2D projected image or analysis. Our atom maps in Figure R3e-f show only cross-sectional areas of three-dimensional catalyst nanoparticles. Based on these atom maps, one can perform 2D compositional analysis, as revealed by the 2D Cr compositional maps (Figure R3g-h), showing that the Cr (with blue/yellow colours) is still present on the top 1 nm surfaces of 100-cycle CoCr_2O_4 and Co_2CrO_4 . More importantly, APT data can show the surface composition of catalyst nanoparticles with complex morphology. In this study, we reveal the surface composition difference in the vertices and flat surface regions of the nanoparticles after OER, as shown by Figure R3i-l. In both surface regions, Cr is still present on the surfaces of 100-cycle CoCr_2O_4 and Co_2CrO_4 , albeit with low concentration due to dissolution (as summarized in Table 2).

8. In general, the original structure of pre-catalyst has an effect on the surface reconstruction degree (J. Am. Chem. Soc. 2020, 142, 12087–12095; J. Am. Chem. Soc. 2020, 142, 12087–12095; J. Am. Chem. Soc. 2024, 146, 11887–11896). The surface reconstruction degree of CoCr_2O_4 and Co_2CrO_4 catalysts should be compared. Is there a certain relationship between the original structure, reconstruction degree, and catalytic performance of CoCr_2O_4 and Co_2CrO_4 catalysts?

Revealing two distinct activation mechanisms on CoCr_2O_4 and Co_2CrO_4 is the key finding of this study. We have dedicated our efforts to establishing a multimodal approach, including electrochemical measurements, ICP/MS, XAS, XPS, Raman spectroscopy, TEM, STEM, and APT, to reveal surface changes and thereby carefully elucidate the mechanisms of Co-Cr-based spinel oxides. Specifically, CoCr_2O_4 undergoes highly reversible intercalation-assisted transition from alpha-type $(\text{Co}^{\text{II}}, \text{Cr}^{\text{III}})(\text{OH})_2$ to gamma-type $(\text{Co}^{\text{III}}, \text{Cr}^{\text{III}})\text{OOH}$, which contributes its high activity and stability. In comparison to CoCr_2O_4 , a ~2 nm thick amorphous self-limiting Cr-based (oxy)hydroxide forms on Co_2CrO_4 upon cycling, contributing to OER activity. CoCr_2O_4 outperforms Co_2CrO_4 at the onset of OER and after OER cycling. It is rational to conclude that the activated CoCr_2O_4 is more active and stable than Co_2CrO_4 since the highly reversible intercalation-assisted transition from alpha-type $(\text{Co}^{\text{II}}, \text{Cr}^{\text{III}})(\text{OH})_2$ to gamma-type $(\text{Co}^{\text{III}}, \text{Cr}^{\text{III}})\text{OOH}$ is more effective than the self-limiting Cr-based (oxy)hydroxide toward OER. We support the Reviewer's notion regarding the importance of pristine spinel structure and have added the text on page 26 and cited the references:

'Our study reveals that Cr at varying Co/Cr ratios induces the formation of different active species through various elementary processes during OER cycling, demonstrating that different spinel oxides facilitate distinct surface transformation mechanisms.^{88,89} (see schematic diagram in Figure 8).'

9. Is it the pure LOM pathway for the OER process for the CoCr₂O₄ and Co₂CrO₄ catalysts? However, the evidence is still insufficient. Here are many results that suggest that the LOM and AEM pathway are co-existent in the OER process. Did the author test the existence of AEM pathway? More reaction intermediates should be monitored to prove the lattice oxygen mechanism for activated CoCr₂O₄ and Co₂CrO₄ spinel catalysts using in situ Fourier transform infrared.

Firstly, we established our multimodal approach by combining electrochemical measurements, ICP/MS, XAS, XPS, Raman spectroscopy, TEM, STEM, and APT to elucidate the surface transformation mechanisms of Co-based spinels during OER cycling. Investigating if LOM, AEM, or mixed mode occurs on CoCr-based spinels is not the focus of this study. To address Reviewer's question, we have additionally performed DEMS measurements in Figure S27-29 and Supplementary Note 4. Specifically, 100-cycle electrocatalytic activation pretreatments were performed inside DEMS using the normal electrolyte (1 M KOH in H₂O). DEMS measurements were subsequently carried out, first in 5% H₂¹⁸O in H₂O (¹⁸O-rich electrolyte) for 15 cycles, followed by another six cycles in normal electrolyte (1 M KOH in H₂O). ¹⁸O/¹⁶O mass spectrometric signals were not measured as they would yield a χ of only ~0.25% in 5% H₂¹⁸O in H₂O, which is too low for reliable evaluation (Figure S27-S29). Thus, Figure S29 shows how the fraction of $\chi(^{16}\text{O}^{18}\text{O})$ changes when the electrolyte is changed from ¹⁸O-rich (5% H₂¹⁸O) to normal electrolyte at various cycles. $\chi(^{16}\text{O}^{18}\text{O})$ is calculated from the following equation, $\chi(^{16}\text{O}^{18}\text{O}) = ^{18}\text{O}^{16}\text{O} / (^{18}\text{O}^{16}\text{O} + ^{16}\text{O}^{16}\text{O})$. Our DEMS data in Figure S29 (see below) show, after activation in the 5% H₂¹⁸O isotope labelled electrolyte, the molar fraction $\chi(^{16}\text{O}^{18}\text{O})$ ($m/z = 34$) is ~0.9 % in 100-cycle Co₂CrO₄ and ~0.6% for CoCr₂O₄ in the normal electrolyte (¹⁸O-lean electrolyte, blue-coloured regions), which is slightly higher than ~0.4, which is the $\chi(^{16}\text{O}^{18}\text{O})$ theoretical value from the normal water with 0.2% H₂¹⁸O natural abundance (Energy & Environmental Science, 15(2022)1988, Electrochemistry Communications, 9(2007)1969). This suggests that lattice oxygen mechanisms might occur in both samples. Because ¹⁸O-containing species would cover the surfaces of electrocatalysts in 5% H₂¹⁸O isotope-labelled electrolyte before cycling in normal water. Such ¹⁸O-containing species would generate a molar fraction of ¹⁸O/¹⁶O ($m/z = 34$) larger than ~0.4. However, it is difficult to make a conclusive statement as $\chi(^{16}\text{O}^{18}\text{O})$ for CoCr₂O₄ is close to ~0.4 with error bars. Additionally, oxygen evolution in 1 M KOH does not proceed via lattice exchange, although ¹⁸O/¹⁶O evolution was detected by performing DEMS measurements on model catalysts (Nat. Catal., 2018, 1, 820–829). The authors indicated that the oxidation of the electrolyte alone can explain the signal at $m/z = 34$, and there was no participation of lattice oxygen or intercalated water in the oxygen evolution. Therefore, we refrain from making conclusive statements regarding AEM, LOM based solely on DEMS measurements as no mechanisms can be confirmed solely based on DEMS measurements.

Additionally, we stress the critical role of the intercalated hydroxide and K ions in the activity of activated CoCr₂O₄ (in our second last paragraph of discussion on page 29-30) since a recent experimental coupled density functional theory study proposed that intercalated hydroxide ions can potentially deprotonate the hydroxyl groups of α -CoFe-LDHs, along with the oxidation of transition metals under the OER potential (Nature Communications, 11(2020)2522). In other words, the intercalated hydroxide ions in the interlayers can serve either as proton acceptors or proton transfer reagents. The kinetics of such a process is generally regarded as highly dependent on the electrolyte pH value. To investigate the pH-dependent OER activity characteristics, we performed CV measurements on 100-cycle CoCr₂O₄ and Co₂CrO₄ from 0.03 M KOH (pH 12.5) to 1.0 M KOH (pH 14) (Figure S26). Interestingly, 100-cycle CoCr₂O₄ shows a pH-dependent OER activity, while 100-cycle Co₂CrO₄ displays a weaker pH-dependent OER activity (Figure S26a-b). The pH dependence of OER activity indicates the

presence of non-concerted proton-electron transfer steps during OER, where the rate-limiting step is a proton transfer step or is preceded by the acid-base equilibrium. This result suggests that non-concerted proton-electron transfer steps likely occur for 100-cycle CoCr_2O_4 during OER, when the intercalated hydroxide ions promote the deprotonation of hydroxyl groups in (Co, Cr) oxyhydroxides by forming water molecules, thus enhancing OER activity.

Figure S27. DEMS results of the electrolyte exchange experiment conducted at Co_2CrO_4 . Top Panels: Faradaic current measured by the potentiostat. Centre: Mass spectrometric signal for mass 32. Bottom: Mass spectrometric signal for mass 34. The mass spectrometer measures the signal in current, but Faraday's laws are not applicable. Left: measured in 1 M KOH featuring 5% H_2^{18}O . Right: measured in 1 M KOH in normal water featuring with natural isotope abundance. Conditions: 5 mV/s, electrolyte flow rate 5 $\mu\text{L/s}$. The electrolyte exchange was performed at 0.53 V vs. Hg/HgO and took 78 seconds. Prior to the experiment Co_2CrO_4 was activated by performing 100 cycles in 1 M KOH electrolyte with normal water.

Figure S28. DEMS results of the electrolyte exchange experiment conducted at CoCr_2O_4 . Top Panels: Faradaic current measured by the potentiostat. Centre: Mass spectrometric signal for mass 32. Bottom: Mass spectrometric signal for mass 34. The mass spectrometer measures the signal in current, but Faraday's laws are not applicable. Left: measured in 1 M KOH featuring 5% H_2^{18}O . Right: measured in 1 M KOH in normal water featuring with natural isotope abundance. Conditions: 10 mV/s, electrolyte flow rate 5 $\mu\text{L/s}$. The electrolyte exchange was conducted at 0.54 V vs. Hg/HgO and took 98 seconds. Prior to the experiment CoCr_2O_4 was activated by performing 100 cycles in the 1 M KOH electrolyte with normal water.

Figure S29. $\chi(^{16}\text{O}^{18}\text{O})$ changes with cycles, which is defined as the fraction of $^{18}\text{O}^{16}\text{O} \cdot 100 / (^{18}\text{O}^{16}\text{O} + ^{16}\text{O}^{16}\text{O})$ versus the cycle number. Between the 5th and the 6th the exchange from the electrolyte featuring 5% H_2^{18}O to the ^{18}O -lean electrolyte with natural abundance is performed. $\chi(^{16}\text{O}^{18}\text{O})$ was determined from the experimental values shown in Figure S27 and Figure S28, respectively.

10. There are some obvious grammatical errors in this manuscript. The order of the diagrams is also quite chaotic, for example, the arrangement order of the impedance diagrams.

We have thoroughly proofread the manuscript and made the necessary grammatical corrections. Additionally, the impedance spectroscopy measurement figure was created in new Figure S9.

Reviewer #3 (Remarks to the Author):

Dear authors, I thank you for the thoroughness and thoughtfulness of your rebuttal to the specific comments of reviewer #3. I believe that your response to each concern and the edits you've made adequately address the initial comments. I have no further concerns about the manuscript.

Thank you for the constructive comments.